# FROM ATTENTION TO ACTIVATION: UNRAVELING THE ENIGMAS OF LARGE LANGUAGE MODELS

**Prannay Kaul**[1]* **Chengcheng Ma**[2] **Ismail Elezi**[1]† **Jiankang Deng**[1]

[1]Huawei Noah's Ark Lab, London, UK
[2]Institute of Automation, Chinese Academy of Sciences (CASIA)

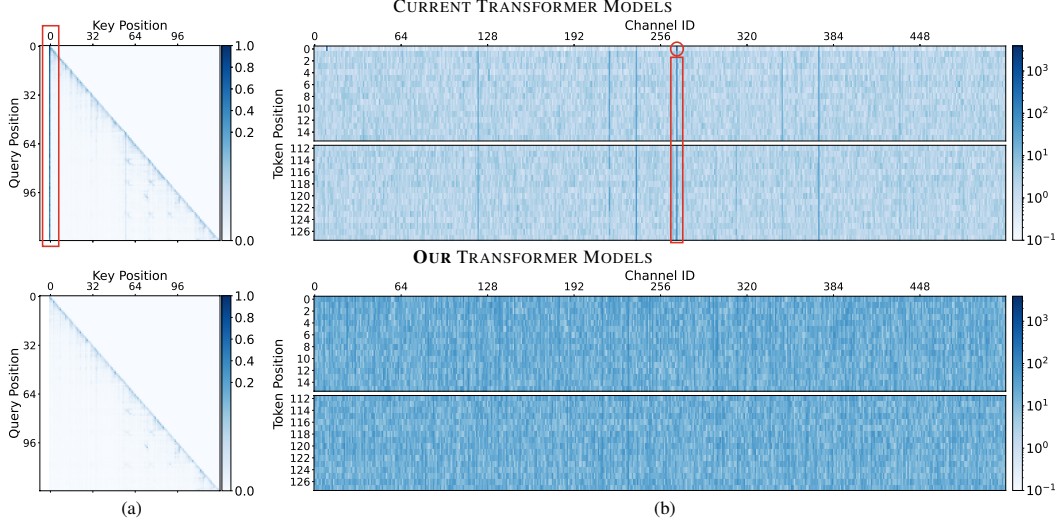

Figure 1: **(top) (a)** The mean attention map across all heads and layers of a GPT2-Medium model—the first token strangely dominates attention (boxed in red). **(b)** The mean hidden state across layers of the same model—outlier activations emerge in specific feature dimensions (boxed in red). The first token position exhibits the most extreme outlier activations—(circled in red). **(bottom) (a)** Replacing the canonical softmax function with our proposed *softmax-1* function eliminates the first token dominance. **(b)** Using our proposed optimiser, *OrthoAdam*, removes outlier activations *without any reduction in model performance*.

## ABSTRACT

We study two strange phenomena in auto-regressive Transformers: (1) the dominance of the first token in attention heads; (2) the occurrence of large outlier activations in the hidden states. We find that popular large language models, such as Llama attend maximally to the first token in 98% of attention heads, a behaviour we attribute to the softmax function. To mitigate this issue, we propose a reformulation of softmax to *softmax-1*. Furthermore, we identify adaptive optimisers, *e.g.*, Adam, as the primary contributor to the large outlier activations and introduce *OrthoAdam*, a novel optimiser that utilises orthogonal matrices to transform gradients, to address this issue. Finally, not only do our methods prevent these phenomena from occurring, but additionally, they enable Transformers to sustain their performance when quantised using basic algorithms, something that standard methods are unable to do. In summary, our methods reduce the attention proportion on the first token from 65% to 3.3%, the activation kurtosis in the hidden states from 1657 to 3.1, and perplexity penalty under 4-bit weight quantisation from 3565 to 0.3. Code is available at `https://github.com/prannaykaul/OrthoAdam`.

---

*Work conducted during internship
†Correspondence to `ismail.elezi@huawei.com`

# 1 INTRODUCTION

Transformers have revolutionised machine learning, achieving state-of-the-art performance across diverse domains, including language, vision and protein structure prediction (OpenAI, 2023; Carion et al., 2020; Jumper et al., 2021). However, the inner workings of auto-regressive Transformers remain enigmatic. Recent studies (Elhage et al., 2022; Olsson et al., 2022; Bansal et al., 2023) unravelled some of their complexities, yet we find that two surprising phenomena remain pervasive:

1. The strong, consistent dominance of the first token in attention maps—see top of Figure 1a.
2. The presence of outlier activation values, across sequence position, in specific feature channels of the hidden states (the intermediate features of each layer *after* the residual connection) that are orders of magnitude larger than other values—see top of Figure 1b.

We ask: What causes these phenomena? Are they essential to performant models? And, if not, how can we mitigate them?

These two phenomena are aesthetically curious, but also have important practical implications. For instance, Llama models (Touvron et al., 2023b; Dubey et al., 2024) exhibit the aforementioned first token dominance of attention, and so requiring complicated attention masking schemes to extend Llama models to tasks with long sequences (Xiao et al., 2024), *i.e.*, increase the maximum context length used during training. This is particularly crucial for instruction-tuned models where long conversations are desirable (Wei et al., 2022; Ouyang et al., 2022). Similarly, the presence of outlier activations leads to challenges in quantising large language models (LLMs). Large outlier activations increase the required quantisation range (to capture the outliers), resulting in low effective bits for the non-outlier activations, causing severe performance degradation post-quantisation. To address this issue, prior work has proposed mixed-precision decomposition of LLMs (Dettmers et al., 2022) or complex scaling of the weights and activations which must be learnt for each model (Xiao et al., 2023). Therefore, our additional motivation is to understand and mitigate these phenomena in a general manner, such that these issues are resolved *during training*.

We begin by examining the attention mechanism, and surprisingly find, across numerous input sequences, query tokens attend *most* to the first key token up to 98% of the time. This is striking considering the limited semantic information the first token typically contains—it is often a special token indicating the start of a sequence, such as `<bos>`. We explore explanations for this, ruling out positional encodings, non-linearity choice, or feature normalisation. Ultimately, we identify the softmax function used in the attention mechanism combined with causal masking as the root cause—excessive attention on the first key token demonstrates an attention head effectively doing nothing (Bondarenko et al., 2023; Clark et al., 2019). The first token is privileged due to causal masking; it is the only key token to which all query tokens can attend. We propose an adjustment to softmax as a solution, *softmax-1*, removing first token dominance in attention (bottom of Figure 1a).

Despite removing first token dominance in attention, using *softmax-1*, we find that the problem of outlier activations in the hidden states persists. Once again, we investigate potential causes of this issue and discover the outliers are primarily caused by the use of adaptive optimisers, *e.g.*, Adam (Kingma & Ba, 2015). Specifically, our experiments show the exponential decaying averages of first and second moments of gradients result in outlier activations. To tackle this, we propose a novel optimiser, *OrthoAdam*, which transforms computed gradients using orthogonal matrices, thus storing gradients in an alternative basis to the model parameters. Our results demonstrate this optimiser eliminates the outliers in the hidden states of Transformers (bottom of Figure 1b).

| Model | #Parameters | PPL | |
|---|---|---|---|
| | | FP16 | 4-bit Quant |
| GPT2-Small | 137M | 37.8 | 4456.1 |
| GPT2-Medium | 350M | 28.8 | 2435.3 |
| GPT2-Large | 812M | 25.2 | 571.0 |
| GPT2-XL | 1.6B | 23.2 | 7981.8 |
| Llama2-7B | 6.7B | 7.7 | 191477.5 |
| Llama3.1-8B | 8B | 10.2 | 2087638.0 |
| GPT2 (Ours) | 350M | 16.3 | 17.1 |
| GPT2 (Ours) | 1.4B | 13.3 | 13.6 |

Table 1: Due to surprising phenomena in Transformer models, basic zeropoint 4-bit weight quantisation leads to catastrophic performance degradation. Our models trained with *softmax-1* and *OrthoAdam* exhibit improved robustness to quantisation.

Our research extends beyond aesthetic curiosities. While LLMs perform well despite first token dominance and outlier activations, they lead to practical challenges. Although advanced schemes have been developed to enable quantised LLMs to maintain their performance, we show our approach enables LLMs to maintain their performance with the most basic quantisation methods, such as per-tensor 8-bit *absmax* weight/activation quantisation and 4-bit *zeropoint* weight quantisation. Thus, our investigation helps to better understand Transformers, while offering practical benefits.

In summary, our **contributions** are as follows:

- We **identify** the dominance of the first token in attention and the occurrence of outliers in the activations of the hidden states as significant issues in auto-regressive Transformers.

- We **propose** two simple, effective solutions: a reformulation of the softmax function, *softmax-1*, to address the former issue, and a novel optimiser, *OrthoAdam*, to tackle the latter. Our methods reduce first token attention from 65% to 3.3% and activation kurtosis from 1657 to 3.1.

- We **demonstrate** that these proposals not only resolve the identified problems but also lead to practical improvements in the performance of Transformers under 8-bit weight/activation and 4-bit weight quantisation. Our method reduces the perplexity penalty under 4-bit weight quantisation from 3565 to 0.3.

## 2 PROBLEM DEFINITION

This work investigates the two most prominent and strange phenomena of auto-regressive Transformer models: (1) strong, consistent dominance of the first token in the attention maps; (2) strong, consistent outlier activations in specific feature channels of the hidden states (the intermediate features computed *immediately after* the residual connections)—see top of Figure 1. We aim to understand the cause of these phenomena and to propose individual solutions for each of them. They have been investigated or commented on previously (Bondarenko et al., 2023; Dettmers et al., 2022; Xiao et al., 2023), but our work reaches different conclusions on the causes and suggests novel solutions. We start by describing these two anomalies in detail.

### 2.1 FIRST TOKEN DOMINANCE IN ATTENTION MAPS

The top of Figure 1a shows the attention map, averaged across all layers and heads, of a Transformer model, specifically a pretrained GPT2-Medium model (Radford et al., 2019), for a single real natural language sequence. Strangely, in this average attention map the key corresponding to the first token receives the highest attention across all queries. Quantitatively, we find the first key token is the most attended to key in 76% of (query, head) pairs and receives 52% of all attention, when evaluating on the `en` validation split of the C4 dataset (Raffel et al., 2020; Dodge et al., 2021). This behaviour is consistent across different LLMs, including the Llama series (Touvron et al., 2023b; Dubey et al., 2024), DeepSeek (Liu et al., 2024), and the GPT2 series (Radford et al., 2019). See Appendix K for detailed examples of attention maps for these models.

Attention is a key component of the Transformer architecture, and work on the interpretability of LLMs often focuses on analysing attention (Elhage et al., 2021). Moreover, many models, such as Llama2, use a special token for the beginning of a sequence (the `<bos>` token), which is *always* the first token in an input sequence. This makes first token dominance particularly puzzling, as such models should learn the initial input structure easily. We hypothesise that this phenomenon in the attention mechanism is a symptom of a fundamental problem in the Transformer architecture and is not necessary for a performant auto-regressive Transformer.

### 2.2 OUTLIER ACTIVATIONS IN THE HIDDEN STATES

The top of Figure 1b shows the activation magnitude in the hidden states of a pretrained GPT2-Medium model. We observe the hidden states of the Transformer model exhibit consistent outlier activations in specific feature channels across all token positions (boxed red), with the most extreme outliers occurring in the first token position (circled red). Once again, this behaviour is consistent across different LLMs and is invariant to the input sequence, *i.e.*, the same feature channels *always* exhibit outlier activations. See Appendix J for examples of hidden states in pretrained models.

From a practical perspective, these outlier activations are problematic with regards to quantising models for deployment (Lin et al., 2021; Dettmers et al., 2022). However, from a theoretical perspective, the cause of these outlier activations is not well understood. Previous works, have suggested these outliers are related to first token domination in attention maps (Xiao et al., 2023; Bondarenko et al., 2023). This is plausible for the most extreme outliers observed in the first token position, but it does not explain the outlier activations observed across all token positions. In this work, we show the two phenomena are unrelated and separate solutions are required to address each.

## 3    METHOD: FIRST TOKEN DOMINANCE OF ATTENTION MAPS

We start by eliminating plausible causes of the first phenomenon of interest: first token dominance of attention maps. We mainly consider GPT2 as a representative auto-regressive Transformer, because of its simplicity, but also consider the more recent Llama2 model to narrow down possible causes of this phenomenon. For all experiments, unless mentioned otherwise, we use a GPT2 model with 130M parameters, trained on the `en` split of the C4 dataset.

### 3.1    ELIMINATING CERTAIN CAUSES OF FIRST TOKEN DOMINANCE OF ATTENTION MAPS

Both GPT2 and Llama exhibit first token dominance in attention maps. Thus, we can rule out parts of their architecture that are different:

- *Positional encoding.* Llama models use Rotary Positional Encodings (RoPE) (Su et al., 2024), while GPT2 models uses learnt absolute positional encodings (Vaswani et al., 2017).
- *Initial token.* Llama models use a `<bos>` token to denote the beginning of a sequence, while GPT2 models do not.
- *Activation function.* Llama models use SiLU (Elfwing et al., 2018) in the feedforward layers, while GPT2 models use GeLU (Hendrycks & Gimpel, 2016).
- *Feature Normalisation.* Llama models use RMSNorm (Zhang & Sennrich, 2019), while GPT2 models use LayerNorm (Ba et al., 2016).

Note that Llama and GPT2 use different positional encoding, but it is possible that any form of positional encoding might be cause of first token dominance. To test this possibility, we train a GPT2 model *without any positional encodings* and observe the attention maps. We find equivalently trained GPT2 models with/without positional encodings exhibit first token dominance in 33%/20% of (query, head) pairs and allocate 17%/10% of all attention to the first token. Thus, we conclude that positional encodings are not the cause of these anomalies. The models mentioned here are trained for relatively few steps and first token dominance is more pronounced in our longer-trained models and in publicly available pretrained models.

### 3.2    REMOVING FIRST TOKEN DOMINANCE OF ATTENTION MAPS

After eliminating the above causes, we have two aspects of Transformers that could cause first token dominance: (1) causal masking in self-attention; and (2) softmax normalisation in attention heads.

Consider the self-attention mechanism on the initial token in a causal Transformer. The first *query* token can only attend to its own key token and therefore it receives an attention score of 1, due to softmax normalisation. Similarly, the second query can only attend to the first two keys, whose attention scores must sum to 1. Prior work establishes attention heads specialise to concepts or concept groups (Bansal et al., 2023; Elhage et al., 2022). However, given a query irrelevant to the specialisation of an attention head, it must still allocate attention across the keys summing up to 1. Moreover, causal masking privileges the first key token above all others; it is the only key token to which *all* tokens can attend. This explains why the *first* token specifically dominates attention maps.

Clearly, a particular attention head should be able to *attend nowhere* if no relevant information is present. Thus, we modify the softmax function to the following:

$$\text{softmax-1}(x_i) = \frac{\exp(x_i)}{1 + \sum_{j=1}^{L} \exp(x_j)}; \qquad \sum_{i=1}^{L} \text{softmax-1}(x_i) < 1 \qquad (1)$$

This modification removes the strict enforcement of attention scores summing to 1, allowing the model to allocate attention as it sees fit, including having low attention scores everywhere. From a registers/attention sink perspective (Darcet et al., 2024; Xiao et al., 2024), the 1 in the denominator is equivalent to a register/attention sink key token which has 0 *dot product* with any query token.

**Validating the hypothesis.**    We train two GPT2 models, one with canonical softmax and one with softmax-1, keeping all other variables the same. The model trained with canonical softmax attention exhibits first token dominance; the first key token is the most attended to key in 53% of (query, head) pairs. However, the model trained with softmax-1 lowers this to just 2%. Furthermore, with canonical softmax 46% of all attention is received by the first key, while using softmax-1 lowers this to 4%, thereby validating our idea.

The difference in attention maps between canonical softmax and softmax-1 is shown in Figure 1a, which compares the attention maps of two models on the same input sequence. Furthermore, we find using softmax-1 has no effect on training stability, convergence or model performance (see Appendix L for the training curves of all our trained models).

**What if causal masking is relaxed?** To verify the first token is privileged by causal masking, causing *first* token dominance, we train a GPT2 model with canonical softmax in which causal masking is removed for the first 10 tokens. (the loss function is appropriately modified). This way, all queries can attend to the first 10 keys. Figure 2 shows one of these tokens (this happens with uniform distribution) still dominates the attention map.

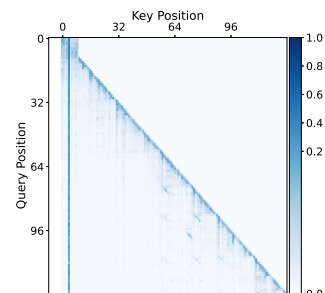

Figure 2: Relaxing causal masking leads to attention domination by a different token

## 4 METHOD: OUTLIER ACTIVATIONS

To quantitatively establish the extent of outliers in the hidden states, we use kurtosis. Kurtosis, in this case, is a measure of tail heaviness of a set of activation values. Activations which are normally distributed have a kurtosis of $\sim 3$, while higher kurtosis indicates a heavier-tailed distribution (*e.g.*, the exponential distribution) and lower kurtosis indicates a lighter-tailed distribution (*e.g.*, the uniform distribution). Given hidden states $\mathbf{X} \in \mathbb{R}^{M \times L \times D}$ of a Transformer model, where $M$ is the number of layers, $L$ is the number of tokens and $D$ is the number of feature channels, we compute the per-layer, per-position kurtosis of the hidden states as:

$$\kappa_{m,l} = \mathrm{Kurt}_{m,l}\left[X_{m,l,d}\right] = \frac{\mathbb{E}_d[(X_{m,l,d} - \mu_{m,l})^4]}{\mathbb{E}_d[(X_{m,l,d} - \mu_{m,l})^2]^2}, \quad \text{where} \quad \mu_{m,l} = \mathbb{E}_d[\mathbf{X}_{m,l,d}] \tag{2}$$

where $X_{m,l,d}$ is the hidden state at layer $m$ at position $l$ for feature $d$, and $\mu_{m,l}$ is the mean hidden state value at layer $m$ at position $l$.

### 4.1 ELIMINATING CERTAIN CAUSES OF OUTLIER ACTIVATIONS

We start by eliminating certain causes which could lead to the presence of outlier activations.

**Feedforward Layer Biases.** GPT2 uses biases in all feedforward layers, while Llama uses none, therefore it is unlikely feedforward layer biases cause of outlier activations.

**Normalisation Layers.** GPT2 uses LayerNorm (Ba et al., 2016) while LLama uses RMSNorm (Zhang & Sennrich, 2019), which both learn individual scaling parameters for each feature channel, potentially causing the outlier activations. To remove such an effect, we replace LayerNorm in our trained GPT2 models with an RMSNorm version which applies a *single* global scale instead of per-channel scaling, and call it "RMSNormSingle"—similar to "Simple RMSNorm" from Qin et al. (2023) which has no learned parameters. We find outlier activations persist in the hidden states of a GPT2 model with RMSNormSingle. In Table 5 we show kurtosis remains high in models trained without biases and/or with RMSNormSingle.

**Optimiser.** Most Transformer models are trained with Adam (Kingma & Ba, 2015) or a variant. These optimisers track the first and second moments of the computed gradients using exponential moving averages, tracking these moments at a parameter level. The main hyperparameters of Adam-like optimisers are $\beta_1$ and $\beta_2$, which control the decay rates of the first and second moments, respectively. If $\beta_2 = 0$, only the first moment of the gradients is tracked, resembling stochastic gradient descent (SGD) with momentum. Conversely, if $\beta_1 = 0$, only the second moment of the gradients is tracked, resembling RMSProp. We suspect that given the optimiser tracks moments in the same basis as the model parameters, it is the most likely cause of the outlier activations in the hidden states auto-regressive Transformer models.

**Validating the hypothesis.** We train a series of GPT2 models using Adam, RMSProp, SGD with and without momentum, tuning the learning rate and training schedule to encourage convergence. The model trained with SGD has the slowest convergence and highest validation perplexity, while the model trained with Adam converges the fastest and has the lowest perplexity. However, we find

models trained with Adam and RMSProp have high kurtosis, 140 and 70, respectively, while training with SGD gives a kurtosis of $\sim 3.0$. We provide these results in our ablation study (Section 5.3).

## 4.2 ORTHOADAM

The previous section leaves an important question for training Transformer models: *"How can we train a model with an optimiser which has the speed and convergence properties of Adam, but produces activations properties similar to SGD"?*

Optimisers which track exponential decaying averages of the first and/or second moments of the gradients lead to outlier activations in the hidden states of Transformer models. Moreover, in the models trained above, the largest absolute *parameter* values correspond to the features which exhibit outlier activations in the hidden states, *i.e.*, if outlier activations occur in feature channel $i$ of the hidden states, the largest model parameter values correspond to specific weights which act on feature channel $i$ of the hidden states, *e.g.*, the $i^{\text{th}}$ output channel of the output projection weights of the attention/MLP layers. Therefore, to arrive at these large model parameter values, the optimiser (*e.g.*, Adam) must provide relatively large updates to these specific parameters and not others. We note here that Adam and similar optimisers calculate gradient moments in the same basis as the model parameters. Additionally, given the channels which contain outlier activations appear invariant to the input sequence, we hypothesise that these channels are an artefact of the optimiser and do not correspond to any meaningful feature in the input sequence—see Appendix J for plots of the hidden states of pretrained models with different input sequences. Given these observations, we discuss an idealised case of observed hidden states below, and show how orthogonal transformations can be used to reduce outlier activations.

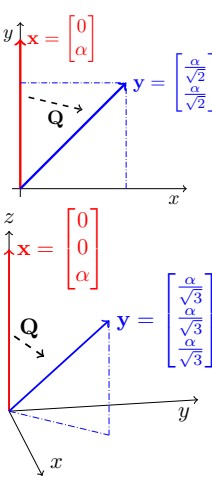

Figure 3: Rotating vectors with dominant components leads to a reduction in the maximum absolute value.

Consider a $D$-dimensional vector, $\mathbf{x} = \alpha \mathbf{e}_i + \mathbf{z}$, where $\mathbf{e}_i$ is the $i^{\text{th}}$ unit vector in the standard basis, $\mathbf{x} \in \mathbb{R}^D$, $\alpha \in \mathbb{R}^+$, $\alpha \gg 1$ and $\mathbf{z} \sim \mathcal{N}(\mathbf{0}, \mathbf{I})$. The first term represents the single outlier activation specific to the $i^{\text{th}}$ channel and the second term represents the "informative" activations. The vector $\mathbf{x}$ represents the hidden states of a Transformer model with high kurtosis. This simplified model makes two assumptions: (1) there is a single outlier activation channel; and (2) the informative activations are normally distributed.

For values of $D$ similar to that of Transformer models, *i.e.*, $D \approx [10^3, 10^5]$, $\text{Kurt}[x_j] = O(D)$. Therefore, we expect larger Transformer models of a given architecture to have larger kurtosis in their hidden states. Moreover, the ratio of the $\ell_\infty$-norm to the $\ell_2$-norm of the hidden states in our simplified model, $\frac{\|\mathbf{x}\|_\infty^2}{\|\mathbf{x}\|_2^2}$, is close to $1$. This ratio is another proxy for the extent of outliers.

Now we consider the effect of an appropriate orthogonal transformation on the vector $\mathbf{x}$. Let $\mathbf{Q} \in \mathbb{R}^{D \times D}$ be an orthogonal matrix, and $\mathbf{y} = \mathbf{Q}\mathbf{x}$. Under a particular orthogonal transformation, $\frac{\|\mathbf{y}\|_\infty^2}{\|\mathbf{y}\|_2^2} \approx \frac{1}{D}$ and $\text{Kurt}[y_j] = 3$. The orthogonal transformation which achieves this is one which rotates the vector $\mathbf{x}$ such that $\mathbf{Q}\mathbf{e}_i = \frac{1}{\sqrt{D}}\mathbf{1}$. Figure 3 illustrates this rotation process in 2D and 3D. The kurtosis and norm ratio results quoted in this section are derived in Appendix G and Appendix H, respectively, and are shown to be empirically valid for models we train from the plots in Appendix I.2 and Appendix I.3, respectively.

One option is to apply orthogonal transformations directly to the hidden states of the model, *i.e.*, make $\mathbf{Q}$ part of the model parameters that are kept fixed during training. Instead, we propose a novel optimizer, *OrthoAdam*, which applies orthogonal transformations to incoming gradients such that the moment calculations (which our experiments in Table 3 show are the key factor in producing outlier activations) are performed in a different basis to the model parameters to prevent gradient updates to any particular set of parameters which lead to outlier activations. We provide the full algorithm in Algorithm 1.

In our experiments, we randomly sample the orthogonal matrix for each parameter (which remains fixed during the training of the model). We find that using *OrthoAdam* leads to a significant re-

---

**Algorithm 1** OrthoAdam, our proposed optimiser for reducing activation outliers. $\bar{\mathbf{g}}_t^2$ is the element-wise square $\bar{\mathbf{g}}_t \odot \bar{\mathbf{g}}_t$. With $\beta_1^t$ and $\beta_2^t$ we mean $\beta_1$ and $\beta_2$ taken to the power of $t$.

---

**given** learning rate: $\eta = 0.001$, first moment decay rate: $\beta_1 = 0.9$, second moment decay rate: $\beta_2 = 0.999$, numerical epsilon: $\epsilon = 10^{-8}$

**initialise** time step: $t \leftarrow 0$, parameter vector: $\theta_{t=0} \in \mathbb{R}^n$, first moment vector: $\bar{\mathbf{m}}_{t=0} \leftarrow \mathbf{0}$, second moment vector: $\bar{\mathbf{v}}_{t=0} \leftarrow \mathbf{0}$, schedule multiplier: $\lambda_{t=0} \in \mathbb{R}$, unique orthogonal matrix: $\mathbf{Q} \in \mathcal{O}^n$

**repeat**

$\quad t \leftarrow t + 1$

$\quad \nabla f_t(\theta_{t-1}) \leftarrow \text{SelectBatch}(\theta_{t-1})$ ▷ select batch and calculate gradient

$\quad \mathbf{g}_t \leftarrow \nabla f_t(\theta_{t-1})$ ▷ store the gradient in model parameter basis

$\quad \bar{\mathbf{g}}_t \leftarrow \text{MatMul}(\mathbf{Q}, \mathbf{g}_t)$ ▷ transform gradient into unique optimiser basis

$\quad \bar{\mathbf{m}}_t \leftarrow \beta_1 \bar{\mathbf{m}}_{t-1} + (1 - \beta_1)\bar{\mathbf{g}}_t$ ▷ update biased first moment estimate

$\quad \bar{\mathbf{v}}_t \leftarrow \beta_2 \bar{\mathbf{v}}_{t-1} + (1 - \beta_2)\bar{\mathbf{g}}_t^2$ ▷ update biased second raw moment estimate

$\quad \hat{\mathbf{m}}_t \leftarrow \bar{\mathbf{m}}_t / (1 - \beta_1^t)$ ▷ compute bias-corrected first moment estimate

$\quad \hat{\mathbf{v}}_t \leftarrow \bar{\mathbf{v}}_t / (1 - \beta_2^t)$ ▷ compute bias-corrected second raw moment estimate

$\quad \bar{\mathbf{s}}_t \leftarrow \hat{\mathbf{m}}_t / (\sqrt{\hat{\mathbf{v}}_t} + \epsilon)$ ▷ calculate the update step in unique optimizer basis

$\quad \mathbf{s}_t \leftarrow \text{MatMul}(\mathbf{Q}^T, \bar{\mathbf{s}})$ ▷ transform the update step back to model parameter basis

$\quad \lambda_t \leftarrow \text{SetScheduleMultiplier}(t)$ ▷ can be fixed, decay, or also be used for warm restarts

$\quad \theta_t \leftarrow \theta_{t-1} - \lambda_t \eta \mathbf{s}_t$ ▷ apply parameter update

**until** *stopping criterion is met*

**return** optimised parameters $\theta_t$

---

duction in the kurtosis of hidden states in Transformer models, effectively eliminating the outlier activations. This is shown qualitatively at the top of Figure 1b, where feature channels with high absolute activation values in the hidden states *are no longer present* across all token positions, and quantitatively in Table 2 showing the kurtosis of hidden states in models trained with *OrthoAdam* is close to 3, with *no performance penalty*.

## 5 EXPERIMENTS

**Datasets.** We train all models on the `en` training split of the C4 dataset (Dodge et al., 2021; Raffel et al., 2020) and evaluate on 100000 samples from the validation `en` split.

**Models.** We train GPT2 models with ∼{60M, 130M, 350M, 1.4B} parameters and Llama2 models with ∼130M parameters. Apart from changing the softmax function, the only other changes we make to the model architectures are the use of RMSNormSingle and we do not use biases in feedforward layers. We ablate these changes in the ablation study at the end of this section.

**Training.** Unless stated otherwise, we use a batch size of 512 and a cosine learning rate schedule with linear warmup for {1000, 2000, 6000, 10000} steps for models with {60M, 130M, 350M, 1.4B} parameters respectively, with a maximum learning rate of $10^{-3}$. We train models with {60M, 130M, 350M, 1.4B} parameters for {160k, 320k, 960k, 600k} steps respectively. Note that we use a reduced number of steps for the 1.4B parameter model due to computational constraints. In the ablation study, we train GPT2 models with 130M parameters for 40k steps only.

**Metrics.** We evaluate our experiments in the following metrics: (1) the perplexity (PPL) of models on the validation set; (2) the mean kurtosis across all layers of the model (evaluated separately for the first token and the remaining tokens); (3) the maximum absolute activation across all layers of the model (again evaluated separately); (4) the percentage of (query, head) pairs in which the first key token is the most attended to key token. We calculate (1) to ensure our method *at least maintains* the vanilla language model performance, *i.e.*, to ensure the model is not harmed by softmax-1 or OrthoAdam. (2) and (3) show quantitatively the extent to which outlier activations are present in the hidden states. Finally, (4) shows the extent to which the first token dominates attention in the model.

### 5.1 MAIN RESULTS

We show the results of softmax-1 and OrthoAdam used to train GPT2 and Llama2 models in Table 2. We observe that across both model architectures and all sizes, the evaluated PPL is the same or slightly lower when comparing a model with softmax-1 and trained with OrthoAdam to the vanilla model with neither, indicating that our method does not change model performance. Despite no

| Model | #Parameters | Softmax+1? | OrthoAdam? | PPL | Kurtosis | | Activation Value | | %First Attn |
|---|---|---|---|---|---|---|---|---|---|
| | | | | | $\mathbb{E}_m[\kappa_{m,1}]$ | $\mathbb{E}_m[\kappa_{m,>1}]$ | $\mathbb{E}_m[\|X_{m,1,d}\|]$ | $\mathbb{E}_m[\|X_{m,>1,d}\|]$ | |
| GPT2* | 60M | ✗ | ✗ | 31.9 | 313.8 | 77.9 | 1856.1 | 266.6 | 0.489 |
| | | ✓ | ✗ | 31.6 | 105.6 | 81.4 | 304.9 | 259.0 | 0.021 |
| | | ✗ | ✓ | 32.4 | 260.8 | 10.6 | 1419.9 | 114.7 | 0.365 |
| | | ✓ | ✓ | 31.8 | 7.6 | 7.0 | 92.8 | 87.8 | 0.019 |
| | 130M | ✗ | ✗ | 22.9 | 514.9 | 141.5 | 7018.1 | 1014.8 | 0.527 |
| | | ✓ | ✗ | 22.7 | 175.4 | 144.2 | 1134.3 | 967.5 | 0.024 |
| | | ✗ | ✓ | 23.1 | 446.4 | 20.2 | 4285.0 | 433.4 | 0.424 |
| | | ✓ | ✓ | 22.8 | 10.1 | 7.3 | 318.1 | 261.6 | 0.019 |
| | 350M | ✗ | ✗ | 16.4 | 820.3 | 161.8 | 40196.0 | 3801.1 | 0.579 |
| | | ✓ | ✓ | 16.3 | 3.1 | 3.1 | 388.1 | 333.3 | 0.021 |
| | 1.4B | ✗ | ✗ | 13.4 | 1656.5 | 351.9 | 56798.3 | 7051.2 | 0.648 |
| | | ✓ | ✓ | 13.3 | 3.1 | 3.0 | 181.9 | 132.1 | 0.033 |
| Llama2 | 130M | ✗ | ✗ | 17.4 | 435.0 | 170.0 | 4622.7 | 1627.4 | 0.105 |
| | | ✓ | ✗ | 17.2 | 208.2 | 181.2 | 1340.4 | 1229.5 | 0.016 |
| | | ✗ | ✓ | 17.4 | 435.8 | 169.5 | 4685.9 | 1629.1 | 0.103 |
| | | ✓ | ✓ | 17.3 | 4.2 | 6.9 | 161.1 | 157.0 | 0.017 |

Table 2: Main results showing the impact of *softmax-1* and *OrthoAdam* on trained GPT2 and Llama2 models. Utilising *softmax-1* and *OrthoAdam*, significantly reduces the kurtosis and the max activation values of hidden states. Using *softmax-1* only is sufficient to reduce first token dominance in attention. We generally find that all combinations of *softmax-1* and/or *OrthoAdam* at a given model size lead to similar performance. $\mathbb{E}_m[\kappa_{m,1}]$: mean kurtosis of the first token; $\mathbb{E}_m[\kappa_{m,>1}]$: mean kurtosis of all other tokens; $\mathbb{E}_m[\|X_{m,1,d}\|]$: mean max absolute activation value of the first token; $\mathbb{E}_m[\|X_{m,>1,d}\|]$: mean max absolute activation value of all other tokens. All values are averaged across all layers.

significant change in PPL, each of our proposed methods lead to a significant reduction in outlier activations in the hidden states (shown by a considerably lower mean layer kurtosis and maximum absolute activation), with the largest reduction observed when both softmax-1 and OrthoAdam are used. In particular, for GPT-2 models with 60M, 130M, 350M and 1.4B parameters, the kurtosis without our modifications were 77.9, 141.5, 161.8 and 351.0, while after our modification they drop to 7, 7.3, 3.1, and 3.0. We observe similar results for Llama2-130M where the perplexity is around the same as the original version, but kurtosis is reduced from 170 to 6.9. Similar to kurtosis, in all cases we see a significant reduction of the mean activation value. Furthermore, we also observe the drastic drop in first token attention. While the vanilla versions of the model have maximal first token attention of up to $64.8\%$, after our modification, it is reduced to 1-3%.

## 5.2 Quantisation

We quantise trained models using *Absmax* and *Zeropoint* quantisation. *Absmax quantisation* scales a given tensor (weight or activation) using the absolute maximum absolute value. On the other hand, *Zeropoint quantisation* shifts the quantised tensor such that the minimum tensor value is the minimum representable value. See Dettmers et al. (2022) for exact details on the quantisation schemes.

**Experimental Setup.** We quantise the trained models using Absmax quantisation using 8-bit integers and the more powerful Zeropoint quantisation using 4-bit integers. In the case of Absmax quantisation, we use 3 different configurations: (1) *fine* quantisation, where "per-channel" scaling is used for input activations and weights; (2) *moderate* quantisation, with "per-tensor" scaling for input activations and weights; and (3) *coarse* quantisation, with "per-tensor" scaling for input *and output* activations and weights. In the case of Zeropoint quantisation, we use a single configuration where "per-channel" scaling is used for *weights only*. We only quantize the linear layers, while the embeddings, normalisation layers and softmax activations are not quantised.

**Results.** In Table 3 we show the results of quantising the trained models using Absmax and Zeropoint quantisation. We experimentally confirm that in all cases, models trained with softmax-1 and OrthoAdam are more robust to Absmax quantisation schemes than models trained with the canonical softmax function and Adam. The difference in performance is most pronounced when using moderate and coarse quantisation schemes—models trained with softmax-1 and OrthoAdam are able to maintain performance while models trained with canonical softmax and Adam suffer a significant degradation in performance. In particular, in the coarse setting, our method outperforms the baseline by up to 36.12 points. For Zeropoint quantisation, we observe that all GPT2 models trained with canonical softmax and Adam become unusable when using 4-bit integer weight quantisation, while models trained with softmax-1 and OrthoAdam suffer only a small drop in performance. Llama2 models in both cases remain usable after quantisation, but the performance drop is more pronounced when using the canonical softmax function and Adam.

| Model | #Parameters | OA + S1? | PPL | | | | | | | | |
| --- | --- | --- | --- | --- | --- | --- | --- | --- | --- | --- | --- |
| | | | full | coarse | Δ | moderate | Δ | fine | Δ | 4-bit | Δ |
| GPT2 | 60M | ✗ | 31.88 | 43.53 | 11.65 | 34.87 | 2.99 | 32.15 | 0.27 | 68.5 | 36.6 |
| | | ✓ | 31.83 | 32.30 | 0.47 | 32.18 | 0.35 | 31.89 | 0.06 | 33.9 | 2.1 |
| | 130M | ✗ | 22.89 | 46.49 | 23.60 | 28.31 | 5.42 | 23.07 | 0.18 | 679.9 | 657.0 |
| | | ✓ | 22.78 | 23.21 | 0.43 | 23.10 | 0.32 | 22.83 | 0.05 | 24.0 | 1.2 |
| | 350M | ✗ | 16.37 | 52.49 | 36.12 | 19.92 | 3.55 | 16.50 | 0.13 | 118507.1 | 118490.7 |
| | | ✓ | 16.31 | 16.50 | 0.19 | 16.46 | 0.15 | 16.33 | 0.02 | 17.1 | 0.8 |
| | 1.4B | ✗ | 13.44 | 45.05 | 31.61 | 15.19 | 1.75 | 13.68 | 0.24 | 3577.7 | 3564.3 |
| | | ✓ | 13.33 | 13.45 | 0.12 | 13.43 | 0.10 | 13.34 | 0.01 | 13.6 | 0.2 |
| Llama2 | 130M | ✗ | 17.39 | 43.61 | 26.22 | 24.46 | 7.07 | 17.69 | 0.30 | 21.5 | 4.1 |
| | | ✓ | 17.31 | 20.85 | 3.54 | 20.11 | 2.80 | 17.38 | 0.07 | 19.7 | 2.4 |

Table 3: Performance of our trained models under various quantisation settings. When using *OrthoAdam* and *softmax-1* (OA + S1), the performance penalty due to quantisation is significantly reduced. The benefits of our proposed changes are more pronounced under more aggressive quantisation settings, *i.e.*, 4-bit weight and coarse 8-bit weight/activation quantisation (vanilla models exhibit catastrophic performance degradation).

## 5.3 ABLATION STUDY

Table 5 shows the results of an ablation study on GPT2 models with 130M parameters. As expected from the discussion in Section 3, we find removing biases from linear layers and varying the position encodings does not prevent first token domination—we see a small reduction in first token domination when positional encodings are removed. Using softmax-1, first token dominance is mitigated with only ∼2% of (query, head) pairs having the first key token as the most attended to key token.

Switching from LayerNorm to RMSNorm with a learnt scale for each channel (RMSNorm-M, the normalisation used in Llama2) does not reduce the prevalence of outlier activations in the hidden states. However, switching to RMSNorm with a single learnt scale (RMSNorm-S) reduces the mean layer kurtosis and max absolute activation by ∼40%, which remains high. In all of the above cases in which Adam is used as the optimiser, we observe similar perplexity to the initial model (top row). Slight exceptions being the use of rotary and no positional encodings, in which perplexity reduces and increases by 1.3 and 0.5, respectively.

Changing the optimiser to RMSProp leads to increased perplexity (0.5 compared to the initial model), reduced mean layer kurtosis and max absolute activation, by ∼50% and ∼30%, respectively, when comparing to the equivalent model trained with Adam. In contrast to all previous cases, using SGD with/without momentum (on a longer schedule to encourage convergence), leads to a significant decrease in mean layer kurtosis and max absolute activation, by up to 98% and 97%, respectively, when comparing to the equivalent model trained with Adam.

| Model | Speed | VRAM |
| --- | --- | --- |
| 60m-vanilla | 14 iter/sec | 16.4GB |
| 60m-S1+OA | 12 iter/sec | 16.8GB |
| 130m-vanilla | 7.5 iter/sec | 22.6GB |
| 130m-S1+OA | 6.0 iter/sec | 23.3GB |
| 350m-vanilla | 3.3 iter/sec | 46.6GB |
| 350m-S1+OA | 3.0 iter/sec | 47.3GB |
| 1.4B-vanilla | 1.0 iter/sec | 61.9GB |
| 1.4B-S1+OA | 1.1 iter/sec | 65.0GB |

Table 4: Time and memory performance.

However, using SGD requires a significantly longer training schedule to approach initial model performance. Using SGD without momentum leads to a significantly higher perplexity (6.8 compared to the initial model). This finding confirms the importance of the optimiser in causing outlier activations in the hidden states.

Using OrthoAdam yields the desirable results from SGD without momentum—namely a significant decrease in mean layer kurtosis (140 to 3.0) and max absolute activation (432 to 43.5) and the desirable results from Adam—namely similar perplexity to a model trained with Adam and therefore much faster and better convergence than SGD without momentum.

The final three rows of Table 5 show that using OrthoAdam with softmax-1 and RMSNorm-S leads to the most desirable results, and critically the removal of softmax-1 and the use of LayerNorm or RMSNorm-M reintroduces first token attention dominance and outlier activations, respectively.

**Time and memory increase.** In Table 4, we show that our modifications come with a small and tolerable increase in time and memory.

**Increasing the sequence length.** In Table 6 of Appendix, we show that our method is robust to increasing the training sequence length. We show results with models trained in 512 and 1024 sequence length, getting similar results to those of Table 3.

| Biases | Position Encoding | Normalisation | Optimizer | Softmax+1? | PPL | Kurtosis | %First Attn | Max Abs. Act? |
|---|---|---|---|---|---|---|---|---|
| ✓ | Absolute | LayerNorm | Adam | ✗ | 26.9 | 291.7 | 0.333 | 1675.9 |
| ✗ | Absolute | LayerNorm | Adam | ✗ | 26.9 | 263.7 | 0.308 | 1104.0 |
| ✗ | None | LayerNorm | Adam | ✗ | 27.4 | 283.3 | 0.197 | 1478.7 |
| ✗ | Rotary | LayerNorm | Adam | ✗ | 25.6 | 391.9 | 0.336 | 2577.4 |
| ✗ | Absolute | LayerNorm | Adam | ✓ | 26.5 | 244.7 | 0.022 | 648.6 |
| ✗ | Absolute | RMSNorm-M | Adam | ✓ | 26.6 | 230.4 | 0.026 | 628.6 |
| ✗ | Absolute | RMSNorm-S | Adam | ✓ | 26.6 | 140.0 | 0.020 | 432.0 |
| ✗ | Absolute | RMSNorm-S | RMSProp | ✓ | 27.4 | 70.5 | 0.021 | 302.2 |
| ✗ | Absolute | RMSNorm-S | SGD w/mom* | ✓ | 25.3 | 5.0 | 0.019 | 17.8 |
| ✗ | Absolute | RMSNorm-S | SGD w/o mom* | ✓ | 33.4 | 3.2 | 0.017 | 13.1 |
| ✗ | Absolute | RMSNorm-S | OrthoAdam | ✓ | 26.8 | 3.0 | 0.022 | 43.5 |
| ✗ | Absolute | RMSNorm-S | OrthoAdam | ✗ | 27.3 | 323.0 | 0.231 | 726.4 |
| ✗ | Absolute | RMSNorm-M | OrthoAdam | ✓ | 26.7 | 380.9 | 0.025 | 737.2 |
| ✗ | Absolute | LayerNorm | OrthoAdam | ✓ | 26.6 | 188.4 | 0.023 | 514.0 |

Table 5: Ablation study on the impact of various architectural choices on the performance of a GPT2 model with $sim$130M parameter model. *SGD models are trained for $8\times$ longer than the others to encourage convergence.

## 6 RELATED WORK

**Language Models.** Language models are based on Transformers (Vaswani et al., 2017). While there are Transformer-based LLMs that used the original encoder-decoder architecture such as T5 (Raffel et al., 2020), researchers developed models such as BERT (Devlin et al., 2019) and RoBERTa (Liu et al., 2019), which are encoder-only. However, most current LLMs such as the GPT (Radford et al., 2018; 2019; Brown et al., 2020) and Llama series (Touvron et al., 2023a;b; Dubey et al., 2024) use a decoder-only architecture. In our work, we focus on this variant using GPT2 and Llama.

**Attention Dominance.** Bondarenko et al. (2023) identify the dominance of bland tokens in the attention maps of the BERT Transformer, and suggest complex clipping schemes, additional hyperparameters, and a gating mechanism to to mitigate this issue. Other researchers found the same issue in long-range attention (Xiao et al., 2024) and found a workaround using "attention sinks" and discontinuous attention masking. In vision Transformers, Darcet et al. (2024) made the same observation and proposed a solution using "registers". In contrast, we find the root cause of this issue, the softmax in attention, and reformulate it to prevent the first token dominance happening.

**Outlier Activations.** Previous works have shown that in certain Transformer models which use post-normalisation the norm of the *weights* of the learnt model must increase (Arora et al., 2019; Soudry et al., 2018). However the same reasoning does not apply for most recent decoder-only Transformers which use pre-normalisation (Xiong et al., 2020)(*i.e.*, normalisation before the residual connection). A blog-post by Elhage et al. (2023) discusses the presence of outlier activations in the hidden states of Transformer models and rules out numerical precision as the cause. Another blog-post by Miller (2023) posits the activation outliers are caused by the attention mechanism, however, we find outliers and attention dominance are disjoint phenomena. He et al. (2024) identify the presence of outliers and propose an "Outlier Protected Transformer Block" which makes many architectural changes such as removing normalisation layers and severely downscaling the activations at the residual connection. In our contrast, similar to first token dominance, we first find the root cause of this strange behaviour, and then fix it without doing architecture changes.

**Outlier-Aware Quantisation.** The presence of outliers in the activations of the hidden states has led to a number of works, such as `LLM.int8` (Dettmers et al., 2022), per-embedding group quantisation (Bondarenko et al., 2021), and SmoothQuant (Xiao et al., 2023) propose varying quantisation schemes to handle the presence of outliers, which require calibration. In contrast, we eliminate the presence of outliers in our trained models thus enabling the use of the most basic quantisation schemes such as Absmax and Zeropoint quantisation.

## 7 CONCLUSION

In this work, we study two surprising phenomena in large auto-regressive Transformers: (1) the strong, consistent dominance of the first token in attention maps; and (2) the presence of outlier activations in the hidden states. We propose novel solutions: (1) the softmax-1 function to remove first token dominance; and (2) the OrthoAdam optimiser which mitigates outlier activations. By doing so, we reduce first token dominance of attention maps by up to 95% and the activation kurtosis by up to 99.8%. Furthermore, our work improves our understanding of Transformers but also offer practical benefits in model quantisation, reducing the quantisation penalty by up to 99.9%.

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

# A  TRAINING DETAILS

In this section, we provide details on the training of our models.

In all experiments we use a batch size of 512 and in all experiments using Adam or OrthoAdam as the optimiser, we use a peak learning rate of $10^{-3}$. This excludes the experiments in Section 5.3 which use SGD as the optimiser, which use a peak learning rate of $0.2$. In all experiments we use a cosine learning rate schedule with linear warmup for $\{1000, 2000, 6000, 10000\}$ steps for models with $\{60M, 130M, 350M, 1.4B\}$ parameters respectively. Note that we use a reduced number of steps for the 1.4B model due to computational constraints. For the main experimental results in Tables 2 and 3, we train the models with $\{60M, 130M, 350M, 1.4B\}$ parameters for $\{160k, 320k, 960k, 600k\}$ steps respectively. For the ablation study in Section 5.3, we train GPT2 models with 130M parameters for 40k steps with 2000 warmup steps. We use a maximum sequence length of 256 tokens, which we find is sufficient to observe the anomalies of first token attention dominance and large outlier activations found in popular pretrained models such as GPT2 (Radford et al., 2019) and Llama (Touvron et al., 2023a;b; Dubey et al., 2024). The result of our training setup is that models trained for the main experimental results with $\{60M, 130M, 350M, 1.4B\}$ parameters are trained on $\{21B, 42B, 126B, 79B\}$ tokens respectively. The ablation experiments are trained on 5B tokens. We train models on 8 NVIDIA 32GB V100 GPUs using the Pytorch deep-learning framework (Paszke et al., 2019) and the HuggingFace Transformers library (Wolf et al., 2020).

## B  LONGER SEQUENCE TRAINING

| Model Size | Setup | Full | Coarse | Δ (Coarse) | Moderate | Δ (Moderate) | Fine | Δ (Fine) |
|---|---|---|---|---|---|---|---|---|
| 60M-256 | Vanilla | 31.88 | 43.53 | 11.65 | 34.87 | 2.99 | 32.15 | 0.27 |
| 60M-512 | Vanilla | 32.66 | 48.55 | 15.89 | 37.24 | 4.58 | 33.07 | 0.41 |
| 60M-1024 | Vanilla | 33.52 | 57.68 | 24.16 | 38.22 | 4.70 | 33.80 | 0.28 |
| 60M-256 | S1+OA | 31.93 | 32.46 | 0.53 | 32.32 | 0.39 | 32.00 | 0.07 |
| 60M-512 | S1+OA | 31.83 | 32.30 | 0.06 | 32.18 | 0.35 | 31.89 | 0.47 |
| 60M-1024 | S1+OA | 32.25 | 32.85 | 0.60 | 32.73 | 0.48 | 32.32 | 0.07 |
| 130M-256 | Vanilla | 22.89 | 46.49 | 23.60 | 28.31 | 5.42 | 23.07 | 0.18 |
| 130M-512 | Vanilla | 22.80 | 42.34 | 19.54 | 28.14 | 5.34 | 22.98 | 0.18 |
| 130M-1024 | Vanilla | 22.93 | 38.78 | 15.85 | 29.04 | 6.11 | 23.16 | 0.23 |
| 130M-256 | S1+OA | 22.78 | 23.21 | 0.43 | 23.10 | 0.32 | 22.83 | 0.05 |
| 130M-512 | S1+OA | 22.73 | 23.16 | 0.43 | 23.04 | 0.31 | 22.79 | 0.06 |
| 130M-1024 | S1+OA | 23.87 | 23.28 | 0.41 | 23.19 | 0.32 | 22.94 | 0.07 |

Table 6: Performance results for various model sizes and setups under longer sequence length.

In Table 6 we provide results when trained with sequence length of 512 and 1024, and compare with sequence length 256. As can be seen, our model is very robust when we increase the sequence length, showing no noticeable performance drop in perplexity be it under the general setting, or when quantized. In all cases, especially under quantization schemes, our method outperforms the vanilla one when trained with longer sequences.

## C  LARGER LLMS

We show that the first token attention and the outliers happen also in large modern LLMs such as Llama-3.1-8B. Furthermore, these issues happen regardless if the training is done in unsupervised manner (next-token prediction) or supervised manner (intruction tuning). We downloaded Llama-3.1-8B (https://huggingface.co/meta-llama/Llama-3.1-8B) and Llama3.1-8B-Instruct (https://huggingface.co/meta-llama/Llama-3.1-8B-Instruct).

In Table 8, we show that in these large models, the first token attention increases (at over $95\%$ compared to the results shown in the main paper. Furthermore, we also checked the cumulative sum of attention to the first token and found it out to be at $73.49\%$. In other words, $73.49\%$ of the entire attention in Llama-3.1-8B is in the first token. This can be interpreted that the larger the network, the more specialized the heads are, and most of the heads will simply do nothing. Attending on the first token is the mechanism the Transformer has developed to learn to do nothing. We also check the kurtosis of Llama-3.1-8B, showing that the method has a very high kurtosis for both the first token and on average.

Finally, we show that these results remain very similar if the model is finetuned in instruction data. We observe that the first token attention and first token kurtosis is virtually the same in Instruct model as in the original one, while the average kurtosis actually increases in the Instruct model. Thus, we conclude that our findings stand for modern large LLMs, regardless if they are finetuned in instruction data or not.

| Method | %1st_attention | Sum_first_token | 1st kurtosis | Average kurtosis |
|---|---|---|---|---|
| Llama-3.1-8B | 95.45 | 73.49 | 1227 | 55 |
| Llama-3.1-8B-Instruct | 95.53 | 70.13 | 1228 | 69 |

Table 7: Comparison of attention and kurtosis metrics for LLama-3.1-8B and Llama-3.1-8B-Instruct.

## D  INSTRUCTION TUNING

To complement the previous experiment, we run a new experiment, doing instruction finetuning (supervised learning) in Alpaca dataset. We compare the results of a model trained with canonical

| Method | $P_{full}$ | $P_{coarse}$ | $\Delta_{coarse}$ | $P_{mod}$ | $\Delta_{mod}$ | $P_{fine}$ | $\Delta_{fine}$ | $1^{st}_{att}(\%)$ | $1^{st}_{att}(sum)$ | $1^{st}_{kurtosis}$ | $A_{kurtosis}$ |
|---|---|---|---|---|---|---|---|---|---|---|---|
| GPT-2-130M | 18.33 | 31.03 | 12.07 | 21.09 | 2.76 | 18.49 | 0.16 | 65.57 | 41.49 | 564.75 | 69.31 |
| GPT-2-130M + S1 + OA | **18.29** | **18.31** | **0.02** | **18.31** | **0.02** | **18.29** | **0.0** | **2.4** | **0.8** | **2.97** | **2.96** |

Table 8: Results on instruction tuning using GPT-2-130M as baseline and comparing with our approach. $P_{full}$ represents perplexity without any quantization. $P_{coarse}$, $P_{mod}$ and $P_{fine}$ represent perplexity under coarse, moderate and fine quantization. Similarly, $\Delta_{coarse}$, $\Delta_{mod}$ and $\Delta_{fine}$ represent the increase in perplexity under these three quantization schemes compared to the not quantized method. $1^{st}_{att}(\%)$ represent the percentage of tokens that attend to the first one, $1^{st}_{att}(sum)$ represent the cumulative sum of the attention in the first token, $1^{st}_{kurtosis}$ represent the first token kurtosis, while $A_{kurtosis}$ represents the average kurtosis in the network. As we can observe, our method reaches best results under every setting, while reducing the attention on the first token and kurtosis.

softmax and Adam, compared to our method with softmax-1 and OrthoAdam. We present the results in GPT-2-130m models.

We show that while both models reach roughly the same perplexity, only our model keeps the same perplexity under all three quantization schemes. On the other hand, the vanilla model drops for 12.07 points in coarse quantization, 2.76 in moderate quantization, and 0.16 in fine quantization.

To evaluate that this is a direct effect of the first token attention and outliers, we also present the results in the percentage of maximum attention in the first token, the cumulative attention score of the first token, the 1st token kurtosis, and the average kurtosis. We show that in the vanilla model, the maximum attention is in 65.57% of cases in the first token, and the cumulative attention on the first token is 41.49%. In contrast, our method has maximum attention on the first token is 2.4% and the first token contributes to only 0.8% of the attention. Furthermore, while the vanilla model has kurtosis of 564.75 for the first token, and 69.31 for the average token, our method has 2.97 kurtosis for the first token, and 2.96 kurtosis for average token, very similar to the kurtosis of normal distribution (3).

In this way, we empirically show that our findings of the main paper stand also for instruction tuning training. Models trained with instruction tuning drop in accuracy under quantization schemes because of their outliers, while our method remains stable and as we showed, has the same kurtosis as normal distribution and not high attention in the first token.

# E  OPTIMISER BASIS

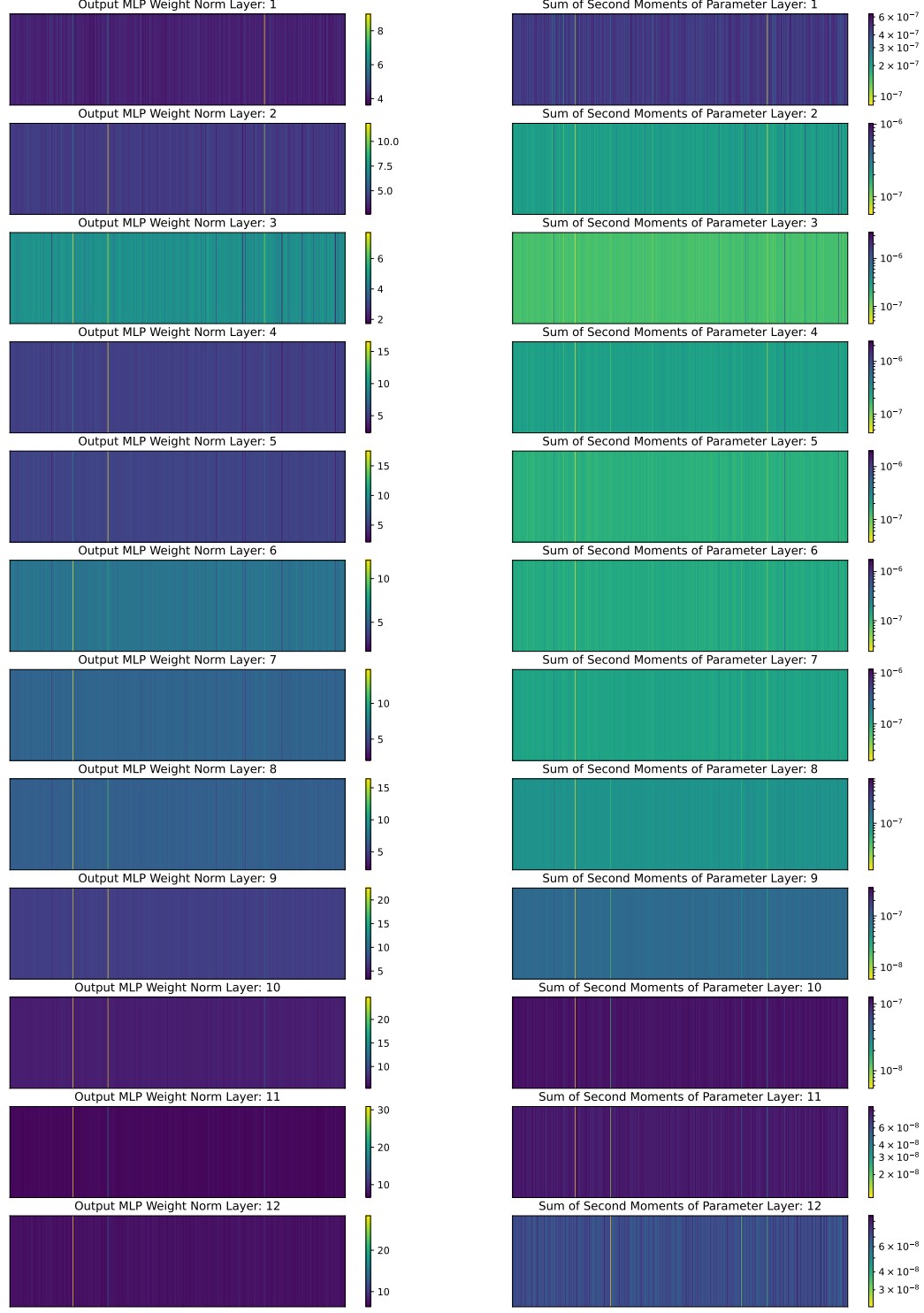

Figure 4: Left: MLP output feature weight euclidean norm for each layer in our GPT2-130M model training with canonical softmax and Adam. Right: Sum of gradient second moments for the corresponding MLP output feature weight when training with canonical softmax and Adam. Training with softmax and Adam leads to small outlier gradient second moment moving averages with lead to disproportionately large gradient steps for the feature dimensions containing these small outliers. This in turn leads to large outliers in the model weights.

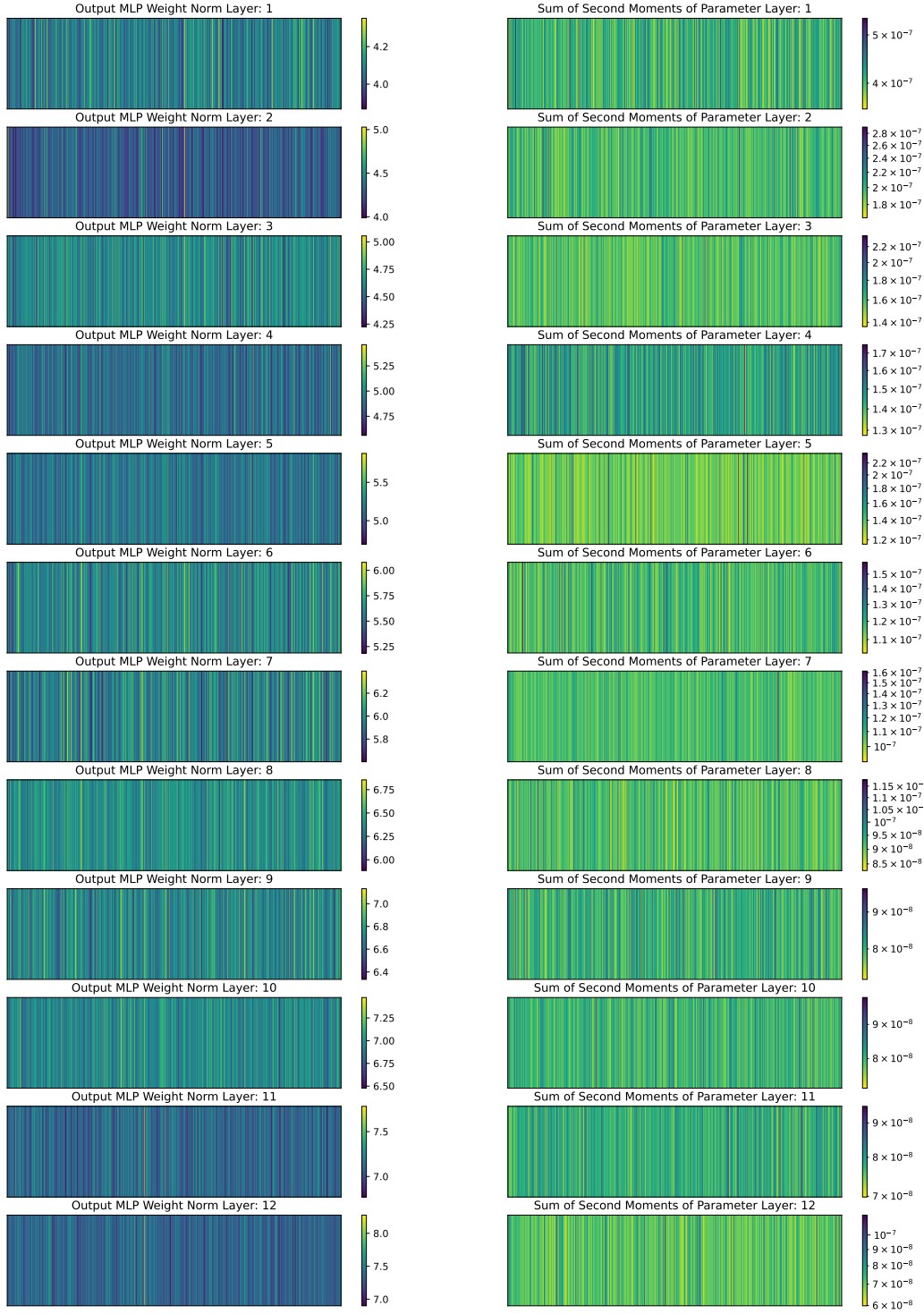

Figure 5: Left: MLP output feature weight euclidean norm for each layer in our GPT2-130M model training with softmax-1 and OrthoAdam. Right: Sum of gradient second moments for the corresponding MLP output feature weight when training with softmax-1 and OrthoAdam. Training with softmax-1 and OrthoAdam leads to gradient second moment moving averages to be considerably more uniform than training with softmax and Adam. This prevents large gradient steps from being taken removing outlier weights in the model.

Our observation of outlier activations in the hidden states of language models leads us to the conclusion that some aspect of the training process is "basis-dependent". "Basis-independent" functions are those which are equivariant under orthogonal transformations. A function $f(x)$ is basis-independent if $f(Qx) = Qf(x)$ for any orthogonal matrix $Q$. We remove biases from our linear layers and introduce a single-scale version of RMSNorm to remove "basis-dependent" effects from the model itself. It is straightforward to show that linear layers with biases (*i.e.*, affine transformations) and multi-scale RMSNorm (as is standard) are not basis-independent (*i.e.*, they are basis-dependent). SGD and SGD with momentum are basis-independent: We can show that the standard SGD update rule is basis-independent.

Given the standard SGD update rule $\theta_{t+1} = \theta_t - \eta\nabla L(\theta_t)$ transforms the parameters to $\rightarrow \hat{\theta}_t = \mathbf{Q}\theta_t$ where $\theta_t$ is an orthogonal matrix:

$$\hat{\theta}_{t+1} = \hat{\theta}_t - \eta\nabla\hat{L}(\hat{\theta}_t) \tag{E.1}$$

By chain rule:

$$\nabla\hat{L}(\hat{\theta}_t) = \nabla L(\theta_t) \cdot \frac{\partial\theta_t}{\partial\hat{\theta}_t} = \nabla L(\theta_t) \cdot \mathbf{Q}^T = \mathbf{Q}\nabla L(\theta_t),$$
$$\hat{\theta}_{t+1} = \hat{\theta}_t - \eta\mathbf{Q}\nabla L(\theta_t) = \mathbf{Q}(\theta_t - \eta\nabla L(\theta_t)) = \mathbf{Q}\theta_{t+1}. \tag{E.2}$$

SGD with momentum follows similarly.

However, Adam and RMSProp are not basis-independent because of tracking the second-order moments. Let $v_t$ be the second-order moment of the gradients at step $t$, then

$$v_{t+1} = \beta_2 v_t + (1 - \beta_2)(\nabla L(\theta_t) \odot \nabla L(\theta_t)) \tag{E.3}$$

$$\begin{aligned}
\hat{v}_{t+1} &= \beta_2\hat{v}_t + (1 - \beta_2)(\nabla\hat{L}(\hat{\theta}_t) \odot \nabla\hat{L}(\hat{\theta}_t)) \\
&= \beta_2\hat{v}_t + (1 - \beta_2)(\mathbf{Q}\nabla L(\theta_t) \odot \mathbf{Q}\nabla L(\theta_t)) \\
&\neq \mathbf{Q}v_{t+1}
\end{aligned} \tag{E.4}$$

The above shows that SGD and SGD with momentum provide a basis-independent update rule which is proportional to the gradient, on the other hand, Adam and RMSProp are not basis-independent and provide updates that depend on the element-wise root of the second-order moment of the gradient. We believe that basis-dependent functions (whether in the model or the optimizer) are the cause of the outlier activations we observe in the hidden states of language models. The second-order moment tracking in Adam and RMSProp allows disproportionately large gradient updates in certain weights of the model especially in the early steps of training where the moment moving averages are not well-calibrated (a result of adaptive per parameter learning rate scaling). Therefore we expect features for which the second-order moment is small to have disproportionately large weights in the model. Disproportionately large weights in particular dimensions of the model cause the outlier activations we observe. Using OrthoAdam, the moving average moments of the gradients are computed in a unique random orthogonal basis for each parameter, which prevents small values of the second-order moment from causing disproportionately large updates in the model parameters (outlier gradients large or small in the model basis are transformed to be likely of similar magnitude in the orthogonal basis).

The outlier weights in the model are generally contained in the output linear layer of each MLP block. To verify our intuition above, we have computed the norm of the output weights in output linear layer of each MLP block in a GPT2-130M model trained with/without softmax-1 and OrthoAdam. We additionally plot the sum of the second-order moments of the gradients of the output weights in each MLP block. We observe that in *all* cases of large outlier weights in the model, the sum of the second-order moments of the gradients of the output weights in the corresponding MLP block are *small* outliers. This is consistent with our intuition that the *small* outlier second-order moments of the gradients are causing the *large* outlier weights in the model. We show these plots in Figure 6 and Figure 5 respectively.

# F   TRANSFORMING THE OUTPUT INTO ORTHOGONAL BASIS

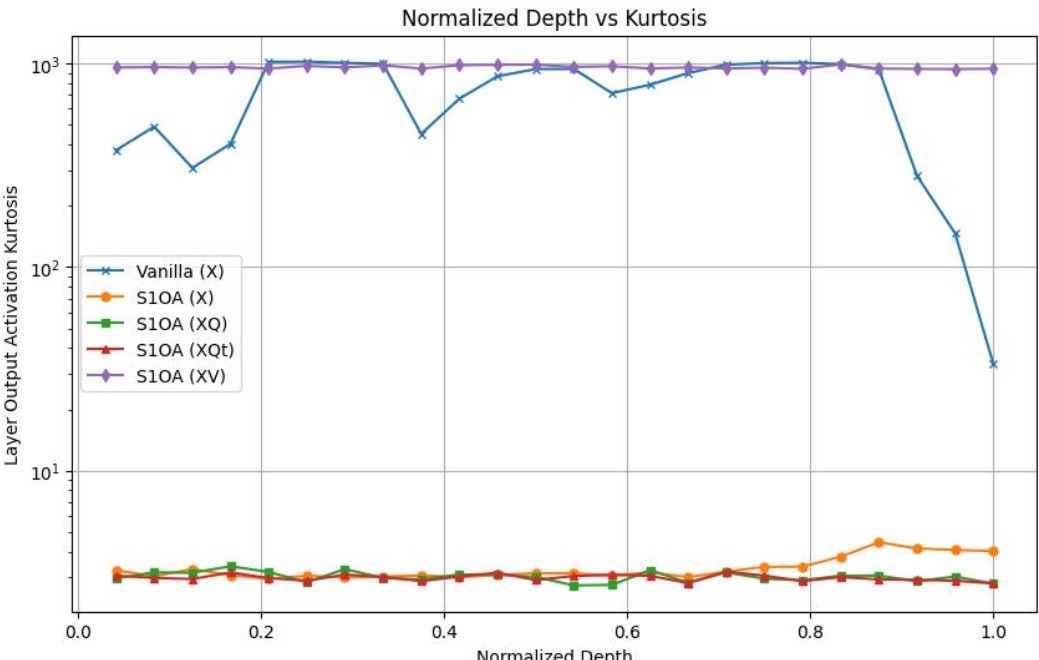

Figure 6: Transforming the output into orthogonal basis

We investigated transforming the output of each layer (*i.e.*, the added activations to the hidden states for said layer) into the orthogonal basis used by that output layer in OrthoAdam. We use our GPT2-350M model trained with Softmax-1 and OrthoAdam. Given layer $i$ has output $\mathbf{X}_i \in \mathbb{R}^{L \times D}$ and the orthogonal basis for the output layer in OrthoAdam is $\mathbf{Q}_i \in \mathcal{O}^D$, where $L$ is the sequence length, $D$ is the hidden dimension and $\mathcal{O}^D$ is the set of $D \times D$ orthogonal matrices. We plot the activation kurtosis of $\mathbf{X}_i$, $\mathbf{X}_i\mathbf{Q}_i$, $\mathbf{X}_i\mathbf{Q}_i^T$ and $\mathbf{X}_i\mathbf{V}_i$ where $\mathbf{V}_i$ is the right singular vectors of $\mathbf{X}_i$. Using $\mathbf{V}_i$ as a transformed basis gives a baseline for how large one could increase the kurtosis of the activations by transforming them into an orthogonal basis. Additionally we give the activation kurtosis for our GPT2-350M vanilla model.

## G    KURTOSIS GROWS WITH THE NUMBER OF DIMENSIONS IN TRANSFORMERS

In this section, we use some observations from the hidden states of transformer models to illustrate how the kurtosis of the hidden states grows with the number of dimensions in the hidden states. This is something we observe empirically in the hidden states of transformer models and is a key motivation for our work. Table 2 shows the kurtosis of the hidden states of transformer models trained *without softmax-1 or OrthoAdam* grows as the model size increases, as does the maximum activation value in the hidden states.

To make this mathematically rigorous, we shall consider a simple example, in which we shall approximate the hidden states of a transformer model at a single token position as a $D$-dimensional vector comprising of the sum of a scaled one-hot vector and a standard normal vector.

Consider a $D$-dimensional vector $\mathbf{x}$ which is the sum of two $D$-dimensional vectors $\alpha \mathbf{e}_i$ and $\mathbf{z}$, where $\mathbf{e}_i$ is the $i$th unit vector in the standard basis, $\mathbf{x} \in \mathbb{R}^D$, $\alpha \in \mathbb{R}$ and $\mathbf{z} \sim \mathcal{N}(\mathbf{0}, \mathbf{I}_D)$. Therefore the elements of $\mathbf{x}$ are given by:

$$x_j = \alpha \delta_{ij} + z_j \quad \text{for } j = 1, 2, \dots, D$$

where $\delta_{ij}$ is the Kronecker delta function. The mean is given by:

$$\begin{aligned}
\mu = \mathbb{E}\left[x_j\right] &= \mathbb{E}\left[\alpha \delta_{ij} + z_j\right] = \alpha \mathbb{E}\left[\delta_{ij}\right] + \mathbb{E}\left[z_j\right] \\
&= \frac{\alpha}{D} + 0 = \frac{\alpha}{D} \quad \text{as } \mathbb{E}\left[z_j\right] = 0 \text{ by definition of the standard normal distribution}
\end{aligned}$$

$$\boxed{\mu = \frac{\alpha}{D}} \tag{G.1}$$

The variance is given by:

$$\begin{aligned}
\sigma^2 = \text{Var}\left[x_j\right] &= \text{Var}\left[\alpha \delta_{ij} + z_j\right] \\
&= \alpha^2 \text{Var}\left[\delta_{ij}\right] + \text{Var}\left[z_j\right] \quad \text{as } \alpha \delta_{ij} \text{ and } z_j \text{ are independent in our model} \\
&= \alpha^2 \text{Var}\left[\delta_{ij}\right] + 1 \quad \text{as } \text{Var}\left[z_j\right] = 1 \text{ by definition of the standard normal distribution}
\end{aligned}$$

$$\text{Var}\left[\delta_{ij}\right] = \mathbb{E}\left[\delta_{ij}^2\right] - \left(\mathbb{E}\left[\delta_{ij}\right]\right)^2 = \frac{1}{D}\left(1 - \frac{1}{D}\right)$$

Therefore:

$$\boxed{\sigma^2 = \frac{\alpha^2}{D}\left(1 - \frac{1}{D}\right) + 1} \tag{G.2}$$

The kurtosis of the elements of $\mathbf{x}$ is given by:

$$\text{Kurt}[x_j] = \mathbb{E}\left[\left(\frac{x_j - \mu}{\sigma}\right)^4\right] = \frac{\mathbb{E}\left[(x_j - \mu)^4\right]}{\sigma^4}$$

When $j \neq i$:

$$\begin{aligned}
\mathbb{E}\left[(x_j - \mu)^4\right] &= \mathbb{E}\left[\left(z_j - \frac{\alpha}{D}\right)^4\right] \\
&= \mathbb{E}\left[z_j^4 - 4z_j^3 \frac{\alpha}{D} + 6z_j^2 \left(\frac{\alpha}{D}\right)^2 - 4z_j \left(\frac{\alpha}{D}\right)^3 + \left(\frac{\alpha}{D}\right)^4\right] \\
&= \mathbb{E}\left[z_j^4\right] - 4\mathbb{E}\left[z_j^3\right]\frac{\alpha}{D} + 6\mathbb{E}\left[z_j^2\right]\left(\frac{\alpha}{D}\right)^2 - 4\mathbb{E}\left[z_j\right]\left(\frac{\alpha}{D}\right)^3 + \left(\frac{\alpha}{D}\right)^4
\end{aligned}$$

As $\mathbb{E}\left[z_j^3\right] = 0$ and $\mathbb{E}\left[z_j^4\right] = 3$:

$$= 3 + 6\left(\frac{\alpha}{D}\right)^2 + \left(\frac{\alpha}{D}\right)^4$$

When $j = i$:

$$
\begin{aligned}
\mathbb{E}\left[(x_j - \mu)^4\right] &= \mathbb{E}\left[\left(\alpha + z_j - \frac{\alpha}{D}\right)^4\right] \\
&= \mathbb{E}\left[\left(\alpha\left(1 - \frac{1}{D}\right) + z_j\right)^4\right] \\
&= \mathbb{E}\left[\left(\alpha\left(1 - \frac{1}{D}\right)\right)^4 + 4\left(\alpha\left(1 - \frac{1}{D}\right)\right)^3 z_j\right. \\
&\quad \left. + 6\left(\alpha\left(1 - \frac{1}{D}\right)\right)^2 z_j^2 + 4\left(\alpha\left(1 - \frac{1}{D}\right)\right) z_j^3 + z_j^4\right] \\
&= \left(\alpha\left(1 - \frac{1}{D}\right)\right)^4 + 6\left(\alpha\left(1 - \frac{1}{D}\right)\right)^2 + 3
\end{aligned}
$$

Therefore, the *overall fourth moment* of the elements of $\mathbf{x}$ is given by:

$$
\begin{aligned}
\mathbb{E}\left[(x_j - \mu)^4\right] &= \frac{1}{D}\left(\left(\alpha\left(1 - \frac{1}{D}\right)\right)^4 + 6\left(\alpha\left(1 - \frac{1}{D}\right)\right)^2 + 3\right) \\
&\quad + \frac{D-1}{D}\left(3 + 6\left(\frac{\alpha}{D}\right)^2 + \left(\frac{\alpha}{D}\right)^4\right)
\end{aligned}
$$

And the kurtosis of the elements of $\mathbf{x}$ is given by:

$$
\mathrm{Kurt}\left[x_j\right] = \frac{\frac{1}{D}\left(\left(\alpha\left(1 - \frac{1}{D}\right)\right)^4 + 6\left(\alpha\left(1 - \frac{1}{D}\right)\right)^2 + 3\right) + \frac{D-1}{D}\left(3 + 6\left(\frac{\alpha}{D}\right)^2 + \left(\frac{\alpha}{D}\right)^4\right)}{\left(\frac{\alpha^2}{D}\left(1 - \frac{1}{D}\right) + 1\right)^2}
$$

$$
\boxed{\mathrm{Kurt}\left[x_j\right] = \frac{3 + \frac{\alpha^4}{D} + \frac{6\alpha^2}{D} - \frac{4\alpha^4}{D^2} - \frac{6\alpha^2}{D^2} + \frac{6\alpha^4}{D^3} - \frac{3\alpha^4}{D^4}}{1 + \frac{2\alpha^2}{D} - \frac{2\alpha^2}{D^2} + \frac{\alpha^4}{D^2} - \frac{2\alpha^4}{D^3} + \frac{\alpha^4}{D^4}}}
\tag{G.3}
$$

At this point, we can see that Kurtosis is a function of $\alpha$ and $D$, however if we consider the limit as $D \to \infty$, we can see that $\mathrm{Kurt}[x_j] \to 3$, *i.e.*, the kurtosis of a Gaussian distribution. However, this neglects the importance of the scaling factor $\alpha$ which we know empirically is larger than the dimensionality of the hidden states. The table below summarises the maximum activation values (analogous to $\alpha$) and the dimension of the hidden states for the models we trained. Given this

| Model | #Parameters | Model Size ($D$) | Max Activation ($\alpha$) |
|-------|-------------|------------------|---------------------------|
| GPT2  | 60M         | 512              | 1856                      |
|       | 130M        | 768              | 7018                      |
|       | 350M        | 1024             | 40196                     |
|       | 1.4B        | 2048             | 56798                     |
| Llama | 130M        | 768              | 4623                      |

Table 9: Model sizes and maximum activation values for the models used in our experiments.

empirical information, we make the *conservative* assumption that $\alpha = D$. Under this assumption

which is supported by our empirical observations, Equation (G.3) simplifies to:

$$\text{Kurt}\left[x_j\right] = \frac{3 + \frac{D^4}{D} + \frac{6D^2}{D} - \frac{4D^4}{D^2} - \frac{6D^2}{D^2} + \frac{6D^4}{D^3} - \frac{3D^4}{D^4}}{1 + \frac{2D^2}{D} - \frac{2D^2}{D^2} + \frac{D^4}{D^2} - \frac{2D^4}{D^3} + \frac{D^4}{D^4}}$$

$$= \frac{3 + D^3 + 6D - 4D^2 - 6 + 6D - 3}{1 + 2D - 2 + D^2 - 2D + 1}$$

$$= \frac{D^3 - 4D^2 + 12D - 6}{D^2}$$

$$\boxed{\text{Kurt}\left[x_j\right] = D - 4 + \frac{12}{D} - \frac{6}{D^2} = O(D)} \tag{G.4}$$

Using our conservative assumption that $\frac{\alpha}{D} = 1$, we can see that the kurtosis of the hidden states grows linearly with the dimensionality of the hidden states when $D$ is in the region of $10^3 - 10^5$ as is the case for transformer models.

This simple example serves as a mathematical illustration of the empirical observations we make in the hidden states of transformer models. We have shown that the kurtosis of the hidden states is expected to grow linearly with the dimensionality of the hidden states, and so the issue of outlier activations is expected to grow as the hidden states of transformer models grow in size.

# H Orthogonal transformations and reduction in $\ell_\infty$–norm and Kurtosis

From our simple model in Appendix G we have a simplified model of Transformer hidden states, $\mathbf{x} \in \mathbb{R}^D$, where the first element is $\alpha$ and the rest are standard normal random variables.

$$\mathbf{x} = \alpha\mathbf{e}_i + \mathbf{z} \quad \text{where } z_j \sim \mathcal{N}(0,1)$$

From this model, we can compute the expected $\ell_2$–norm:

$$\mathbb{E}\left[\|\mathbf{x}\|_2^2\right] = \sum_{j=1}^{D} x_j^2 = \alpha^2 + \sum_{j=1}^{D} z_j^2 = \alpha^2 + D\text{Var}\left[z_j\right] = \alpha^2 + D \tag{H.1}$$

Using the triangle inequality, we can compute a range for the $\ell_\infty$–norm:

$$\mathbb{E}\left[\|\mathbf{x}\|_\infty\right] = \mathbb{E}\left[\max_{1 \le j \le D}\left(|\alpha + z_i|, \max_{j \ne i} |z_j|\right)\right]$$

Given $\alpha \gg 1$, we can drop the terms for $j \ne i$ and compute the expected $\ell_\infty$–norm using the $i^{\text{th}}$ element:

$$\mathbb{E}\left[\|\mathbf{x}\|_\infty\right] = \mathbb{E}\left[|\alpha + z_i|\right]$$
$$|\alpha + z_i| \le |\alpha| + |z_i|$$

Using folded normal distribution properties, $\mathbb{E}\left[|z_i|\right] = \sqrt{\frac{2}{\pi}} \ll \alpha$, therefore:

$$\mathbb{E}\left[\|\mathbf{x}\|_\infty\right] \approx \alpha$$

Given that $\alpha \gg 1$, we can safely assume that $\|\mathbf{x}\|_\infty^2 \approx \alpha^2$. Therefore:

$$\mathbb{E}\left[\frac{\|\mathbf{x}\|_\infty}{\|\mathbf{x}\|_2}\right] \approx \frac{\alpha}{\sqrt{D + \alpha^2}}$$

Note from Table 9 that the maximum activation value, $\alpha$, is generally much larger than the model size, $D$.

$$\mathbb{E}\left[\frac{\|\mathbf{x}\|_\infty}{\|\mathbf{x}\|_2}\right] \approx 1 \tag{H.2}$$

We find this empirically to be the case in the middle layers of the Transformer models we study (see plots in Appendix I.3).

The $\infty$-norm of $\mathbf{x}$ can be thought of as a proxy for the extent of outliers in a vector. If $\frac{\|\mathbf{x}\|_2}{\|\mathbf{x}\|_\infty} \approx 1$, then a vector has at least one large outlier and consequently a high kurtosis.

We will now show that applying an orthogonal transformation to a vector can reduce the $\ell_\infty$-norm constrained to a fixed $\ell_2$-norm. Using the same definition of $\mathbf{x}$ as above, let $\mathbf{Q} \in \mathbb{R}^{D \times D}$ be an orthogonal matrix and let $\mathbf{y} = \mathbf{Q}\mathbf{x}$.

$$\|\mathbf{y}\|_2^2 = \mathbf{y}^T\mathbf{y} = \mathbf{x}^T\mathbf{Q}^T\mathbf{Q}\mathbf{x} = \mathbf{x}^T\mathbf{x} = \|\mathbf{x}\|_2^2$$
$$\mathbb{E}[\|\mathbf{y}\|_2^2] = \mathbb{E}[\|\mathbf{x}\|_2^2] = \alpha^2 + D \tag{H.3}$$

This standard proof shows that applying an orthogonal transformation to a vector does not change the $\ell_2$–norm of the vector. It can however lead to a dramatic reduction in the $\ell_\infty$–norm of the vector. We will now show that for a vector, $\mathbf{y} \in \mathbb{R}^D$, constrained to have a fixed $\ell_2$–norm, $\sqrt{\alpha^2 + D}$, the $\ell_\infty$–norm of a vector can be reduced significantly by applying an orthogonal transformation such that $y_j = \frac{\alpha}{\sqrt{D}} + z', \forall j \in [1, D]$, where $z' \sim \mathcal{N}(0,1)$.

$$\mathbf{y} = \mathbf{Q}\mathbf{x} = \mathbf{Q}\left(\alpha\mathbf{e}_i + \mathbf{z}\right) = \alpha\mathbf{Q}\mathbf{e}_i + \mathbf{Q}\mathbf{z}$$

Select $\mathbf{Q}$ such that $\mathbf{Q}\mathbf{e}_i = \left( \frac{1}{\sqrt{D}}, \frac{1}{\sqrt{D}}, \ldots, \frac{1}{\sqrt{D}} \right)$, given $\mathbf{Q}$ is orthogonal, $\mathbf{Q}\mathbf{z} = \mathbf{z}' \sim \mathcal{N}(0, \mathbf{I}_D)$.

$$\mathbb{E}\left[ \|\mathbf{y}\|_\infty \right] \approx \frac{\alpha}{\sqrt{D}} + \sqrt{2 \ln D}, \quad \text{using extreme value theory (Cramér, 1946)}$$

$$= \frac{\alpha + \sqrt{2D \ln D}}{\sqrt{D}}$$

The expected ratio of $\ell_\infty$–norm to $\ell_2$–norm is:

$$\mathbb{E}\left[ \frac{\|\mathbf{y}\|_\infty^2}{\|\mathbf{y}\|_2^2} \right] = \frac{\mathbb{E}\left[ \|\mathbf{y}\|_\infty^2 \right]}{\mathbb{E}\left[ \|\mathbf{y}\|_2^2 \right]} = \frac{\left( \alpha + \sqrt{2D \ln D} \right)^2}{D \left( \alpha^2 + D \right)}$$

Using the same *conservative* assumption as in Appendix G that $\alpha = D$, Table 9 shows empirically $\alpha > D$:

$$\mathbb{E}\left[ \frac{\|\mathbf{y}\|_\infty^2}{\|\mathbf{y}\|_2^2} \right] = \frac{D^2 + 2D \ln D + 2D\sqrt{2D \ln D}}{D^3 + D^2} = \frac{D + 2\sqrt{2D \ln D} + 2 \ln D}{D^2 + 1}$$

As $D$ grows, the last term of the numerator and the $1$ in the denominator become negligible:

$$\mathbb{E}\left[ \frac{\|\mathbf{y}\|_\infty^2}{\|\mathbf{y}\|_2^2} \right] \approx \frac{1}{D} + \frac{D + 2\sqrt{2 \ln D}}{D^{\frac{3}{2}}} = O\left( \frac{1}{D} \right)$$

Therefore, under an orthogonal transformation, the $\ell_\infty$–norm to $\ell_2$–norm ratio can be reduced significantly. It is trivial to show that $\text{Kurt}[y_j] = 3$ and we see many of our experiments which use OrthoAdam and softmax-1 exhibit this behaviour (see plots in Appendix I.2).

$$\boxed{\begin{array}{c} \mathbf{x} = \alpha \mathbf{e}_i + \mathbf{z}, \quad \mathbb{E}\left[ \dfrac{\|\mathbf{x}\|_\infty^2}{\|\mathbf{x}\|_2^2} \right] \approx 1 \quad \to \quad \mathbf{y} = \mathbf{Q}\mathbf{x}, \quad \mathbb{E}\left[ \dfrac{\|\mathbf{y}\|_\infty^2}{\|\mathbf{y}\|_2^2} \right] \approx \dfrac{1}{D} \\[3mm] \text{Kurt}\left[ x_j \right] = D - 4 + \dfrac{12}{D} - \dfrac{6}{D^2} = O(D) \quad \to \quad \text{Kurt}\left[ y_j \right] = 3 \end{array}}$$

The exact form of $\mathbf{Q}$ can be computed numerically or constructed using appropriately normalised Hadamard matrices (Sylvester, 1867).

# I  Layer Progression of First Token Attention Dominance, Kurtosis, $\ell_\infty$-Norm to $\ell_2$-Norm Ratio and Maximum Absolute Activation

For brevity, we give metrics for the first token attention dominance, hidden state kurtosis and absolute maximum activation *averaged over all layers* in Table 2 which gives the results of the main experiments in our work.

However, the layer-wise progression of these metrics is also of interest and can provide insights into the behaviour of the model. Additionally, we provide the same metrics for popular pretrained GPT2 and Llama models to show the similarity to our models *trained without softmax-1 and OrthoAdam*.

Finally, to establish a relationship between activation kurtosis and the $\ell_\infty$-norm to $\ell_2$-norm ratio, we calculate the Pearson's correlation coefficients between per-layer kurtosis and per-layer $\ell_\infty$-norm to $\ell_2$-norm ratio for all models in our main experimental results from Table 2.

All metrics are computed on the same validation set of the C4 dataset (Raffel et al., 2020) as in the main paper (Section 5).

## I.1  First Token Attention Dominance

We begin by examining the progression of first token attention dominance across layers. We calculate the percentage of (head, query) pairs where the query token attends most to the first (key) token. Given different models have a different number of layers, we normalise the layer index to the range $[0, 1]$ for each model.

We find a general trend across our trained models which use the canonical softmax function where the first token attention dominance begins low in the initial layers where models do initial processing of all input tokens. The dominance rises to a peak in the middle layers where heads specialise to specific sub-tasks and so the first token is attended to as a default "no-op" (Bondarenko et al., 2023; Clark et al., 2019). Finally, the dominance decreases in the final layers where the model "detokenises" the features back into token space.

### I.1.1  GPT2-60M

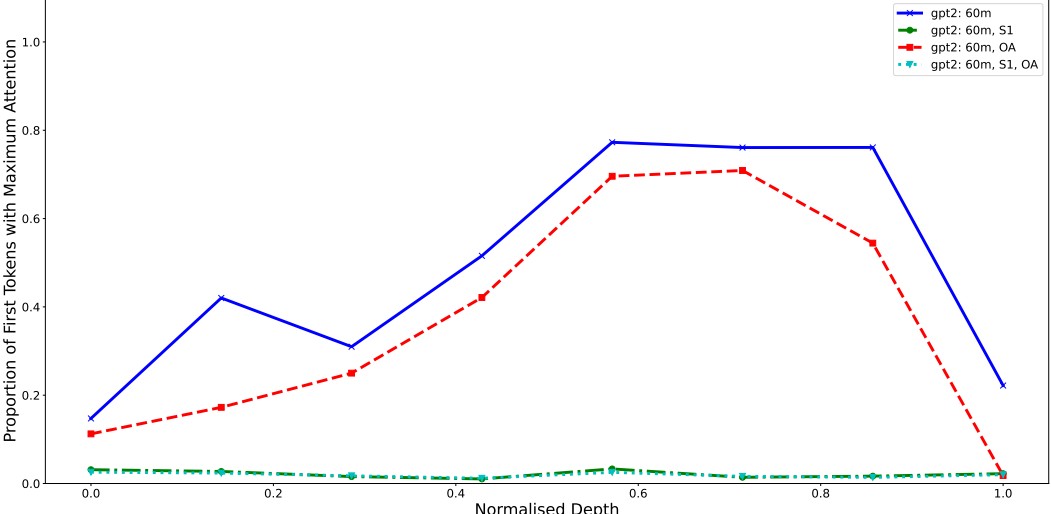

Figure 7: Layer-wise progression of first token attention dominance for GPT2-60M. The x-axis is normalised to the range $[0, 1]$. S1/OA denote models trained with softmax-1 and/or OrthoAdam.

### I.1.2 GPT2-130M

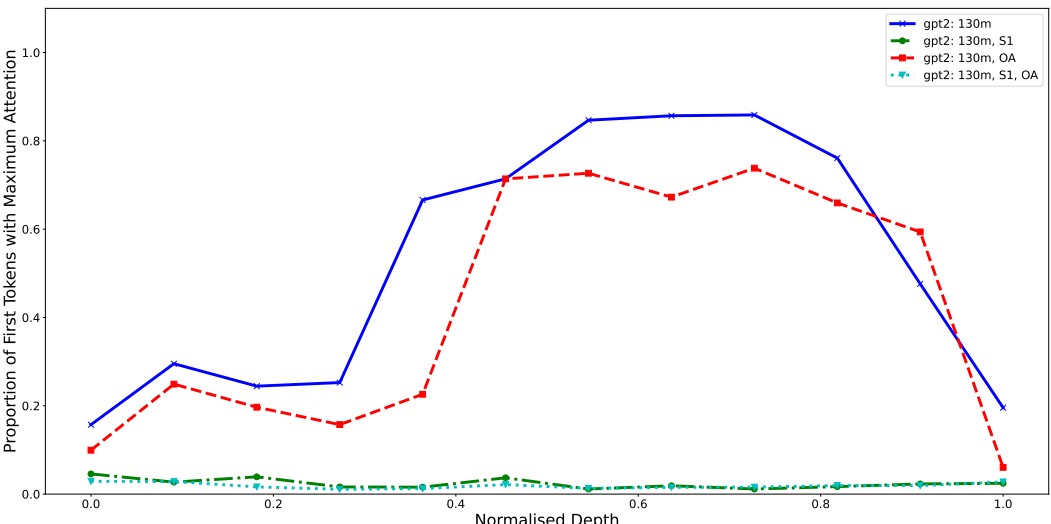

Figure 8: Layer-wise progression of first token attention dominance for GPT2-130M. The x-axis is normalised to the range $[0, 1]$. S1/OA denote models trained with softmax-1 and/or OrthoAdam.

### I.1.3 GPT2-350M AND GPT2-1.4B

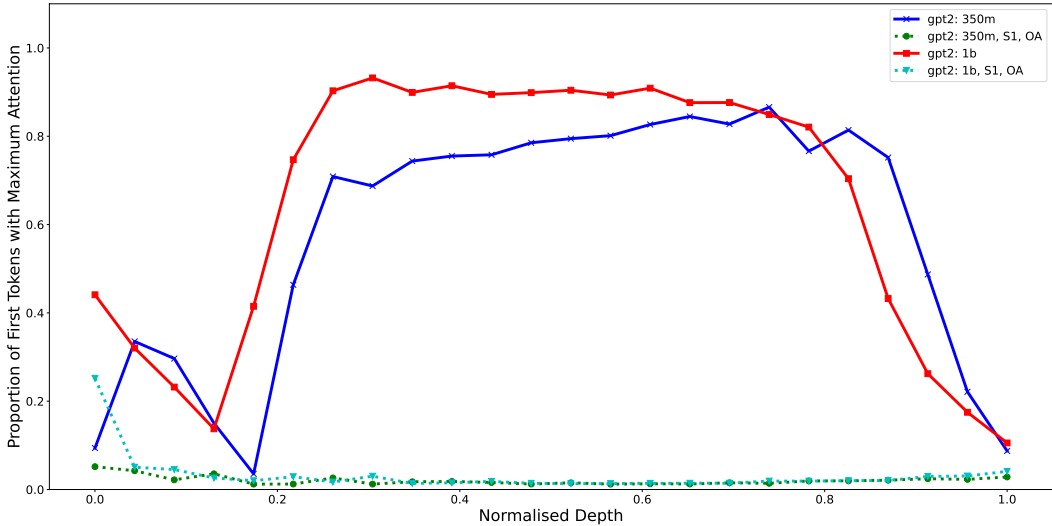

Figure 9: Layer-wise progression of first token attention dominance for GPT2-350M and GPT2-1.4B. The x-axis is normalised to the range $[0, 1]$. S1/OA denote models trained with softmax-1 and/or OrthoAdam.

### I.1.4 LLAMA-130M

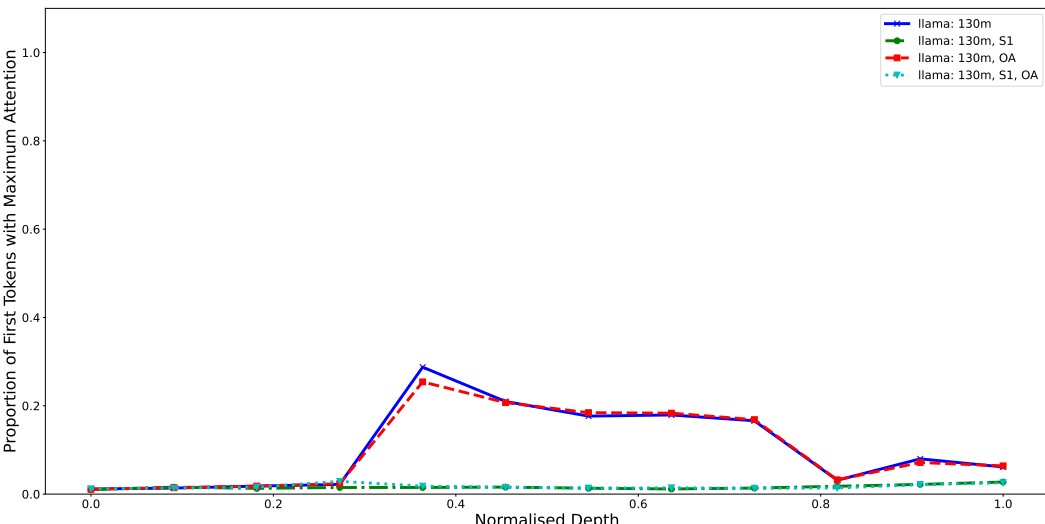

Figure 10: Layer-wise progression of first token attention dominance for Llama-130M. The x-axis is normalised to the range $[0, 1]$. S1/OA denote models trained with softmax-1 and/or OrthoAdam.

### I.1.5 POPULAR PRETRAINED MODELS—GPT2 AND LLAMA

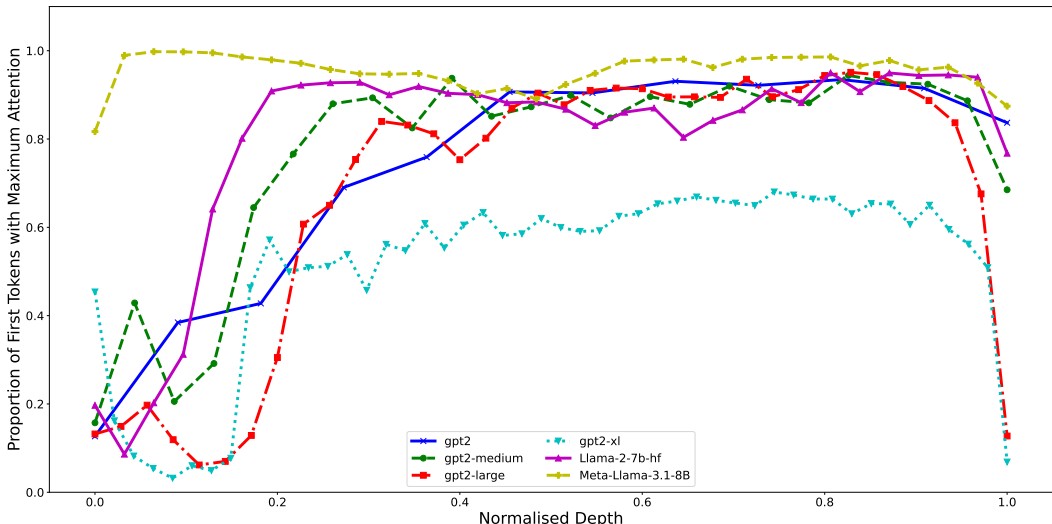

Figure 11: Layer-wise progression of first token attention dominance for popular pretrained GPT2 and Llama models. The x-axis is normalised to the range $[0, 1]$.

### I.2 ACTIVATION KURTOSIS

Next, we examine the progression of activation kurtosis across layers. As observable in Table 2, the kurtosis of the first hidden state is significantly higher than the other hidden states and so we plot the kurtosis of the first hidden state only for brevity.

We observe in the plots below that models trained without OrthoAdam exhibit a general trend of increasing kurtosis as the hidden states progress through the layers. Demonstrating that multiple layers of the model contribute to the emergence of large activation values. Models trained with

OrthoAdam *but not softmax-1* exhibit a similar trend, but with lower kurtosis values initially. Finally, models trained with both OrthoAdam and softmax-1 exhibit a consistent small kurtosis across layers—around the value of 3 which is the kurtosis of a Gaussian distribution. Interestingly, GPT2-60M and GPT2-130M show small rises in the final layers—the cause of this is left for future work.

We find that some models show a reduction in kurtosis in the final layers, we again attribute this to the "detokenisation" of the features back into token space.

### I.2.1  GPT2-60M

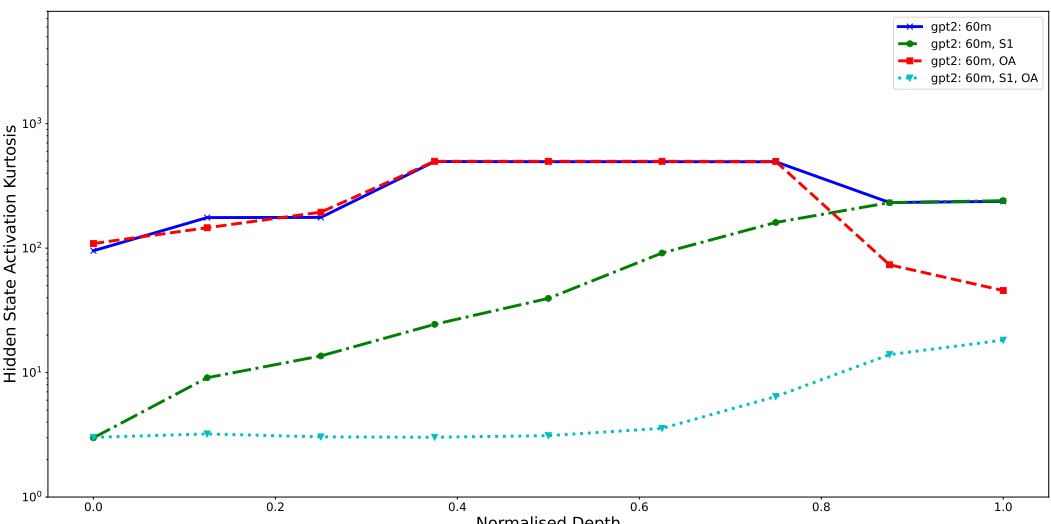

Figure 12: Layer-wise progression of activation kurtosis of the first token position for GPT2-60M. The x-axis is normalised to the range $[0, 1]$. S1/OA denote models trained with softmax-1 and/or OrthoAdam.

### I.2.2  GPT2-130M

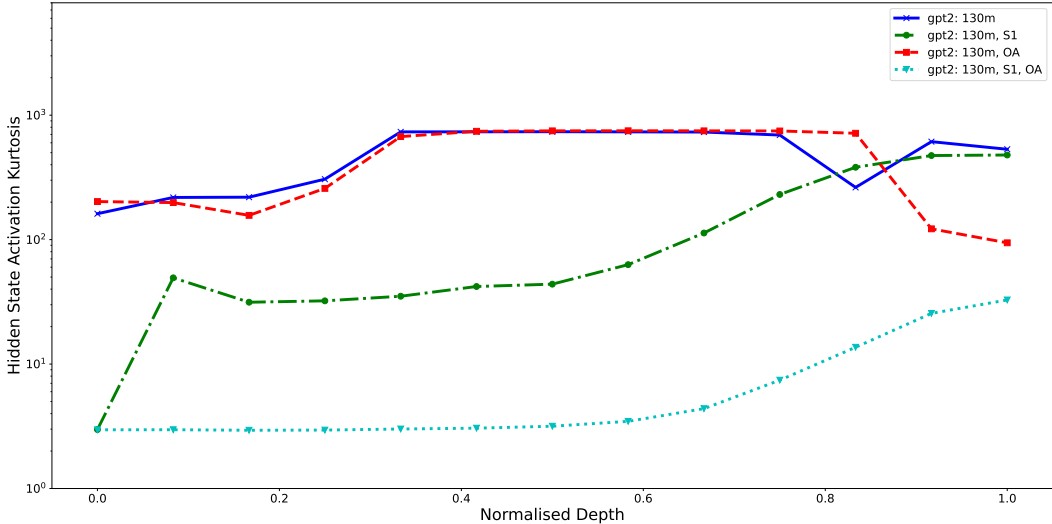

Figure 13: Layer-wise progression of activation kurtosis of the first token position for GPT2-130M. The x-axis is normalised to the range $[0, 1]$. S1/OA denote models trained with softmax-1 and/or OrthoAdam.

### I.2.3  GPT2-350M AND GPT2-1.4B

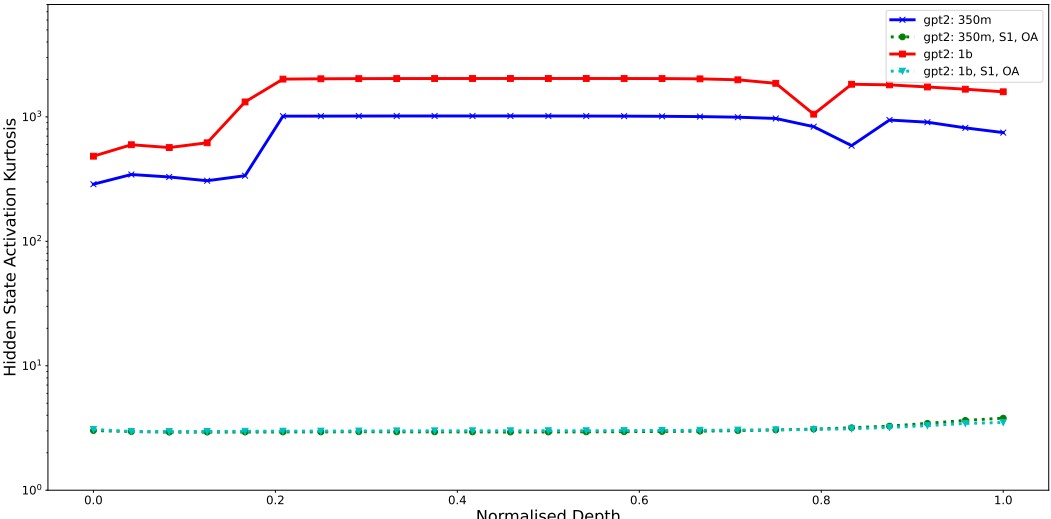

Figure 14: Layer-wise progression of activation kurtosis of the first token position for GPT2-350M and GPT2-1.4B. The x-axis is normalised to the range $[0, 1]$. S1/OA denote models trained with softmax-1 and/or OrthoAdam.

### I.2.4  LLAMA-130M

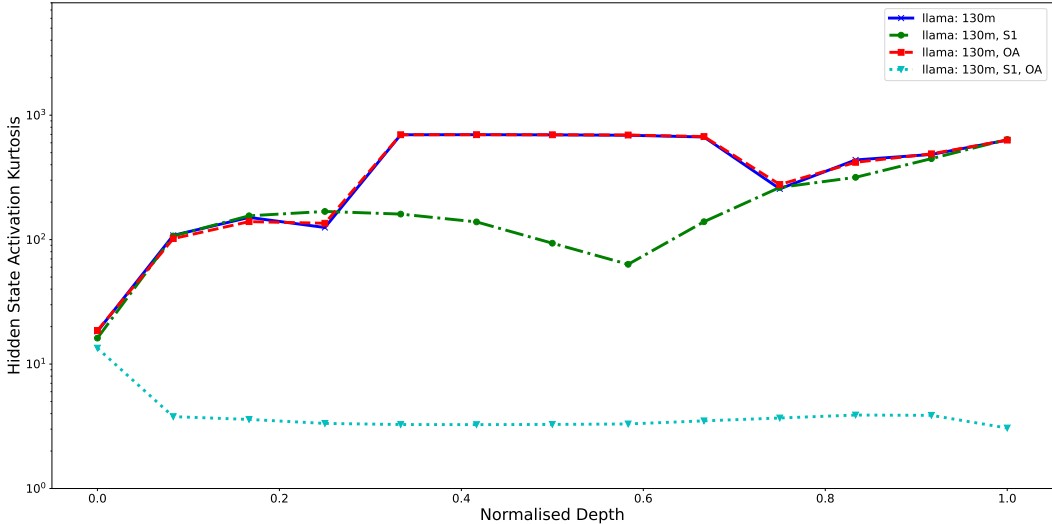

Figure 15: Layer-wise progression of activation kurtosis of the first token position for Llama-130M. The x-axis is normalised to the range $[0, 1]$. S1/OA denote models trained with softmax-1 and/or OrthoAdam.

### I.2.5 POPULAR PRETRAINED MODELS—GPT2 AND LLAMA

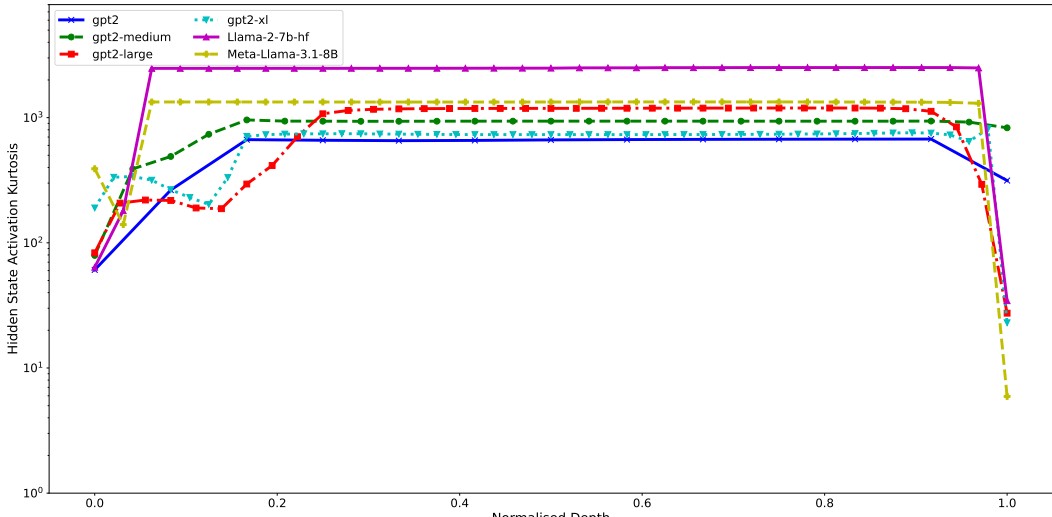

Figure 16: Layer-wise progression of activation kurtosis of the first token position for popular pretrained GPT2 and Llama models. The x-axis is normalised to the range $[0, 1]$.

### I.3 $\ell_\infty$-NORM TO $\ell_2$-NORM RATIO

The plots below show the progression of the $\ell_\infty$-norm to $\ell_2$-norm ratio across layers. We observe that models trained without OrthoAdam exhibit a general trend of increasing ratio as the hidden states progress through the layers. Once again as this ratio is maximal in the first hidden state, we plot the ratio of the first hidden state only for brevity (as done for kurtosis).

The trends are similar to the kurtosis plots and so the same commentary applies.

### I.3.1 GPT2-60M

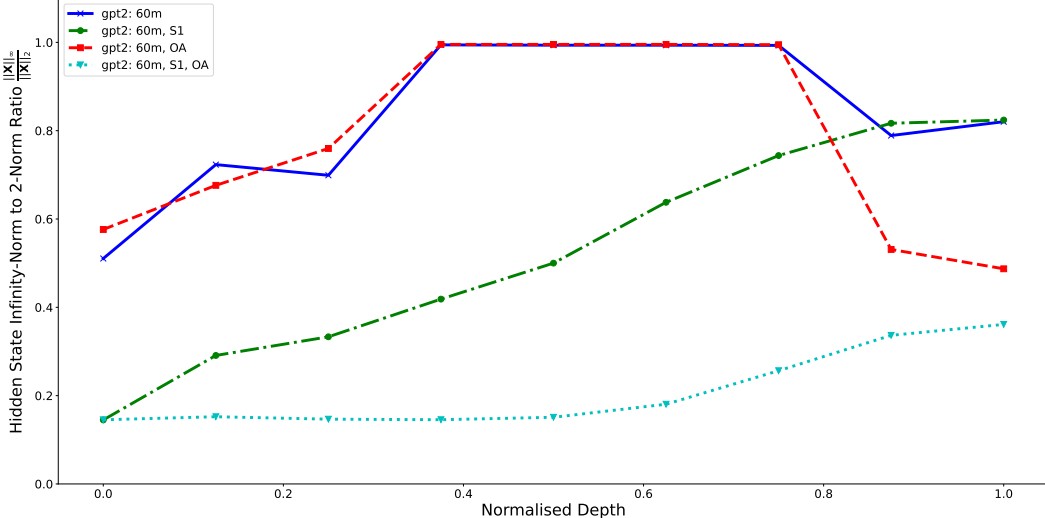

Figure 17: Layer-wise progression of the $\ell_\infty$-norm to $\ell_2$-norm ratio in the hidden states of the first token position for GPT2-60M. The x-axis is normalised to the range $[0, 1]$. S1/OA denote models trained with softmax-1 and/or OrthoAdam.

### I.3.2 GPT2-130M

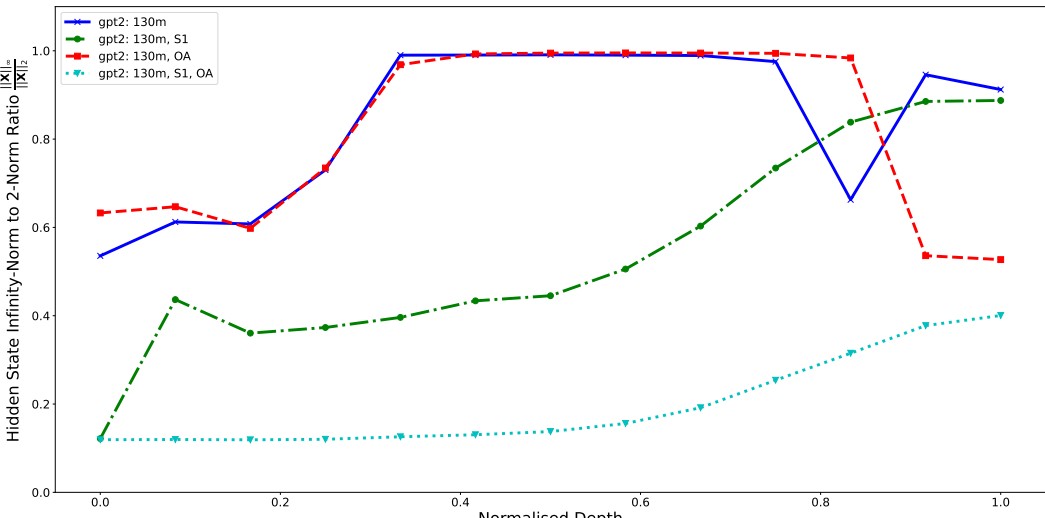

Figure 18: Layer-wise progression of the $\ell_\infty$-norm to $\ell_2$-norm ratio in the hidden states of the first token position for GPT2-130M. The x-axis is normalised to the range $[0, 1]$. S1/OA denote models trained with softmax-1 and/or OrthoAdam.

### I.3.3 GPT2-350M AND GPT2-1.4B

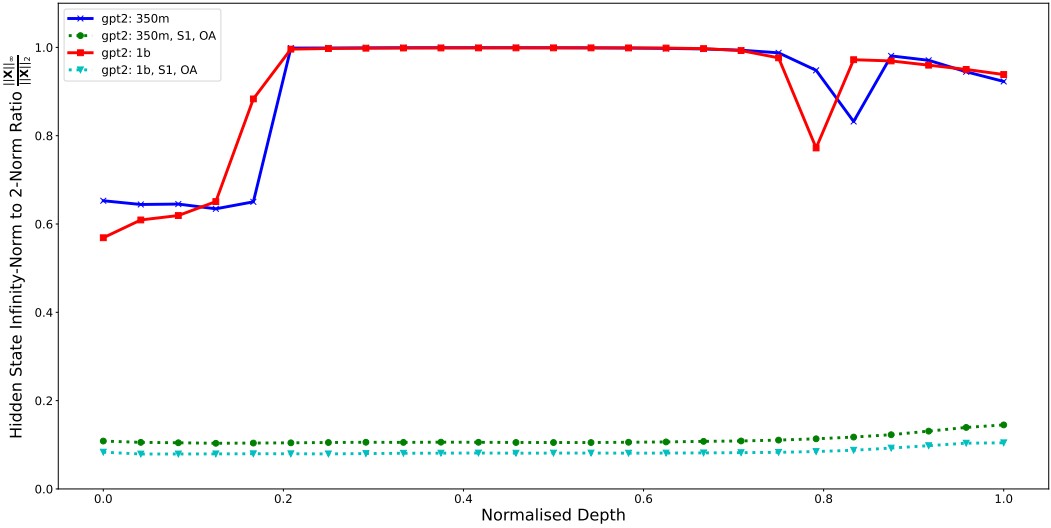

Figure 19: Layer-wise progression of the $\ell_\infty$-norm to $\ell_2$-norm ratio in the hidden states of the first token position for GPT2-350M and GPT2-1.4B. The x-axis is normalised to the range $[0, 1]$. S1/OA denote models trained with softmax-1 and/or OrthoAdam.

### I.3.4 LLAMA-130M

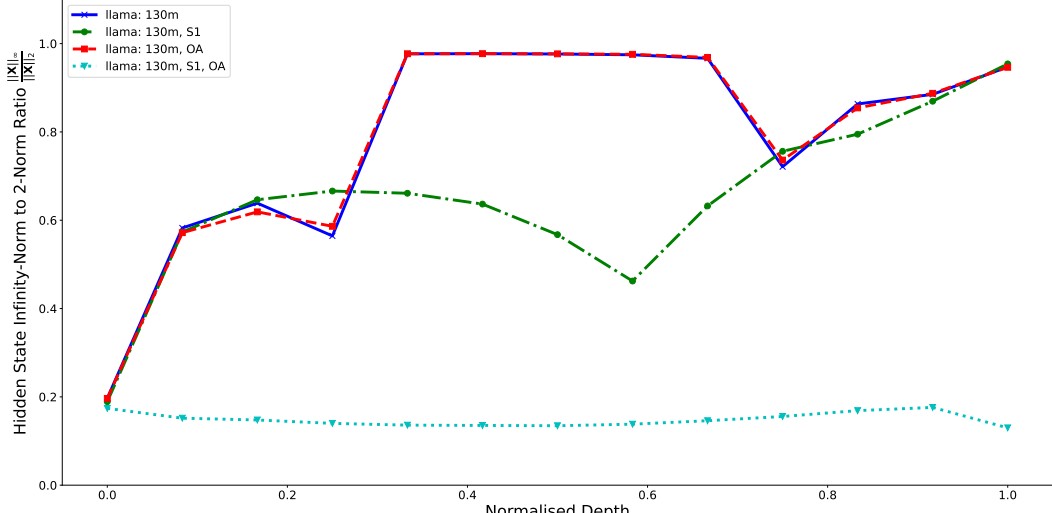

Figure 20: Layer-wise progression of the $\ell_\infty$-norm to $\ell_2$-norm ratio in the hidden states of the first token position for Llama-130M. The x-axis is normalised to the range $[0, 1]$. S1/OA denote models trained with softmax-1 and/or OrthoAdam.

### I.3.5 POPULAR PRETRAINED MODELS—GPT2 AND LLAMA

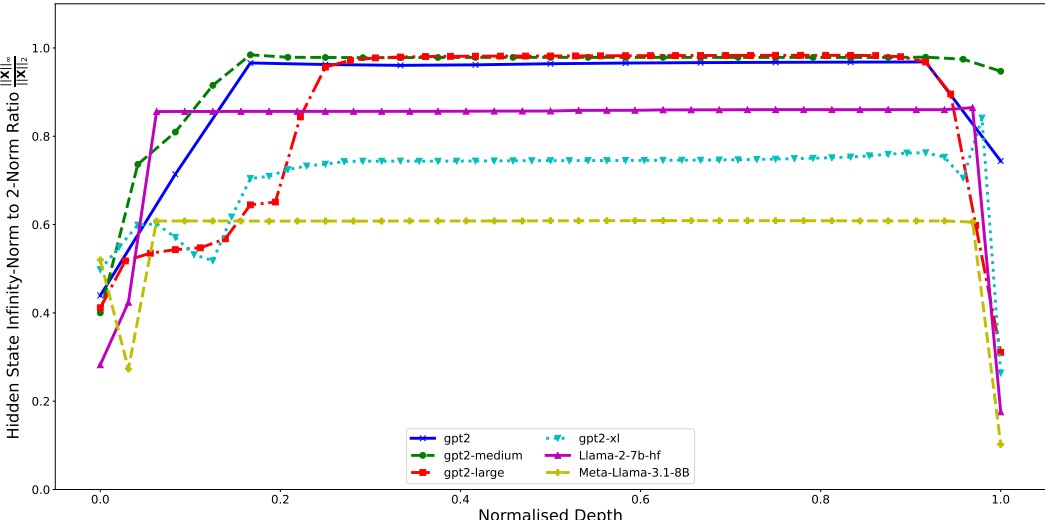

Figure 21: Layer-wise progression of the $\ell_\infty$-norm to $\ell_2$-norm ratio in the hidden states of the first token position for popular pretrained GPT2 and Llama models. The x-axis is normalised to the range $[0, 1]$.

To further clarify that in Transformer models the $\ell_\infty$-norm to $\ell_2$-norm ratio is a proxy for activation kurtosis, we calculate the Pearson's correlation coefficients between the two metrics for all models in our main experimental results from Table 2 and public GPT2 and Llama models. The results are shown in Table 10. We find a strong positive correlation between the two metrics across all models which reinforces our intuition that using orthogonal matrices to transform the gradients in the optimiser is an effective way to mitigate the emergence of large activation values, as an orthogonal transformation can reduce the $\ell_\infty$-norm of a vector substantially for a given $\ell_2$-norm.

| Model | #Parameters | Softmax+1? | OrthoAdam? | Correlation of Kurtosis to Norm Ratio First Token | Other Tokens |
|---|---|---|---|---|---|
| GPT2 (Ours) | 60M | ✓ | | 0.961 | 0.932 |
| | | | | 0.932 | 0.934 |
| | | | ✓ | 0.986 | 0.972 |
| | | ✓ | ✓ | 0.968 | 0.970 |
| | 130M | ✓ | | 0.988 | 0.932 |
| | | | | 0.927 | 0.924 |
| | | | ✓ | 0.992 | 0.962 |
| | | ✓ | ✓ | 0.935 | 0.953 |
| | 350M | | | 0.990 | 0.929 |
| | | ✓ | ✓ | 0.998 | 0.997 |
| | 1.4B | | | 0.988 | 0.952 |
| | | ✓ | ✓ | 0.994 | 0.995 |
| Llama2 (Ours) | 130M | ✓ | | 0.931 | 0.903 |
| | | | | 0.864 | 0.877 |
| | | | ✓ | 0.931 | 0.905 |
| | | ✓ | ✓ | 0.560 | 0.975 |
| GPT2 (Public) | 137M | | | 0.985 | 0.944 |
| GPT2-Medium (Public) | 350M | | | 0.969 | 0.846 |
| GPT2-Large (Public) | 812M | | | 0.985 | 0.896 |
| GPT2-XL (Public) | 1.6B | | | 0.956 | 0.939 |
| Llama2-7B (Public) | 6.7B | | | 0.987 | 0.902 |
| Llama3.1-8B (Public) | 8B | | | 0.928 | 0.915 |

Table 10: Correlation of the kurtosis and norm-ratio of the hidden states of our trained models and popular pretrained models.

### I.4 MAXIMUM ABSOLUTE ACTIVATION

Finally, we examine the progression of the maximum absolute activation across layers.

### I.4.1 GPT2-60M

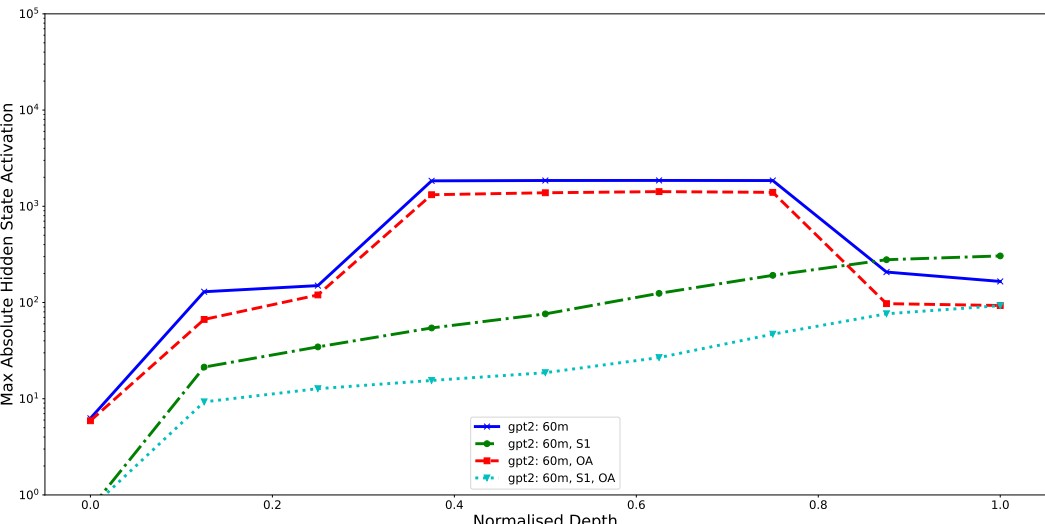

Figure 22: Layer-wise progression of the maximum absolute activation in the hidden states of the first token position for GPT2-60M. The x-axis is normalised to the range $[0, 1]$. S1/OA denote models trained with softmax-1 and/or OrthoAdam.

### I.4.2 GPT2-130M

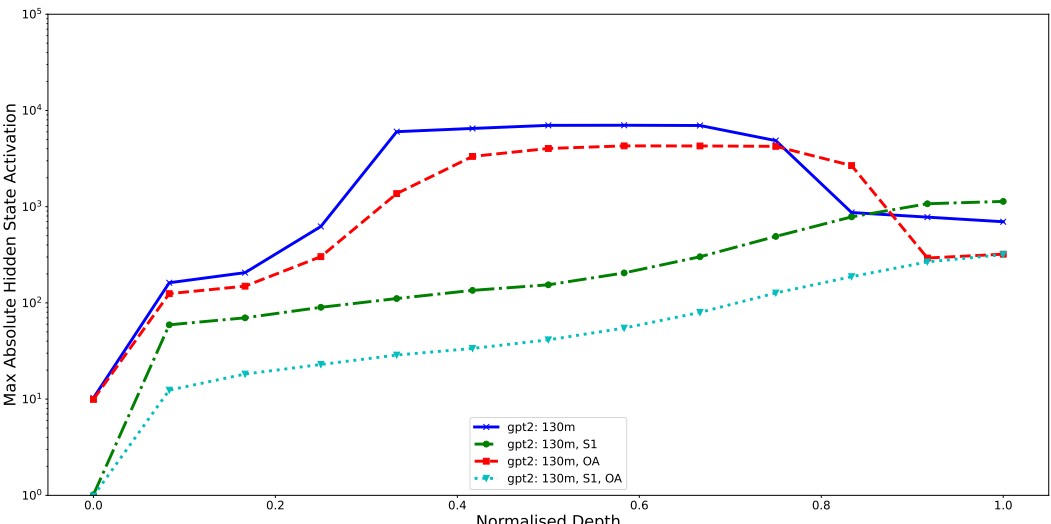

Figure 23: Layer-wise progression of the maximum absolute activation in the hidden states of the first token position for GPT2-130M. The x-axis is normalised to the range $[0, 1]$. S1/OA denote models trained with softmax-1 and/or OrthoAdam.

### I.4.3 GPT2-350M AND GPT2-1.4B

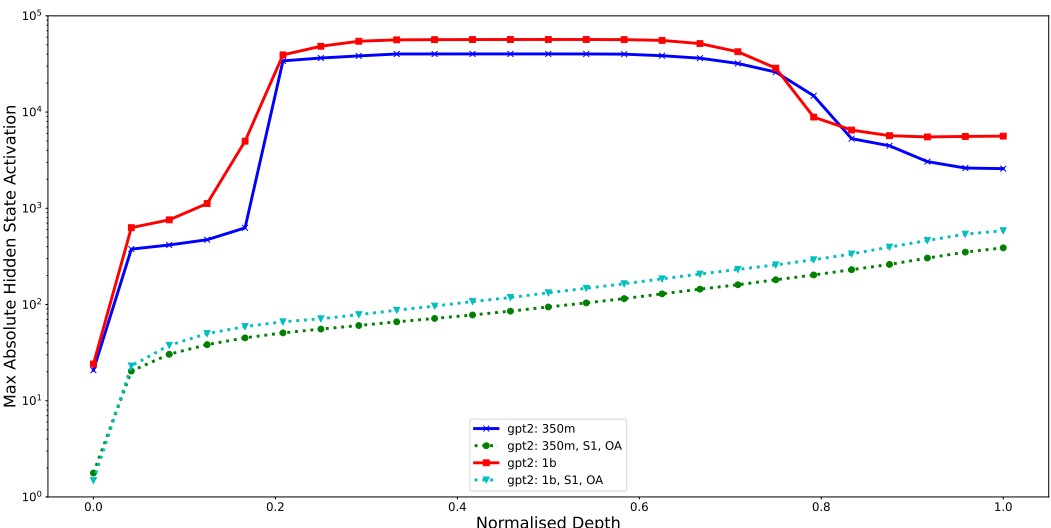

Figure 24: Layer-wise progression of the maximum absolute activation in the hidden states of the first token position for GPT2-350M and GPT2-1.4B. The x-axis is normalised to the range $[0, 1]$. S1/OA denote models trained with softmax-1 and/or OrthoAdam.

### I.4.4 LLAMA-130M

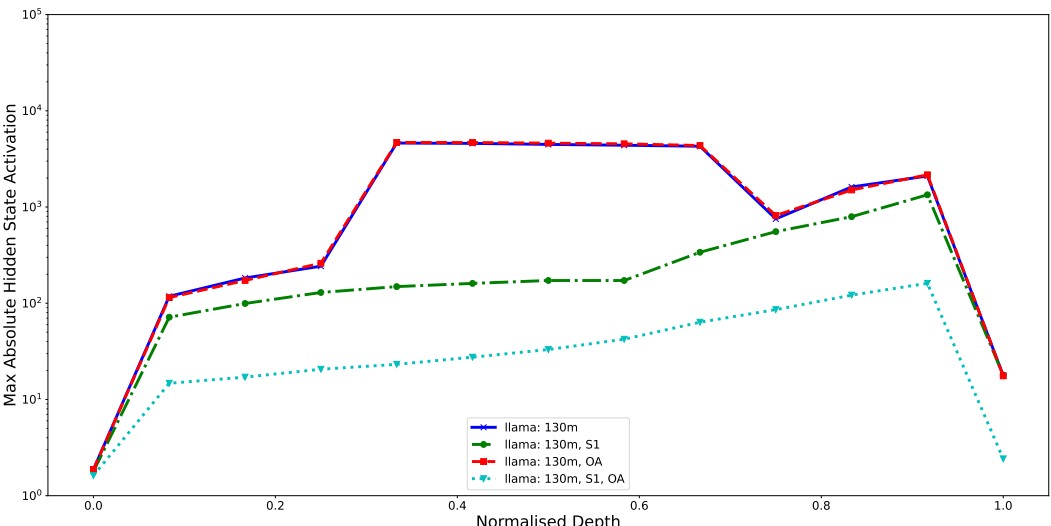

Figure 25: Layer-wise progression of the maximum absolute activation in the hidden states of the first token position for Llama-130M. The x-axis is normalised to the range $[0, 1]$. S1/OA denote models trained with softmax-1 and/or OrthoAdam.

### I.4.5 POPULAR PRETRAINED MODELS—GPT2 AND LLAMA

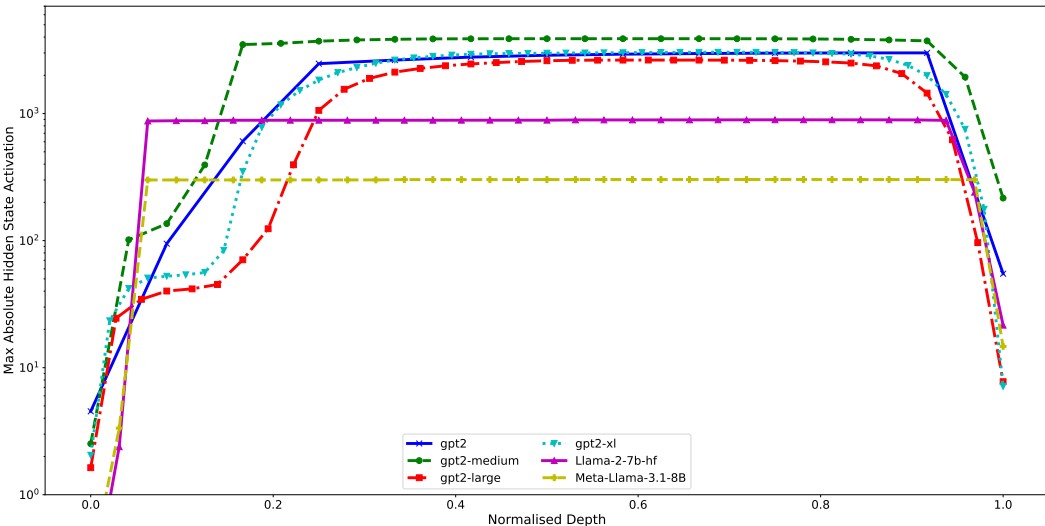

Figure 26: Layer-wise progression of the maximum absolute activation in the hidden states of the first token position for popular pretrained GPT2 and Llama models. The x-axis is normalised to the range $[0, 1]$.

## J  HIDDEN STATES OF PRETRAINED MODELS

In this section, we present the progression of hidden states of popular pretrained models. This shows how models establish outlier activations and how they persist in the same feature dimensions across layers. For each model we show the *absolute* activation values in the features containing the largest activations. We show the mean across layers, the first layer, $\frac{1}{4}$ and $\frac{3}{4}$ of the layers.

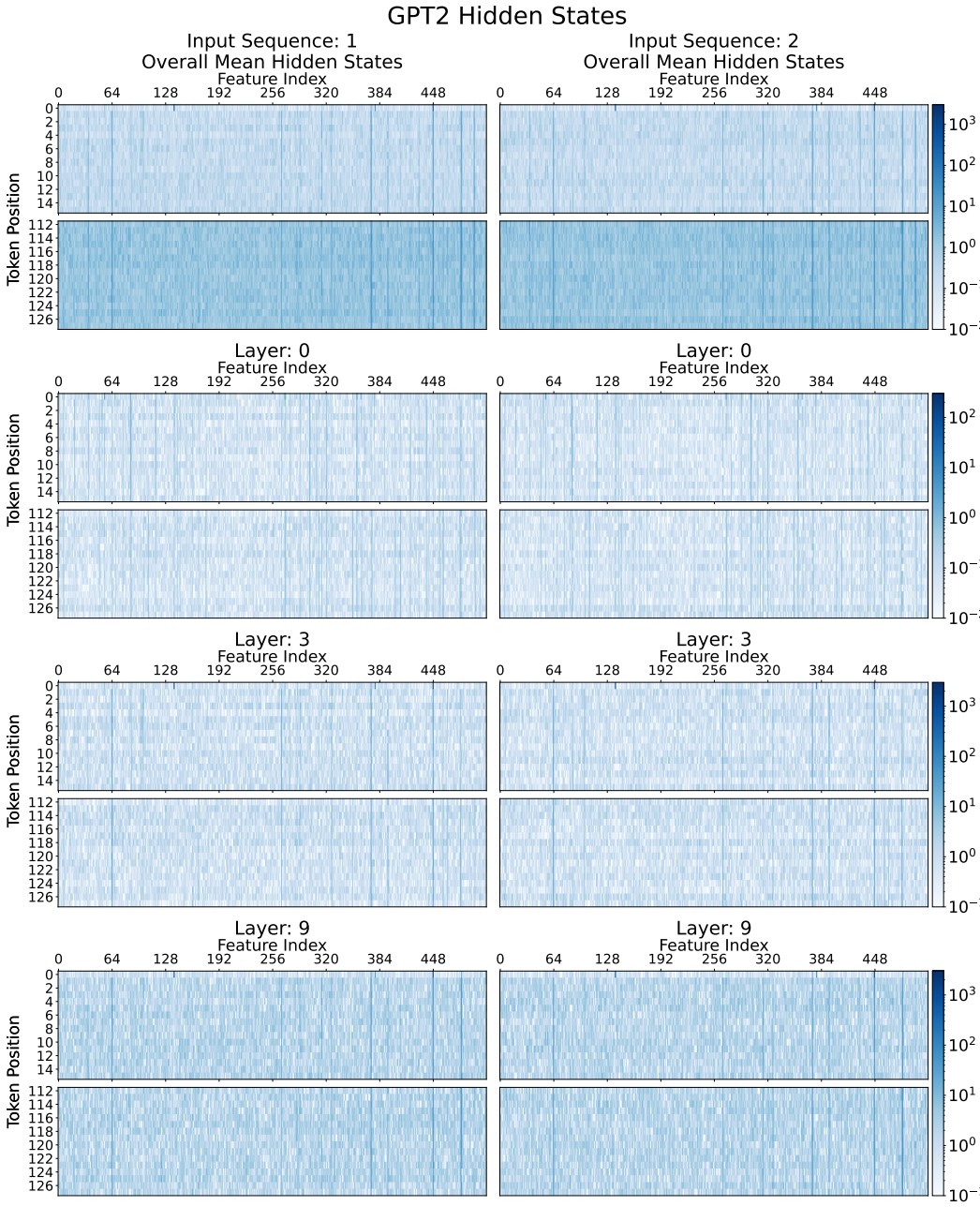

Figure 27: Example hidden state plots for a GPT2-Small model.

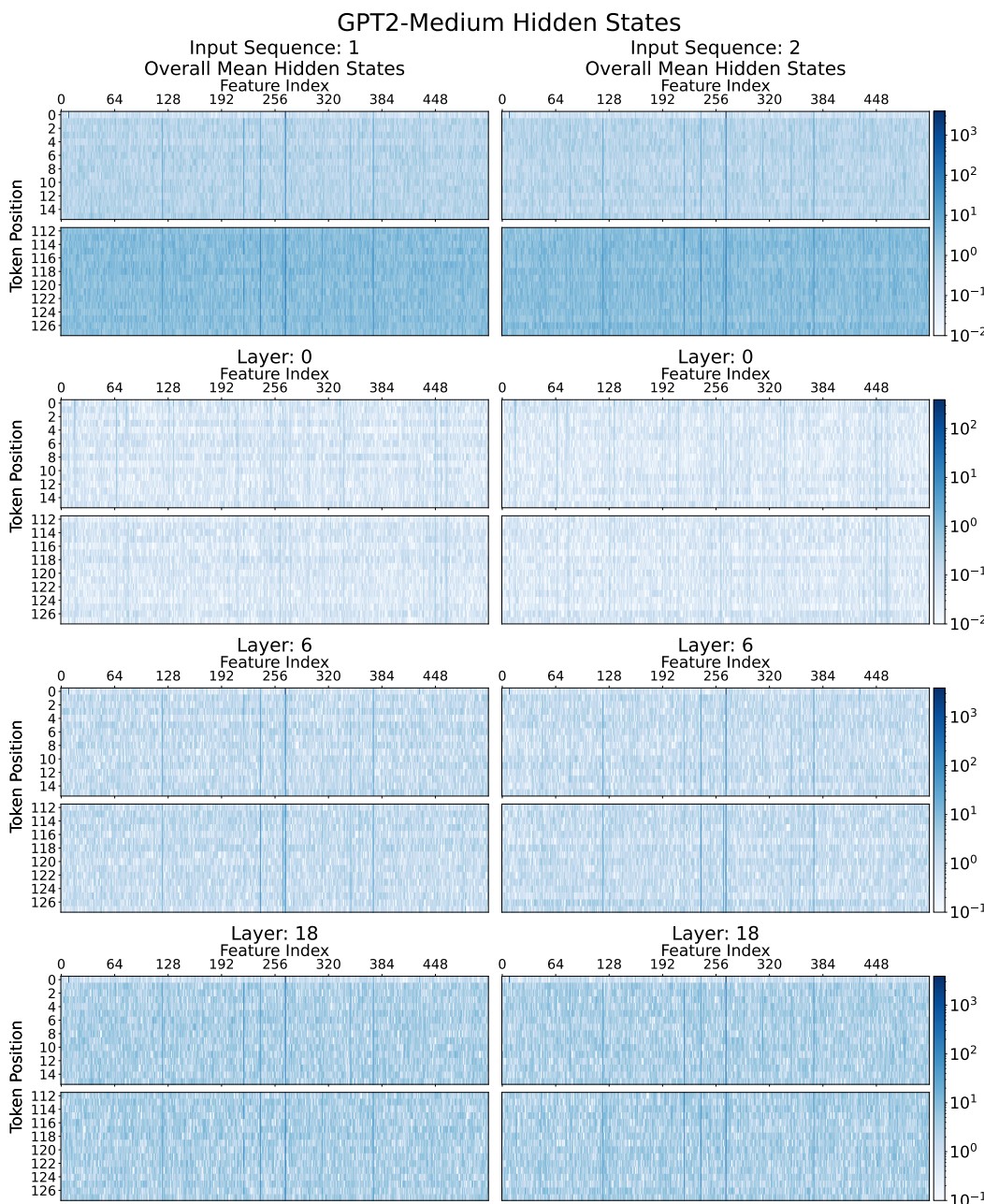

Figure 28: Example hidden state plots for a GPT2-Medium model.

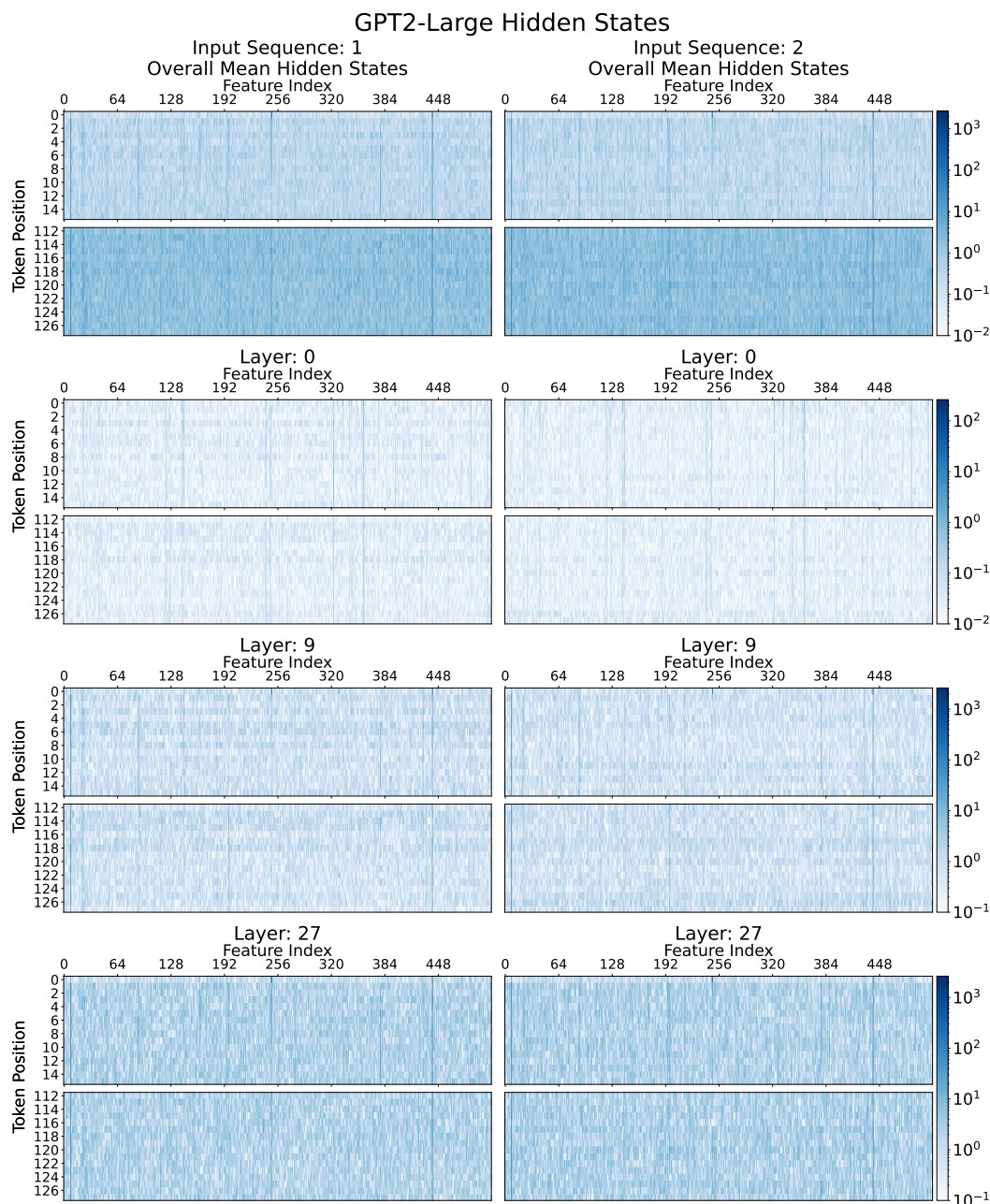

Figure 29: Example hidden state plots for a GPT2-Large model.

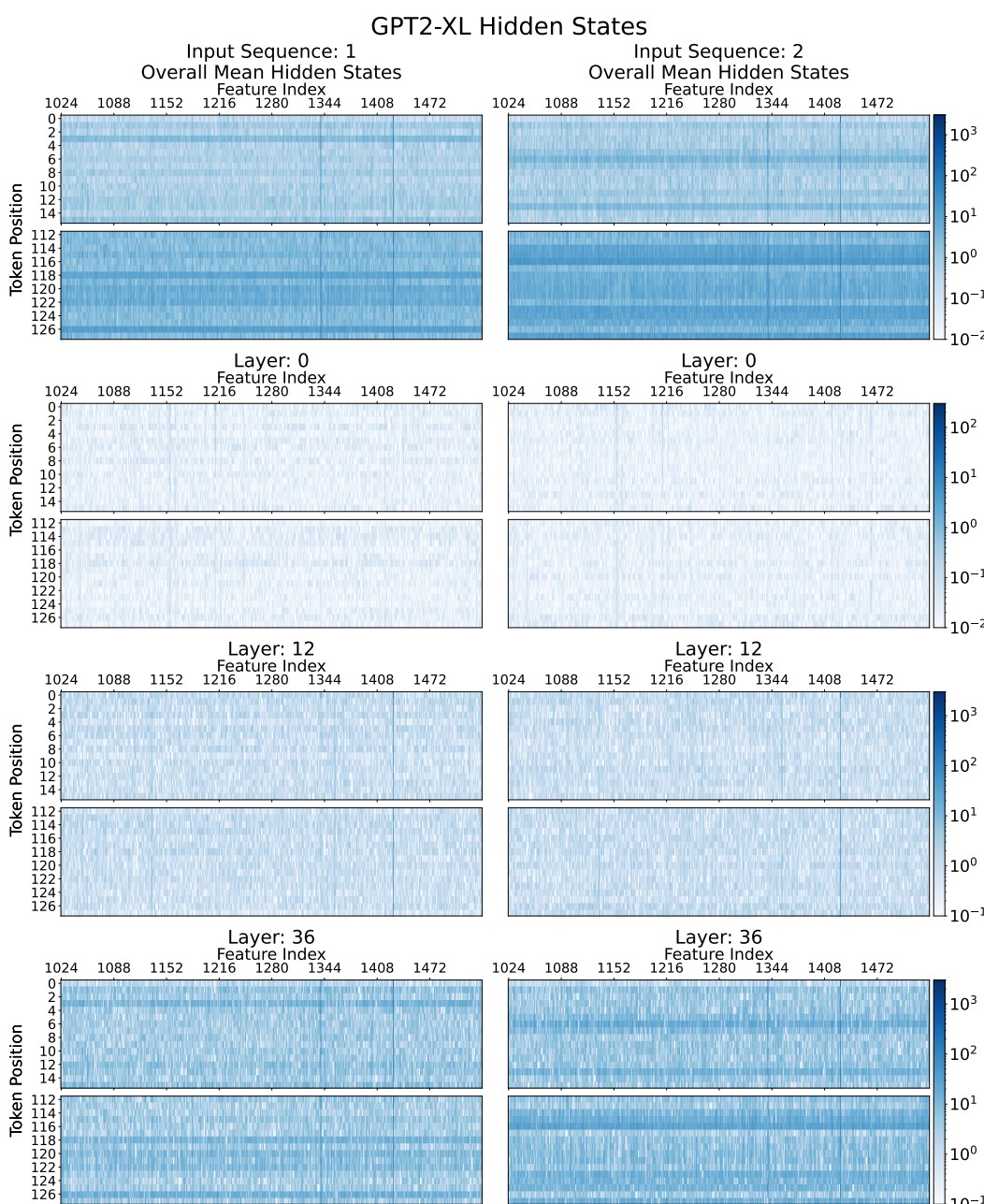

Figure 30: Example hidden state plots for a GPT2-XL model.

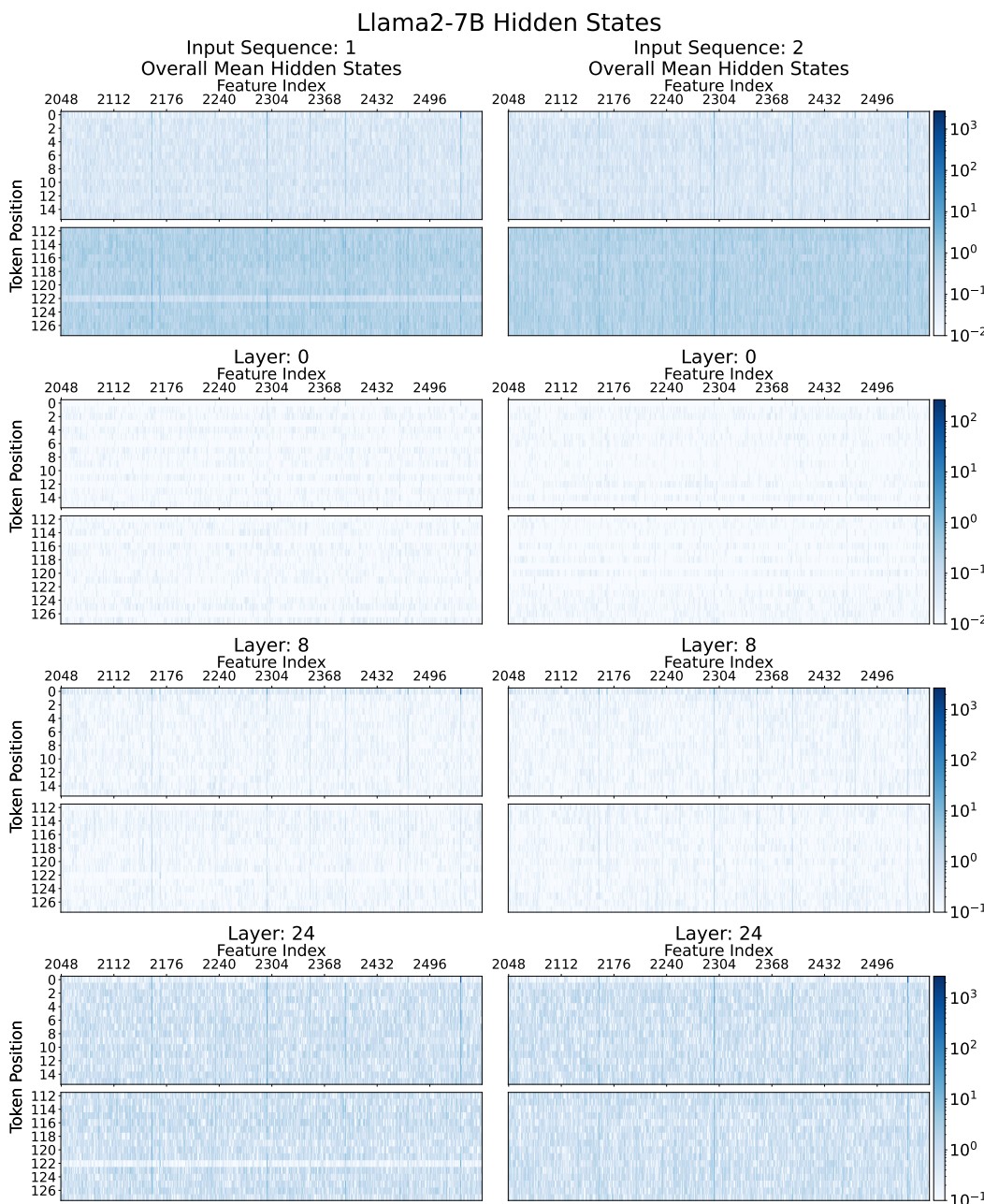

Figure 31: Example hidden state plots for a Llama2-7B model.

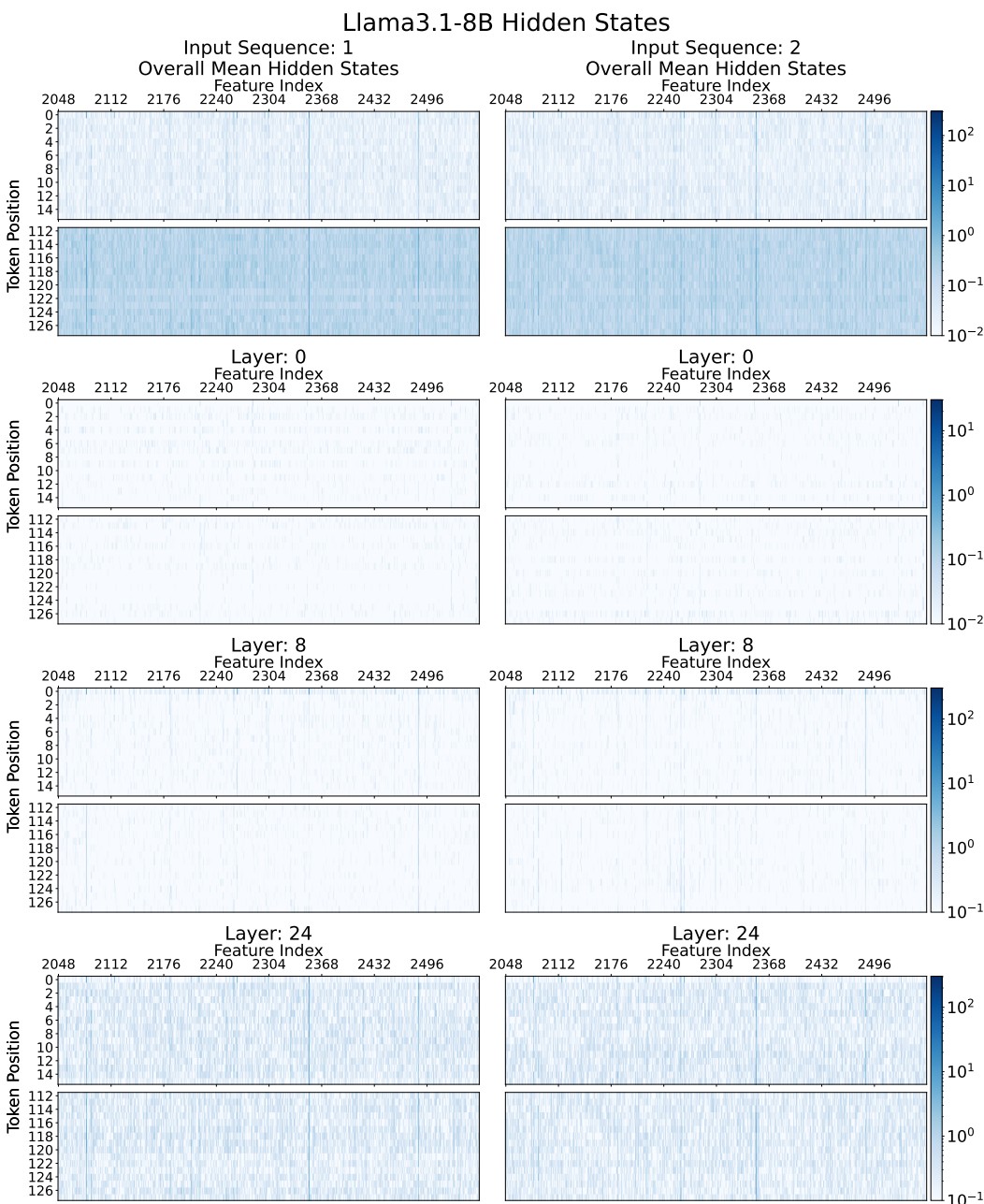

Figure 32: Example hidden state plots for a Llama3.1-8B model.

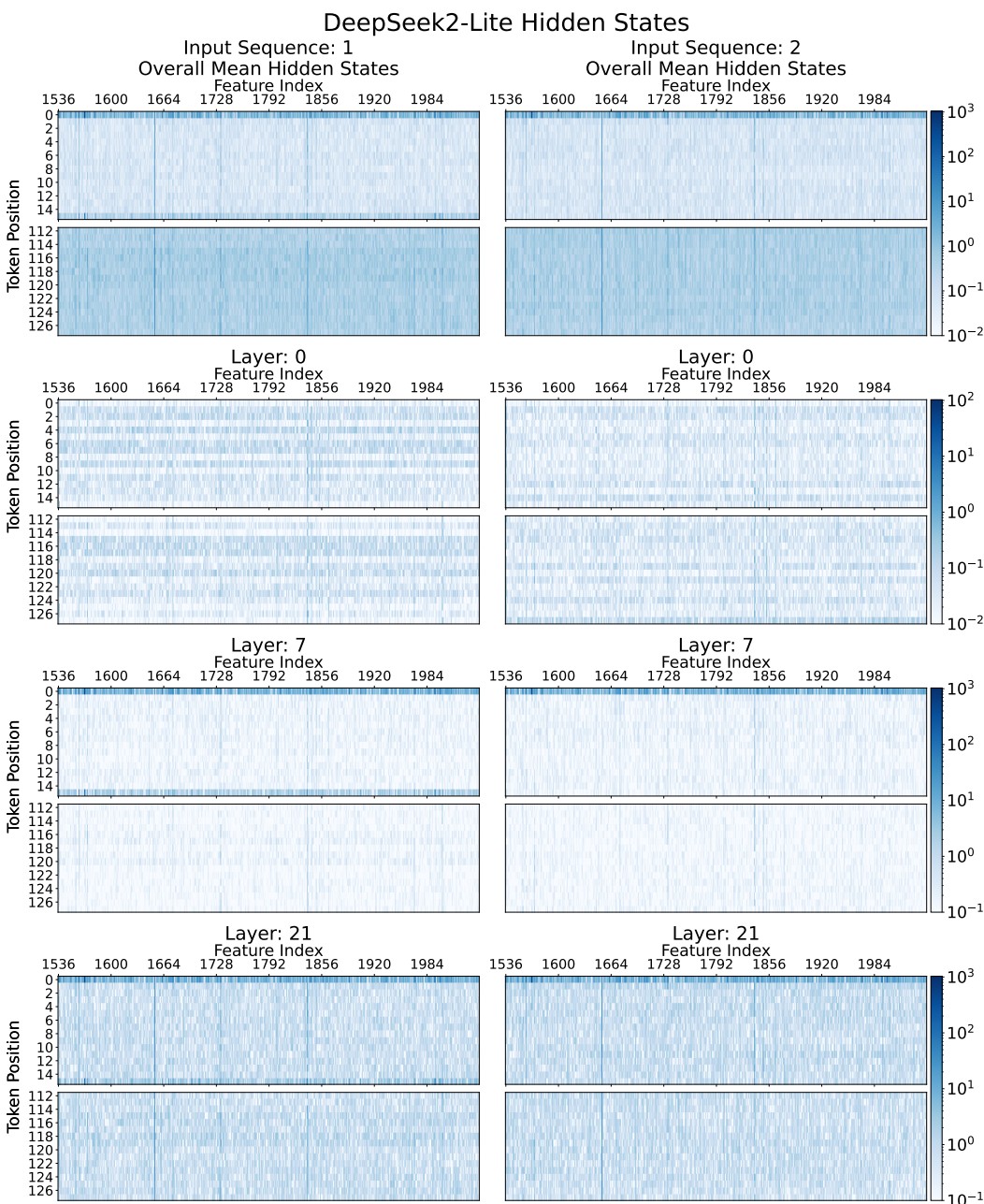

Figure 33: Example hidden state plots for a DeepSeekv2-Lite model.

# K ATTENTION MAPS OF PRETRAINED MODELS

In this section, we present the attention maps of popular pretrained models. This shows how models establish attention patterns and how they persist after initial layers. This shows that generally after the first or second layer, first token attention dominance is highly established and persists across layers.

We show the mean across layers, the first layer, $\frac{1}{4}$ and $\frac{3}{4}$ of the layers—averaging over all heads in each case.

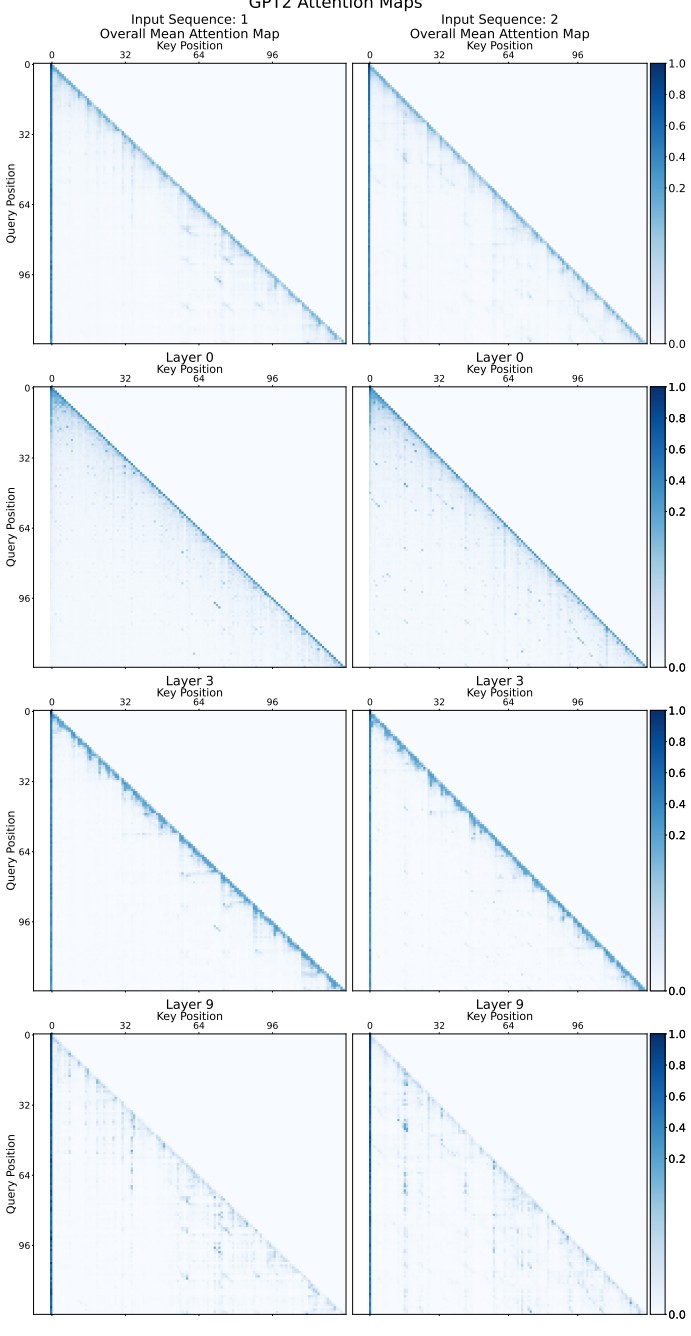

Figure 34: Example attention maps for a GPT2-Small model.

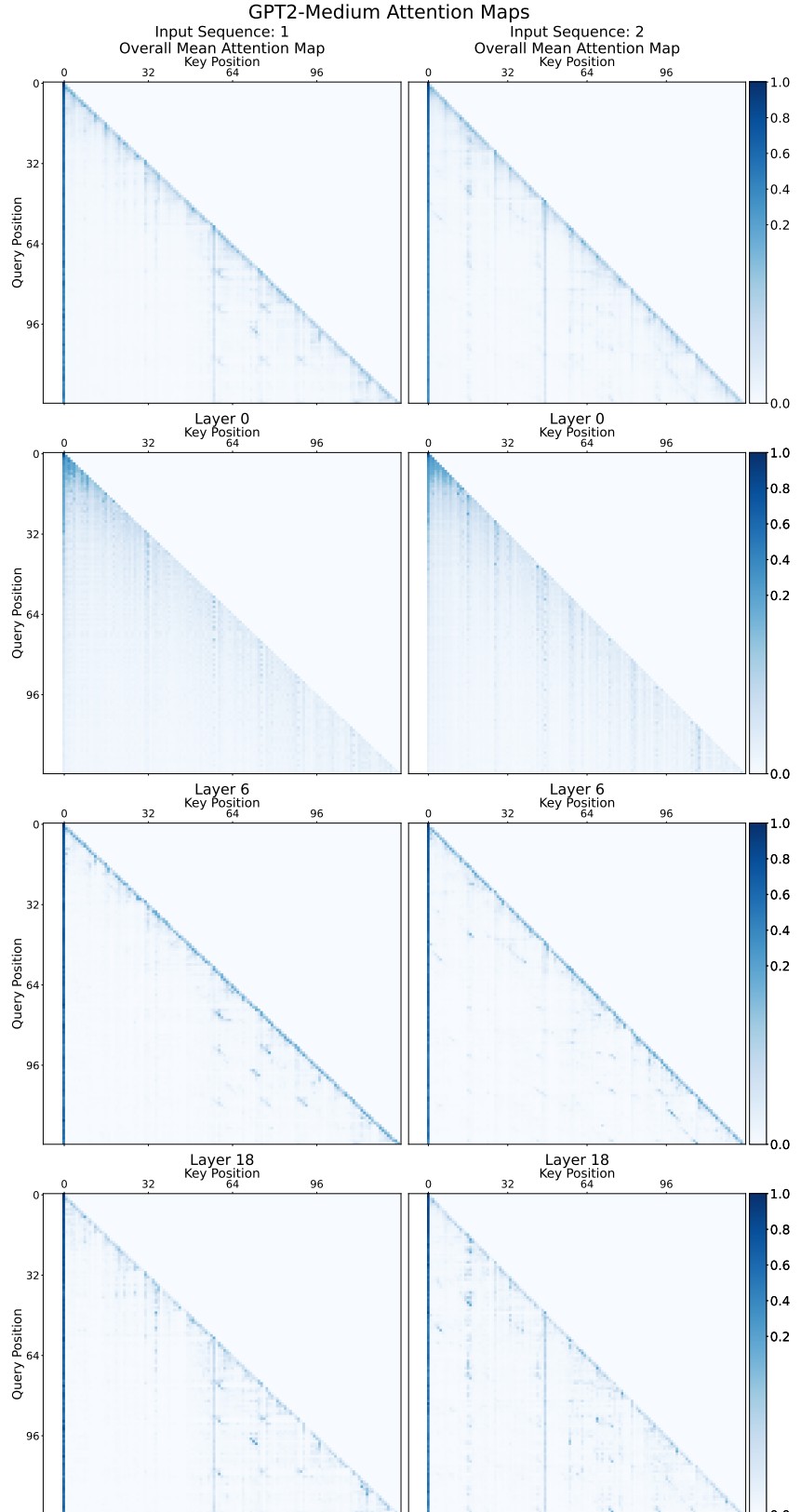

Figure 35: Example attention maps for a GPT2-Medium model.

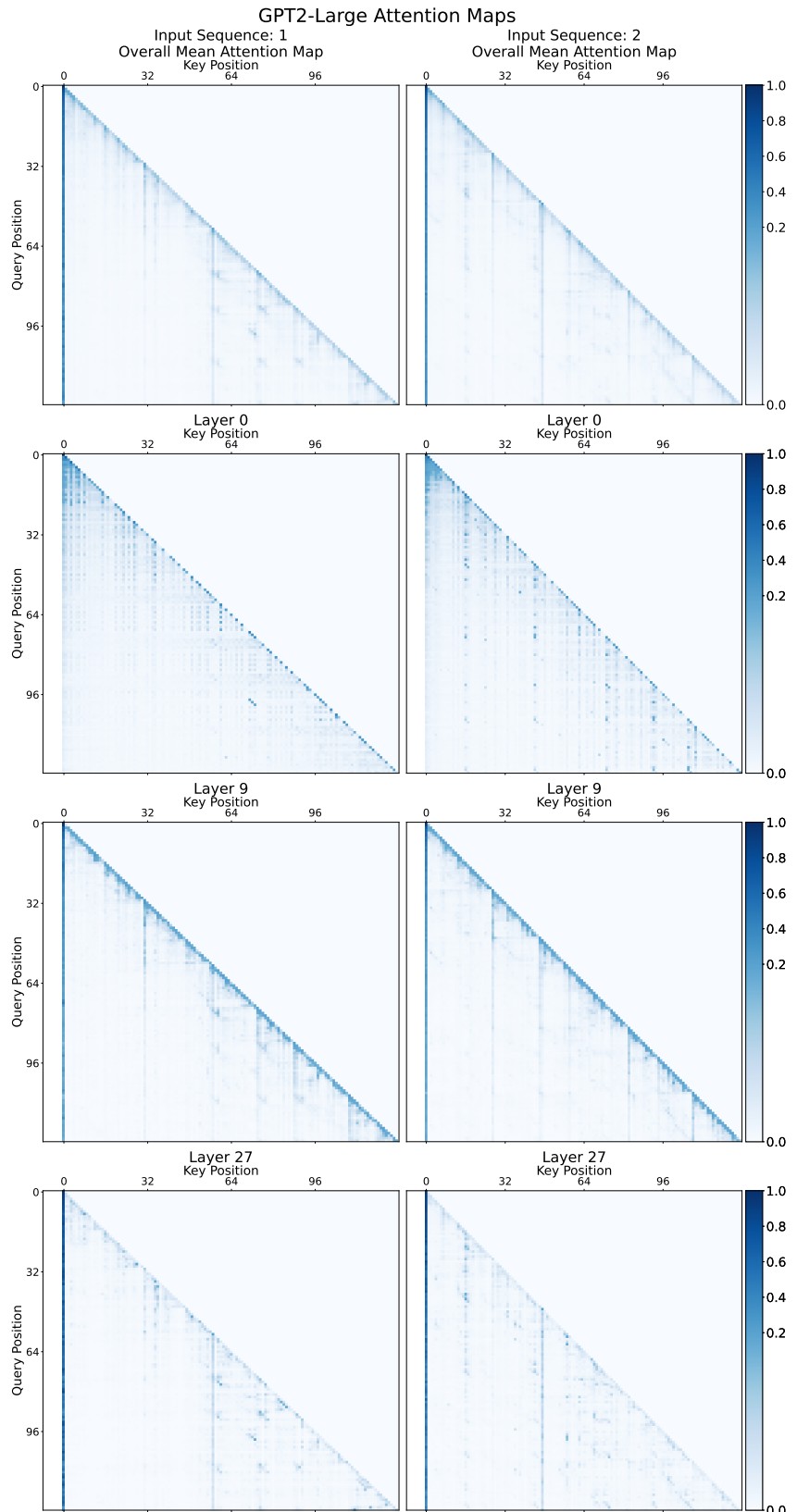

Figure 36: Example attention maps for a GPT2-Large model.

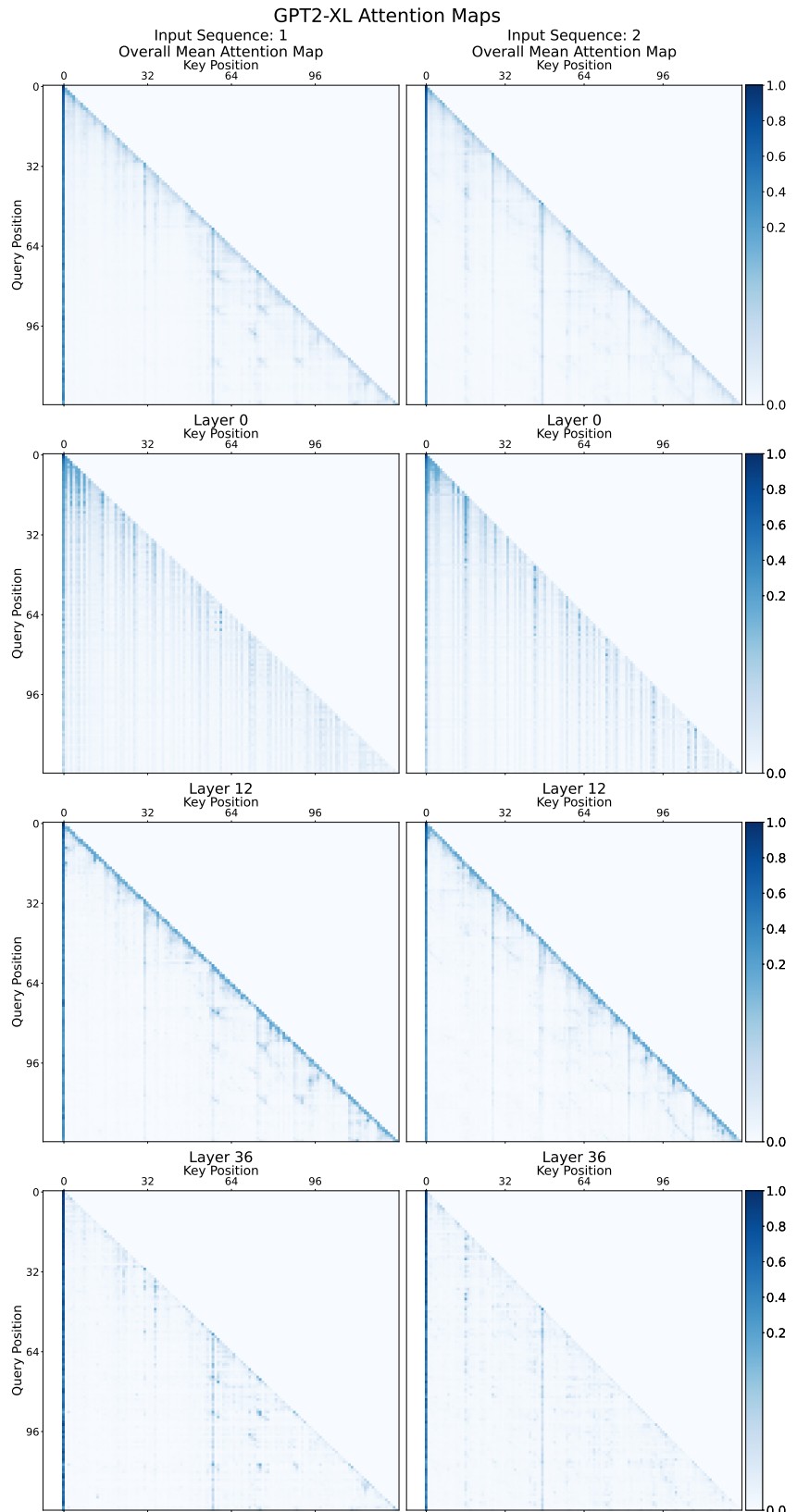

Figure 37: Example attention maps for a GPT2-XL model.

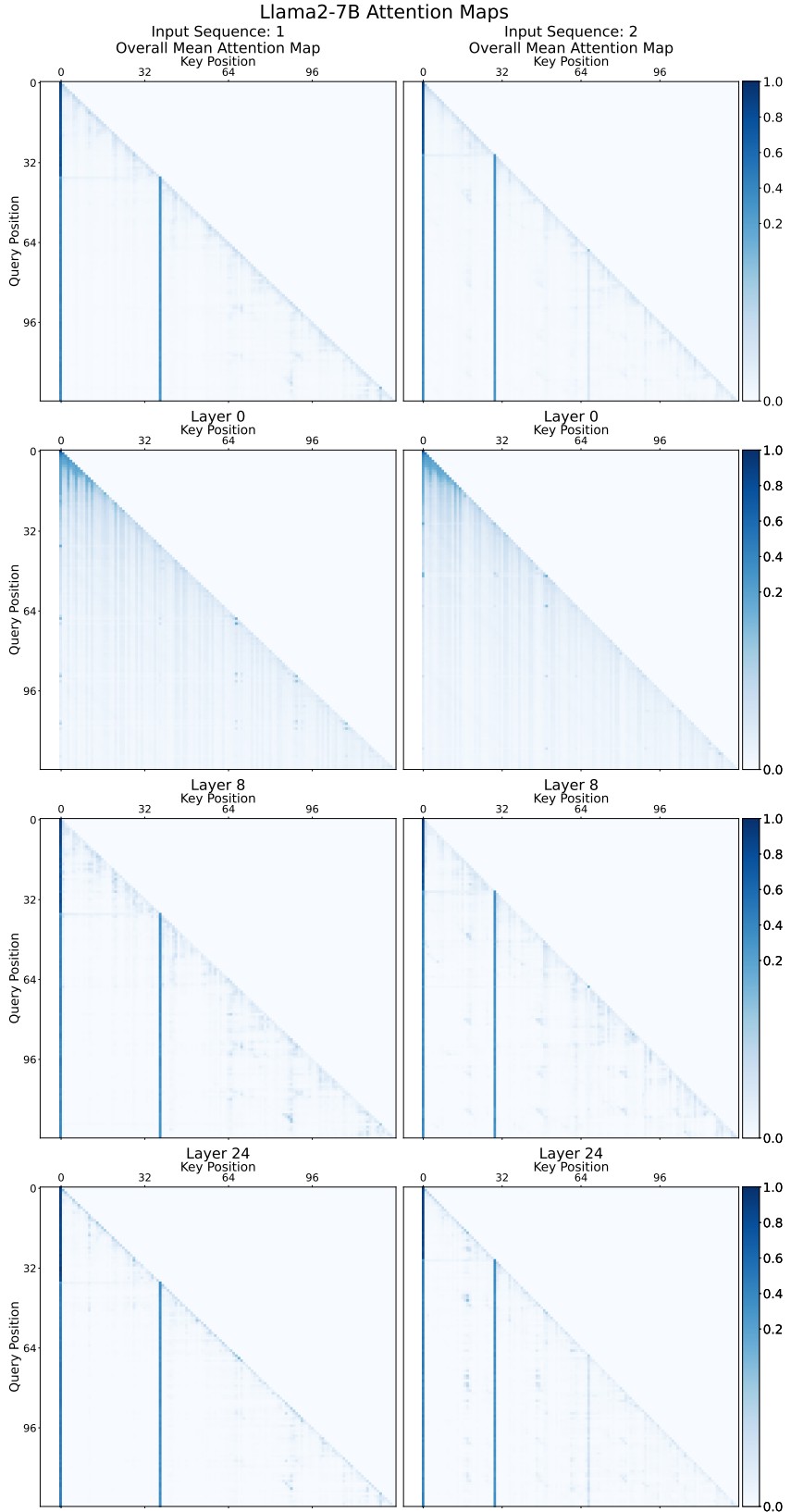

Figure 38: Example attention maps for a Llama2-7B model.

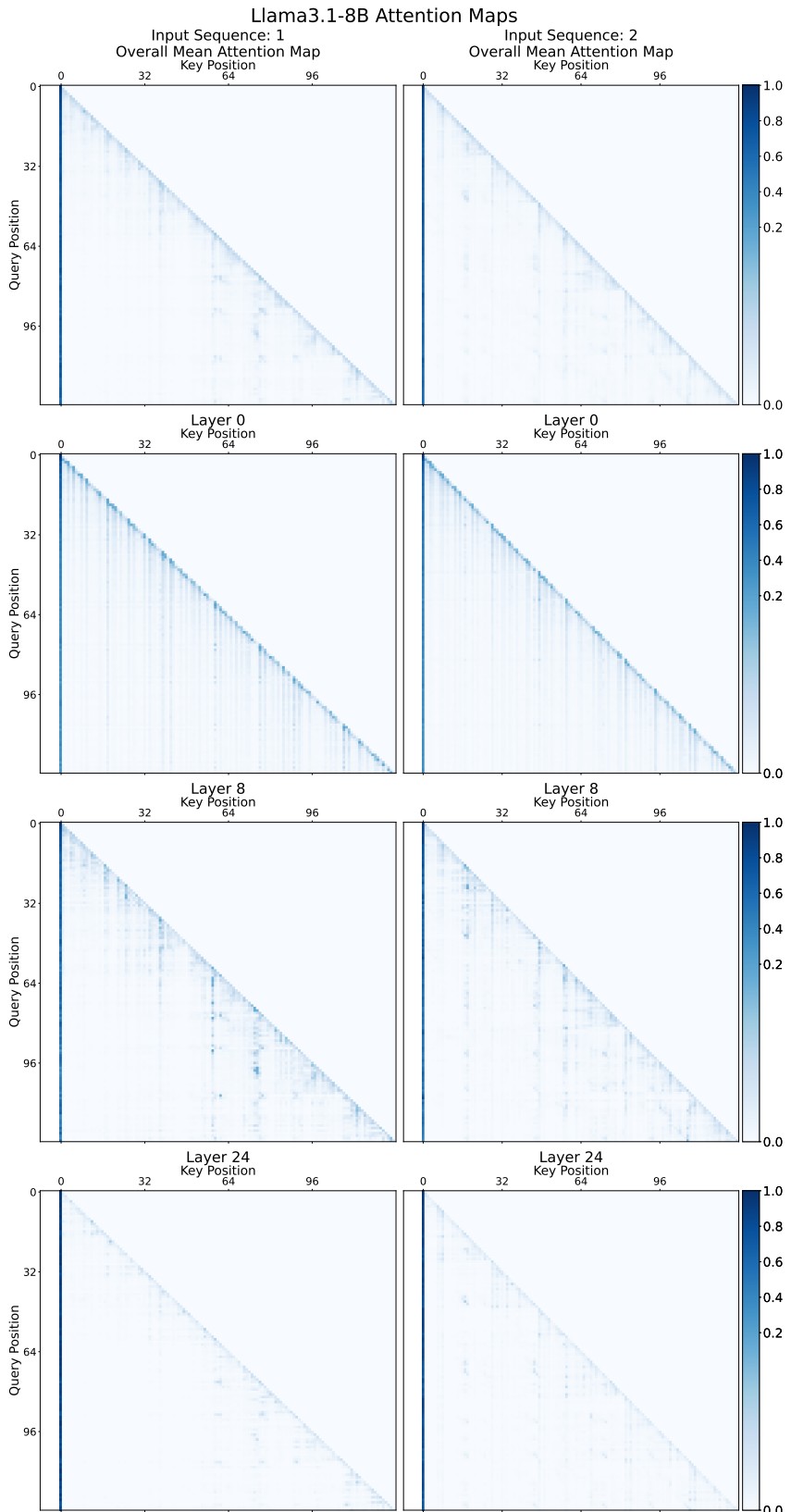

Figure 39: Example attention maps for a Llama3.1-8B model.

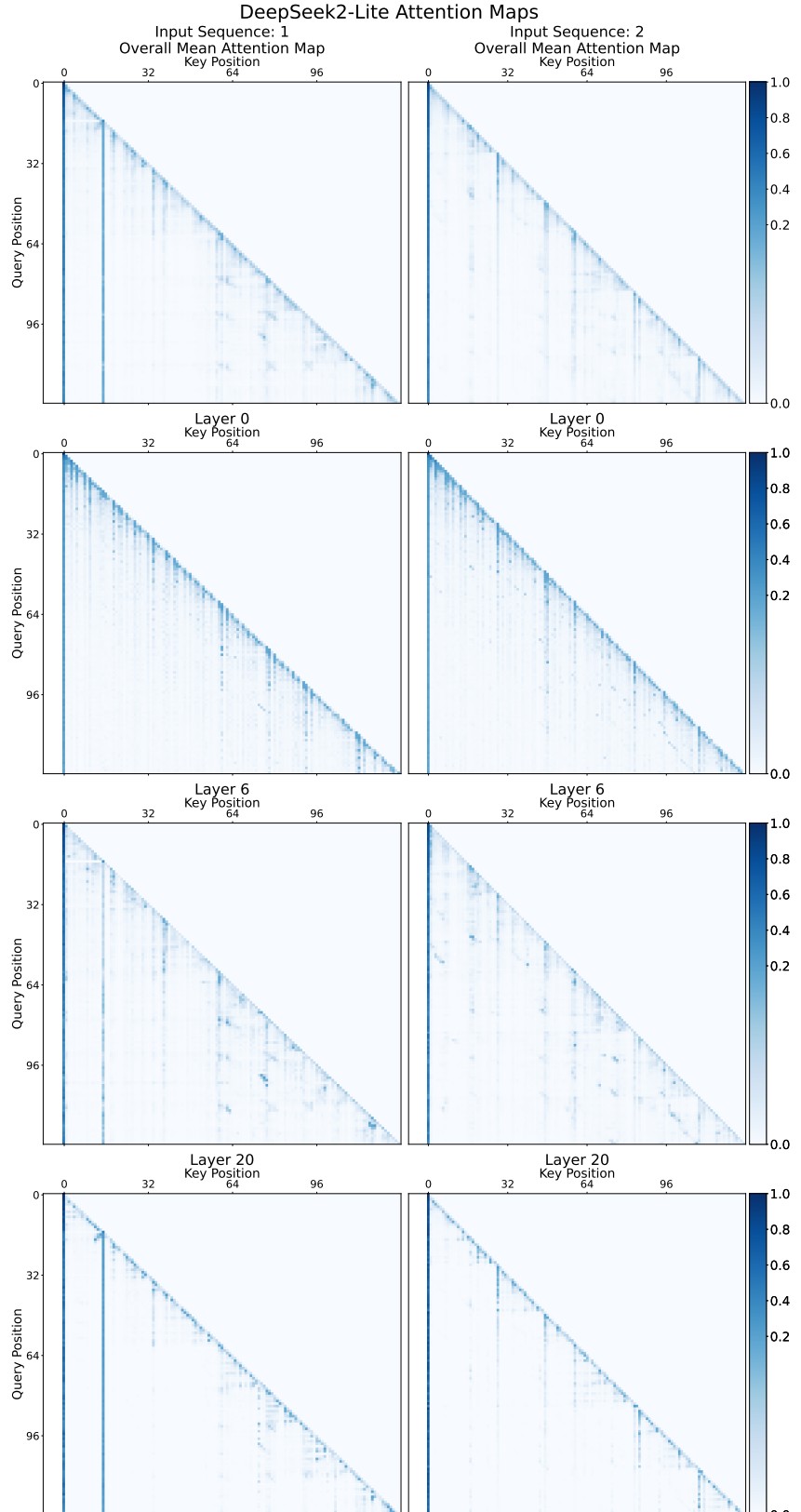

Figure 40: Example attention maps for a DeepSeekv2-Lite model.

# L    TRAINING CURVES

To demonstrate that our proposed methods, *i.e.*, replacing the canonical softmax function with *softmax-1* and using our proposed optimiser, *OrthoAdam*, do not negatively impact the training of large language models, we provide the training curves for our models here. One can observe that the training curves for models using either or both of our proposed changes are stable and converge to a similar loss value as the baseline models.

## L.1    GPT2-60M

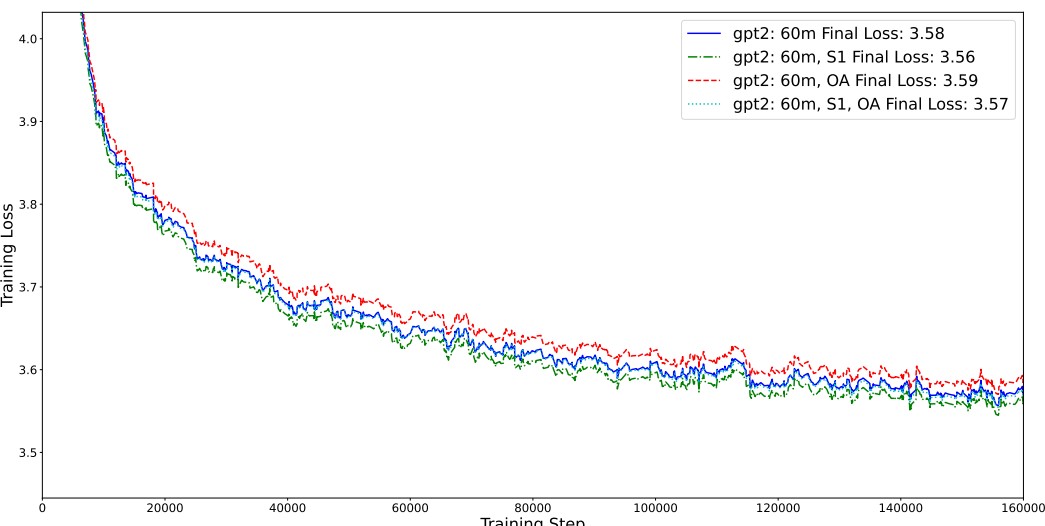

Figure 41: Training curves for GPT2-60M models with different optimisers and softmax functions. The models using *OrthoAdam* and *softmax-1* are stable and converge to a similar loss value as the baseline models. S1/OA denotes the model using softmax-1 and/or OrthoAdam.

## L.2    GPT2-130M

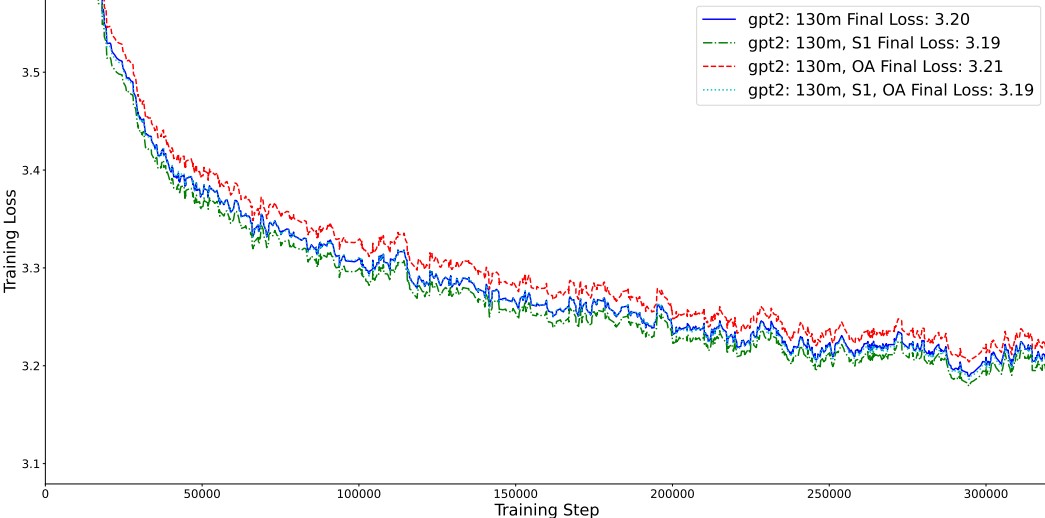

Figure 42: Training curves for GPT2-130M models with different optimisers and softmax functions. The models using *OrthoAdam* and *softmax-1* are stable and converge to a similar loss value as the baseline models. S1/OA denotes the model using softmax-1 and/or OrthoAdam.

## L.3    GPT2-350M AND GPT2-1.4B

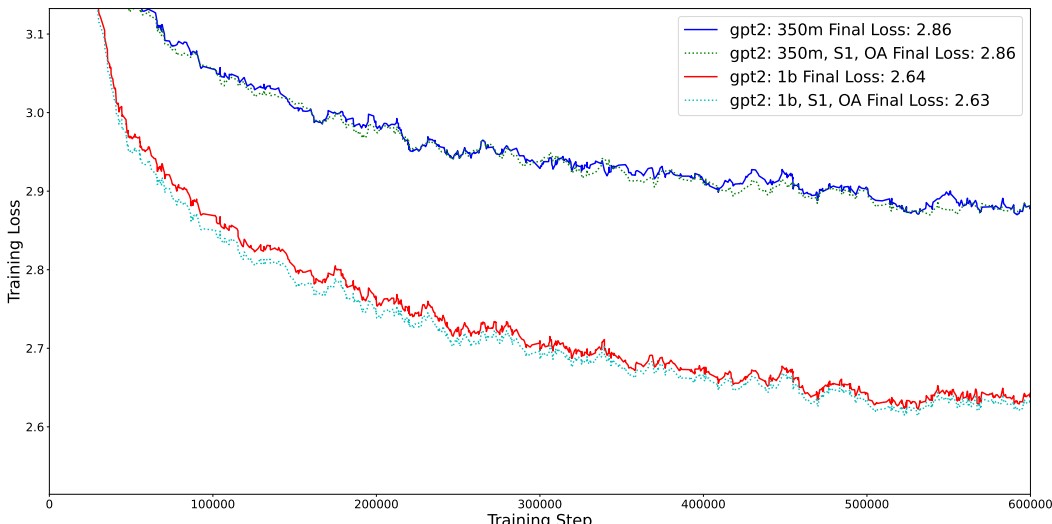

Figure 43: Training curves for GPT2-350M and GPT2-1.4B models with different optimisers and softmax functions. The models using *OrthoAdam* and *softmax-1* are stable and converge to a similar loss value as the baseline models. S1/OA denotes the model using softmax-1 and/or OrthoAdam.

## L.4    LLAMA-130M

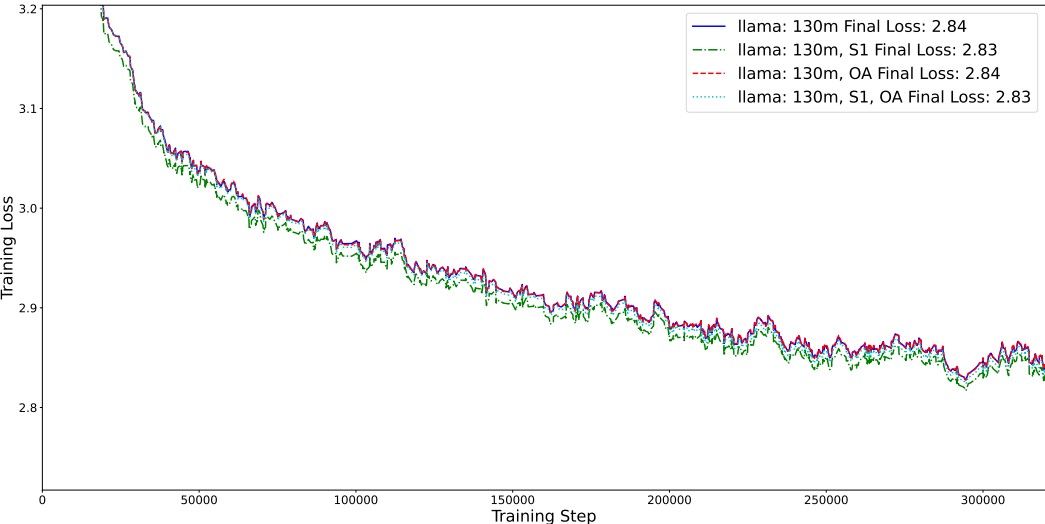

Figure 44: Training curves for Llama-130M models with different optimisers and softmax functions. The models using *OrthoAdam* and *softmax-1* are stable and converge to a similar loss value as the baseline models. S1/OA denotes the model using softmax-1 and/or OrthoAdam.

