# OpenReview forum: "From Attention to Activation: Unraveling the Enigmas of Large Language Models"
_ICLR.cc/2025/Conference — ICLR 2025 Poster_

### Official Review · Reviewer_hVNT · 2024-10-27

**Soundness:** 3
**Presentation:** 3
**Contribution:** 3
**Rating:** 6
**Confidence:** 3

**Summary:**

This paper studies two interesting phenomena in transformer-based LLMs. First, the large amount of attention the first token receives. Second, the emergence of outlier neurons in the transformer layer activation. They attribute the first phenomenon to the softmax function, and the second to the Adam optimizer. They propose variants of these two methods (softmax-1 and OrthoAdam), which resolve these two issues. They show that using these methods allows models to quantize far better than without them.

**Strengths:**

- Interesting and important research questions
- A well executed study. In particular, several hypotheses are considered for the phenomena in question, and the authors are able to pin-down on what seems to be their root cause
- Very interesting findings
- Practical results that allow simpler quantization

**Weaknesses:**

- softmax-1 has been introduced before in the streamingLLM paper (https://arxiv.org/abs/2309.17453, referred to as $Softmax_1$ or Zero-sink in that work). The authors in that work showed that "_while the zero sink alleviates the attention sink problem to some extent, the model still relies on other initial tokens as attention sinks_". This work seems to argue differently. It seems important to pin down the differences between both setups that lead to the different behaviors.

In addition, I found some of the major claims to require further discussion.

- From table 4, it seems RMSNorm-s is also important for mitigating the outlier effect. This is currently not really
discussed, and the authors only mention softamx-1 and OrthoAdam as the important parts.

- The softmax-1 method requires some discussion and intuition. Why is the specific method appropriate here? Specifically, the authors say (#208) that it allows "having low attention scores everywhere." But standard softmax also allows that, e.g., by assigning uniform attention scores to all tokens. Is it about a better inductive bias? It would be helpful to see a more detailed comparison between the two methods. E.g., an explanation of why softmax-1 is more effective than standard softmax for this task, and/or a discussion of any potential trade-offs or downsides to using softmax-1.

- If I understand the discussion around OrthoAdam correctly, the authors say the reason for the outliers is (a) the appearance of high values of features, and (b) that optimizers like Adam lead to such high values. Is my understanding correct? If so, this should be stated more explicitly.



Minor:

- Using *x*s to contrast with *v*s in table 2 (as used in tables 3 and 4) would make it clearer than leaving it blank.

Also, there are several typos and such in the paper:

- #195: "... therefore it *in* receives ..." (drop "in")
- #306: "... under *an* particular ..." (should be "a")
- #313: " the model *during* i.e.  ..." (something's strange here, either drop during or add something after it)
- #420: "Note that only linear layers are *quantised the* embeddings, ..." (Missing period/, while)

**Questions:**

- As the authors note, Llama2 remains usable after quantization even in the vanilla version. Do the authors have some intuition regarding this?

- I had a hard time understanding the discussion in #222 around causal masking. What does it mean to relax causal masking? train with an MLM loss? and why would the fifth token dominate in this case?

---

> ### Author Response · Authors · 2024-11-24
> **Response to Reviewer hVNT**
>
> We thank the reviewer for finding our study **interesting**, **important**, and **well-executed** and **practical**. Below, we address the reviewer's main concerns.
>
> **Weakness 1: StreamingLLM comparison**
>
> The code regarding the experiments in Table 3 of the work is not available and so it is
> difficult to explain why the authors of Streaming LLM do not find “Zero Sink” usable.
> One thing to note for readers of Table 3 is that in the caption they say “Perplexity is
> evaluated on the first sample in the PG19 test set.” It is difficult to take general
> performance results from a single example.
>
> We would also note that StreamingLLM focuses on evaluating models once the context length of an
> input text has exceeded the length of the model’s context window and therefore initial tokens
> are evicted. We evaluate softmax1 under the standard language modeling case in which all
> previous tokens are in the context window. Therefore the activation function may be the same but
> the experiments run are fundamentally different and contain zero overlap with our work. (E.g.
> they do not show the attention on the first text token to show the effect of softmax1 on
> attention domination).
>
> Moreover as explained by the authors in a GitHub issue
> (https://github.com/mit-han-lab/streaming-llm/issues/44#issuecomment-1768958019),
> the large 0+1024 result for ZeroSink is because the model has effectively been trained with
> softmax-1 but evaluated with canonical softmax, it is therefore expected to see a large
> difference in performance when the attention activation function has been changed.
>
> Finally, it is slightly unclear from their work whether position encodings are added to the “ZeroSink” — see Section 3.2 paragraph 2 and 3 which seem to indicate positional information is added to the “ZeroSink” in which case the ZeroSink formulation and using softmax1 are not equivalent. Future work would investigate if our formulation of softmax-1 can be extended successfully to the streaming case as in StreamingLLM.
>
> **Weakness 2: RMSNorm-S**
>
> The reviewer is indeed correct that RMSNorm-S is required to mitigate outliers.
> LayerNorm applies a scale per-channel and a bias,
> both of these operations could potentially cause outliers in a specific channel,
> either through the bias or through the per-channel scale.
> The standard RMSNorm (RMSNorm-M) again applies a per-channel scale and so could cause outliers
> in a specific channel. Both of these operations are “basis-dependent”,
> i.e. given an input and corresponding output of the operation (RMSNorm-M or LayerNorm),
> applying an orthogonal transformation of the same input **does not** result in an orthogonal
> transformation of the output.
> Whereas RMSNorm with a single scale (RMSNorm-S) is “basis-independent”,
> i.e. given an input and corresponding output of RMSNorm-S, applying an orthogonal transformation
> of the same input **does** result in an orthogonal transformation of the output.
> In our work we aim to eliminate causes of outliers and so we make this change.
> Rows 6 & 7 of Table 2 in the main paper shows that RMSNorm-S has no performance disadvantage
> over RMSNorm-M. We agree that this should be made more explicit in the paper and will do so in the final version.

---

> ### Author Response · Authors · 2024-11-24
> **Continuing the response**
>
> **Weakness 3: Softmax-1 and casual masking**
>
> Thank you for your question.
> The key issue with the standard softmax function lies in its design—it inherently forces the
> model to attend somewhere.
> Even small differences in logits can result in disproportionately large differences in attention scores after applying softmax,
> leading to problematic behaviors such as strong attention on the first token. While softmax can theoretically produce a uniform attention distribution,
> achieving this in practice is challenging due to the exponential transformation applied to the logits.
> Small imbalances in the logits can amplify into significant disparities in attention.
> The softmax-1 method addresses this by adding a corrective mechanism,
> effectively allowing low attention scores across all tokens without forcing the model to focus unnecessarily on specific tokens.
> This provides a more balanced inductive bias, better aligning with the goal of avoiding ill-behaved attention patterns like the disproportionate focus on the first token.
> Softmax-1 allows an attention head to "do-nothing" by actually attending to nothing,
> whereas standard softmax is forced to learn a "do-nothing" behavior such as by attending
> to the first token strongly (as observed) or by attending uniformly across all tokens (proposed by the reviewer but not observed in practice).
> Moreover, causal masking is used in language models. This means in query token position (1, 2, 3, 4) softmax is calculated over (1, 2, 3, 4) tokens and so a uniform distribution still yields relatively large values of (1, 1 / 2, 1 / 3, 1 / 4) for canonical softmax. Softmax-1 allows uniformly small attention values even at positions (1, 2, 3, 4) providing the calculated attention similarity (pre-exp) is low. Canonical softmax does not allow this (i.e. the attention
> scores of the query tokens are always large due to causal masking).
> In L208 "having low attention scores everywhere" refers to the ability to have low attention scores across all query positions which is not possible with canonical softmax in the early positions of the sequence. We have clarified this in the updated manuscript.
>
> **Weakness 4: OrthoAdam**
>
> Your outline is correct with perhaps the causality reversed.
> We see empirically that the dynamics of Adam cause the model to have
> large weight values in specific dimensions which in turn causes the model to have large outlier activations in these specific dimensions (whereas using SGD does not).
> Using OrthoAdam prevents the model from having large weight values in specific dimensions and therefore prevents the model from having large activations outlier activations in any dimension.
>
> **Weakness 5: Typos and Table 2**
>
> Thanks for this advice, we added xs and followed your advice for typos in L195, L306 and rewrote the sentences in L313 and L420 (blue in the updated manuscript).
>
> **Question 1: Llama2**
>
> We note that while Llama-2 is a bit more robust to outliers, the performance there also significantly degrades especially under coarse and moderate quantization strategies. More precisely, the perplexity increases by 26.22, 7.07, 0.3 and 4.1 perplexity points under coarse, moderate, fine and 4-bit strategies. Except under 4-bit strategies, the performance degradation is similar to that of GPT-2 models, and even under 4-bit it still performs worse than 1.4 perplexity points compared to our method.
>
> **Question 2: Causal Masking**
>
> The main idea here is to remove the casual masking for the first 10 tokens. This means
> all query tokens can attend to the first 10 key tokens, whereas
> with causal masking, the only key token that all query tokens can attend to is the first token.
> In other words, this means that the first 10 tokens are treated equally.
> When we do this, then instead of the first token, one of the first 10 tokens (with almost uniform probability) dominates the attention calculation. If we had plot an attention map for a model with a different initialisation any one of the first 10 tokens would have come to dominate the attention calculation. With proper causal masking (as is typical), the first token always dominates the attention calculation as it is the only key token all query tokens can attend to (i.e. it is priviledged). We hope this clarifies the discussion starting at L222, and we have updated the manuscript to make this clearer.

---

> > ### Comment · Reviewer_hVNT · 2024-11-26
> >
> > Thank you for the detailed response.
> >
> > My concerns have mostly been resolved.
> >
> > The main thing missing is the novelty framing of softmax-1, which is overstated given that it was already introduced in a different paper.

---

> ### Author Response · Authors · 2024-11-27
> **Thank you and clarification**
>
> We thank the reviewer for acknowledging their concerns have been resolved.
>
> [1] - As we mentioned in the previous response, the details of the implementation of softmax-1 in [1] are lacking (the code for the experiments in Table 3 is not available) and so understanding the potential difference in findings is difficult. Moreover, the experiments conducted in Table 3 involve "streaming perplexity" i.e. evaluating the model beyond the context length for which it is trained by dropping tokens. This is distinct to our experiments in which all models are tested using the context length for which they were trained. The difference in evaluations may be a source of the apparent difference in findings.
>
> Our intention is to propose softmax-1 as a solution to the issue of first token dominance in attention (but not outlier activations), rather than to propose the exact formulation of the equation. Where we may have blurred this distinction we shall adjust the manuscript accordingly.
>
> [1] Xiao et al. Efficient Streaming Language Models with Attention Sinks

---

### Official Review · Reviewer_rxLH · 2024-10-30

**Soundness:** 2
**Presentation:** 3
**Contribution:** 4
**Rating:** 6
**Confidence:** 4

**Summary:**

This work studies two unexpected phenomena that occur in Transformers-based language models: the dominance of first tokens in the self-attention maps, and the presence of outlier dimensions throughout the hidden representations of these models. The authors propose a different mitigation technique for each phenomenon, and successfully mitigate both first-token dominance and outlier dimensions. They finally show that combining both mitigation techniques yields language models that can be quantized easily, with minimal performance loss.

The first mitigation technique relies on a softmax-1 activation function, that adds a bias to the denominator of the softmax function to allow for null rows in the attention map. The authors conduct experiments that show that this technique solves the first-token dominance problem but not outlier dimensions. This second problem is mitigated by OrthoAdam, a slight modification to the Adam optimizer that compute gradients in a given orthonormal basis to avoid specific canonical dimensions being used for massive coefficients.

**Strengths:**

This paper successfully addresses two issues that have been identified by extensive literature in the language modeling field when training Transformers-based models. It provides relevant methods to mitigate these issues and show that such methods can help with performance
 (slightly) and significantly eases the quantization process.
The paper is well-written and pedagogical. The proposed solutions are elegant and simple to implement.
Overall, this paper paves the way for a better understanding of observed self-attention maps and for more expressive inductive biases for language models. It also provides immediate and substantial benefits for the field of quantization.

**Weaknesses:**

The main weakness of the paper is its failure to support its main theoretical claim. From the abstract on, the authors argue that they identify "Adam as the primary contributor to the large outlier activations", but they fail to properly discuss the theoretical background of this claim. They support this claim with the following arguments, which all have flaws:
- *the optimiser tracks moments in the same basis as the model parameters* : this reason does not explain why the optimiser is pushing for these outlier activations in gradients and their moments, but only why these outlier dimensions are passed on to the model's weights;
- *Adam and RMSProp have high[er] kurtosis [than SGD]* : this indeed shows that Adam and RMSProp are particularly accelerating the emergence of outlier activations, but it should be noted that they are also accelerating convergence. Hence, this could simply imply that Adam and RMSProp are efficiently facilitating the emergence of outlier activations that could still **be caused by another element of the system** that would rely on outlier activations to increase performance. In other words, the "outlier activation" state could be a favorable state for the model in terms of performance because of another component, and this state could be reached more quickly/easily using Adam and RMSProp.

It could be argued that the fact that OrthoAdam mitigates the outlier dimension problem proves that Adam is the culprit for such a phenomenon. Nevertheless, OrthoAdam is almost explicitly designed to avoid outlier dimensions *in the canonical basis*, as discussed in section 4.2, and could be seen as a variation of Adam that enforces the absence of outlier dimensions. Hence, this just shows that it is possible to train a model under this constraint without substantial performance decrease, but it does not show that Adam is the key **causal** element in the emergence of outlier dimensions.

Another limitation of the paper lies in some parameters and models that were used to validate the approaches:
- The models were trained using a sequence length of 256, which questions the scalability of softmax-1 to longer sequences, as sequence length may be a crucial factor for this activation function;
- All resulting models seem to have a noticeably high perplexity on the validation set. As a comparison, the 70M Pythia model reaches approximately a 3.2 loss (=24 perplexity) at half-training (~150B tokens), and the 410M version reaches 2.2 for cross-entropy (=9 perplexity)(https://wandb.ai/eleutherai/pythia-extra-seeds/reports/Some-loss-curves-for-smaller-Pythia-models--Vmlldzo2NTkxNDIw). This is significantly smaller than what is described in the paper for equivalent models, which raises questions about implementation, hyperparameter choices and evaluation.

Finally, the authors do not include a discussion on training dynamics, showing how the models converge during training, and they do not provide an analysis of OrthoAdam in terms of memory requirements and training latency, which may question the scalability and technical relevance of the method. My intuition is that the cost should be tolerable, but I think this should be mentioned in the paper.

**Questions:**

- The idea of using an orthogonal basis $Q$ in Adam is very elegant, especially in the context of quantization. Nevertheless, from a modeling/theoretical viewpoint, it would be interesting to check whether the models are still using outlier "dimensions" in this orthogonal basis, that is if the projection of their activations in such an orthogonal basis still have outlier coefficients. Did you conduct such experiments? In other words, did you verify that the models are not still relying on outlier "dimensions" in a different basis than the canonical one?
- It is common to decrease the learning rates for larger models. Why did you choose a single learning rate for the different model sizes?

**Typos / Remarks**

L419 : this sentence (*Note that only linear layers...*) seems a bit unformal.

L459: *but still remain high* -> *which remains high* ?

---

> ### Author Response · Authors · 2024-11-24
> **Response to Reviewer rxLH**
>
> We thank the reviewer for finding our paper **well-written**, **pedagogical**, **elegant** and **simple**. In particular, we are very excited that the reviewer finds the contribution of the paper **excellent**. Below, we address the main concerns of the reviewer.
>
> We apologize in advance for a very long rebuttal. We did so to respond to all the points raised by the reviewer and do all the needed experiments, in order to remove any doubt about the soundness of the paper.
>
> **Soundness: Results being lower than Pythia models**
>
> We thank the reviewer for raising this point. At the same time, we would like to clarify that a) Pythia models are much more modern than GPT-2, so in a general setting they outperform GPT-2; b) (far more importantly) the results the reviewer points out are from a Pythia model trained on Pile dataset and evaluated on Pile dataset. On the other hand, our GPT-2 models have been trained and validated in C4 dataset. There aren't many reasons to believe that the numbers should much each other.
> In any case, to convince the reviewer that our training is valid, we take our trained models and evaluate them in Pile dataset. We also do the reverse, evaluate Pythia models in the C4 dataset.
> We provide below the results of Pythia and our models, with similar sizes, in the Pile dataset:
>
> | Model Pythia | Perplexity | Model GPT-2 | Perplexity |
> |--------------|------------|-------------|------------|
> | 70M          | **37.08**  | 60M         | 48.09      |
> | 160M         | **11.86**  | 130M        | 36.88      |
> | 410M         | **8.96**   | 350M        | 27.86      |
> | 1B           | **9.28**   | 1.4B        | 22.42      |
>
> Pythia models outperform GPT-2 models. We argue that this is because having been trained in Pile dataset, while our models have been trained in C4 dataset. If we compare the models in C4 dataset, we get a very different picture:
>
> | Model Pythia | Perplexity | Model GPT-2 | Perplexity |
> |--------------|------------|-------------|------------|
> | 70M          | 55.17        | 60M         | **31.9**   |
> | 160M         | 35.68        | 130M        | **22.9**   |
> | 410M         | 24.03      | 350M        | **16.4**   |
> | 1B           | 20.52      | 1.4B        | **13.4**   |
>
> As we clearly show, our models significantly outperform Pythia models. Which is not a surprise, the training and validation distributions are similar, be it on Pile, or in C4, so the methods trained in train split of a dataset will perform better in the val split of the same dataset.
>
> **Soundness: Longer context**
>
> Thank you for this interesting question. We trained a 60M model and a 130M vanilla model and our corresponding models with 512 and 1024 context, showing the results below:
> | Model_Size-Context | Setup   | Full   | Coarse | △ (Coarse) | Moderate | △ (Moderate) | Fine  | △ (Fine) |
> |------------|---------|--------|--------|------------|----------|--------------|-------|----------|
> | 60M-256        | Vanilla  | 31.88  | 43.53  | 11.65  | 34.87    | 2.99    | 32.15  | 0.27   |
> | 60M-512        | Vanilla  | 32.66  | 48.55  |  15.89 | 37.24    | 4.58    | 33.07  | 0.41   |
> | 60M-1024      | Vanilla  | 33.52  | 57.68  |  24.16 | 38.22    | 4.70    |  33.80 |  0.28  |
> | 60M-256        | S1+OA | 31.93  | 32.46  | 0.53    |32.32     | 0.39    | 32.00  | 0.07   |
> | 60M-512        | S1+OA | 31.83  | 31.89  | 0.47    | 32.18   | 0.35    | 32.30 | 0.06     |
> | 60M-1024      | S1+OA | 32.25  | 32.85  | 0.60    |  32.73  |  0.48   | 32.32 | 0.07     |
> | 130M-256      | Vanilla | 22.89   | 46.49  | 23.60  | 28.31   |  5.42   | 23.07 |  0.18    |
> | 130M-512      | Vanilla | 22.80  | 42.34 | 19.54    | 28.14   | 5.34    | 22.98 | 0.18     |
> | 130M-1024    |Vanilla |  22.93  | 38.78 | 15.85 | 29.04 | 6.11 | 23.16  | 0.23 |
> | 130M-256      | S1+OA   | 22.78 | 23.21  | 0.43       | 23.10    | 0.32         | 22.83 | 0.05     |
> | 130M-512      | S1+OA   | 22.73  | 23.16 | 0.43       | 23.04    | 0.31         | 22.79 | 0.06     |
> | 130M-1024      | S1+OA   | 23.87  | 23.28 | 0.41       | 23.19    | 0.32         | 22.94 | 0.07     |
>
> As can be seen, our model is very robust when we increase the sequence length, showing no noticeable performance drop in perplexity be it under the general setting, or when quantized. In all cases, especially under quantization schemes, our method outperforms the vanilla one when trained with longer sequences. We also incorporated these results in the updated manuscript, Appendix C.

---

> ### Author Response · Authors · 2024-11-24
> **Continuing the response**
>
> **Soundness: Decreasing the learning rate for larger models**
>
> Indeed, it is common to lower the learning rates for larger models (see LLama 3.1 models). However, we note that our models despite increasing in size, do not reach extremely large number of parameters which might require lower learning rates for the training to be correct. Furthermore, to ensure the reviewer that our training is correct, we show that our models reach significantly better results (lower perplexity) compared to GPT-2 models from OpenAI (https://huggingface.co/openai-community). We show the results in the table below.
>
> | Model              | GPT-2-OpenAI       | GPT-2-ours | GPT-2-ours+S1+OA
> |--------------------|--------------|--------------| --------------|
> | GPT-2-60M  | 37.81 |    31.88   |  **31.83**   |
> | GPT-2-130M  | 28.85  |  22.89     |  **22.78**   |
> | GPT-2-350M  | 25.16 |  16.37     |  **16.31**   |
> | GPT-2-1.4B | 23.21  |  13.44     | **13.33**    |
>
> **Showing the Time and memory performance**
> We thank the reviewer for raising this point. Indeed, there is some small tolerable cost both in time and memory. We give the results below, and we further added them in the paper's appendix. In all cases, the effective batch size is 64, with the largest model having 32 batch size with gradient accumulation set to 2. We added these results in a new Table 4, in the updated manuscript.
>
> | Model              | Speed        | VRAM per GPU |
> |--------------------|--------------|--------------|
> | GPT-2-60M-Vanilla  | 14 iter/sec  | 16.4GB       |
> | GPT-2-60M-S1+OA    | 12 iter/sec  | 16.8GB       |
> | GPT-2-130M-Vanilla | 7.5 iter/sec | 22.6GB       |
> | GPT-2-130M-S1+OA   | 6.0 iter/sec | 23.3GB       |
> | GPT-2-350M-Vanilla | 3.3 iter/sec | 46.6GB       |
> | GPT-2-350M-S1+OA   | 3.0 iter/sec | 47.3GB       |
> | GPT-2-1.4B-Vanilla | 1.1 iter/sec | 61.9GB       |
> | GPT-2-1.4B-S1+OA   | 1.0 iter/sec | 65.0GB       |
>
> Regarding the training dynamics of OrthoAdam compared to Adam, Appendix H of the original submission (Appendix L in the updated manuscript) contains the training curves of
> all models we train, demomstrating that OrthoAdam converges along a similar trajectory to Adam.
>
> **Typos**
>
> Thanks for this advice. We have updated the second typo based on your suggestion and rewritten the first sentence/typo to "We only quantize the linear layers, while the embeddings, normalisation layers and softmax activations are not quantised".

---

> > ### Author Response · Authors · 2024-11-24
> > **Continuing the response**
> >
> > ### Weakness 1 - Adam as the cause of outlier activations
> >
> > We have put this weakness separate from the other ones because it requires a much longer discussion (and several experiments).
> >
> > Our study of outlier activations in language model hidden states suggests that some aspects of training are "basis-dependent." Basis-independent functions, invariant under orthogonal transformations, ensure f(Qx)=Qf(x) for any orthogonal matrix Q. We mitigate basis-dependent effects by removing biases from linear layers and employing single-scale RMSNorm, as both affine transformations and multi-scale RMSNorm are basis-dependent.
> >
> > While SGD and SGD with momentum are basis-independent, Adam and RMSProp are not, as their updates depend on the element-wise second-order gradient moments. This adaptive scaling can amplify certain weights disproportionately, particularly early in training when moments are poorly calibrated. These disproportionate weights lead to outlier activations.
> >
> > To address this, OrthoAdam computes gradient moments in random orthogonal bases, ensuring updates are evenly scaled, preventing extreme outlier weights. Analysis of GPT2-130M shows that large outlier weights in MLP block output layers correspond to small second-order gradient moment outliers, validating our hypothesis.
> >
> > At the interest of space in the rebuttal, we have given an **extended, more formal, and mathematical explanation** in the updated manuscript, Appendix E, and show plots in Figures 4 (canonical softmax and Adma) and 5 (Softmax-1 + OrthoAdam) there.
> >
> > **Adam Converges Faster than SGD**
> >
> > We agree with the reviewer that Adam and RMSProp converge faster than SGD.
> > We consider this in our ablation of optimizer choice in Table 4 of the submitted paper
> > (Table 5 in the updated manuscript). Therefore to ensure SGD trained models are similarly converged to Adam trained models, we train SGD models for 8x the number of steps as Adam/RMSProp trained models.
> > The model trained with SGD w/ momentum yields a lower evaluation perplexity than the models
> > trained with Adam and RMSProp. However, despite performing better and being similarly converged, the SGD w/ momentum model does not exhibit the same outlier activations as the Adam and RMSProp trained models. Therefore we respectfully disagree with the reviewer that the convergence speed
> > is the main factor in the emergence of outlier activations.
> >
> > **OrthoAdam is designed to avoid outliers**
> >
> > We agree with the reviewer that OrthoAdam is designed to avoid outlier dimensions.
> > However we believe our results show that their is no intrinsic need for a performant model to have outlier dimensions, we train a model (with OrthoAdam) that follows the same trajectory as the Adam trained model but without the outlier activations.
> > We argue that Adam pushes the model to have these outlier activations as a characteristic of the
> > optimizer, rather than any other component of the system.
> > To further back up our claim, we note that previous work
> > has observed outliers in computer vision models trained with Adam see Figure 3 Bondarenko et al. (https://arxiv.org/pdf/2306.12929) which shows Vision Transformers trained with Adam have outlier activations in the hidden states. We maintain that the per-parameter learning rate scaling of Adam is the cause of the outlier activations in the hidden states of language models.
> > Despite our work, we welcome other suggestions for the cause of the outlier activations aside from eliminating the basis-dependent effects of the model (RMSNorm-S) and optimizer (OrthoAdam).
> >
> > **Question 1**
> >
> > We investigated transforming the output of each layer (i.e. the added activations to the hidden states for said layer) into
> > the orthogonal basis used by that output layer in OrthoAdam.
> > We use our GPT2-350M model trained with Softmax-1 and OrthoAdam.
> > Given layer $i$ has output $\mathbf{X}_i \in \mathbb{R}^{L \times D}$ and the orthogonal basis for the output layer in OrthoAdam is $\mathbf{Q}_i \in \mathcal{O}^D$, where $L$ is the sequence length, $D$ is the hidden dimension and $\mathcal{O}^D$ is the set of $D \times D$ orthogonal matrices.
> > We plot the activation kurtosis of $\mathbf{X}_i$, $\mathbf{X}_i \mathbf{Q}_i$, $\mathbf{X}_i \mathbf{Q}_i^{T}$ and $\mathbf{X}_i \mathbf{V}_i$
> > where $\mathbf{V}_i$ is the right singular vectors of $\mathbf{X}_i$. Using $\mathbf{V}_i$ as a transformed basis gives a baseline for how large one could increase
> > the kurtosis of the activations by transforming them into an orthogonal basis. Additionally we give the activation kurtosis for our GPT2-350M vanilla model in Figure 6 of the updated manuscript (Appendix F).

---

> ### Comment · Reviewer_rxLH · 2024-11-25
> **Discussion**
>
> Thank you for the detailed answer. I particularly appreciate the care that was given to additional experiments and their relevance to the questions asked. Here are my comments on the different points:
>
> - **Pythia perplexity**: Thank you for conducting these experiments. I simply did not expect C4 and The Pile's domains to be different enough so that the perplexities would differ that much. This can also be explained by these smaller models being less robust to slight domain changes.
> - **Longer context**: These are interesting results. If not for the space limitation, I think you should add a mention/discussion in the core of the paper. The fact that quantization is still (almost) lossless for small models with larger context makes a stronger argument for your method. Yet, one interesting data point is the higher perplexity of the 130M model before quantization. This raises concerns for larger context length or model sizes.
> - **Learning rates**: My point is not that the training is not *correct*, but instead that it may be suboptimal in your own data/architecture/compute regime. Hence, I am not convinced by the comparison with OpenAI's GPT2, especially as I believe the training data is different which makes it difficult to directly compare perplexities (cf. previous point).
> - **Overheads**: I appreciate the reporting of these numbers. The overhead (10-15% time, 5% memory) is actually less negligible than I would have assumed. Nevertheless, it is still tolerable. Otherwise, the training curves are reported with training steps on the x-axis which does not allow for compute overhead analysis. Moreover, these training curves picture something that is not clear from the initial reading of the paper: OrthoAdam used alone seems to deteriorate the performance of the models. Although it is not prohibitive, I think it could be stressed out more clearly in the conclusions of the paper.
> - **On the claim that Adam causes outliers**: Thank you for the thorough addition to the original work. This analysis strengthens your claims. In my opinion, it is not exactly clear what to conclude from Appendix F, and it could be more informative if you provided a clear analysis of these results.
>
> Overall, I still find the main contribution of this paper very promising (it can be summarized as : transforming gradients using orthogonal bases to make weights distributions more suitable for quantization). Nevertheless, I am a bit puzzled by the soundness of the experiments and their analysis. In the rebuttal, the authors have added experiments, but I feel that they failed to connect them to the core of the paper to expose limitations, raise questions for future work or nuance their conclusions. Among points that are missing in the paper:
> - A compute-equivalent analysis of perplexity (e.g. a training curve with training time / FLOPS on the x-axis)
> - Comments on the performance degradation caused by the OrthoAdam optimizer (when used by itself)
> - A reference to the comments from Appendix F.
>
> **All-in-all, the authors conducted some convincing experiments that address my concerns and improve the soundness of their claims. However, some efforts could be made to improve the "connectivity" between parts of the paper and appendices, so that all elements can be combined to lead to solid and nuanced conclusions. Thus, I trust the authors to improve this aspect of the paper in the final version, and I improve my score in the meantime.**
>
> **Remark**
> - The first name of one of the authors was left visible in the legend of Figure 5. Overall (and this is understandable), the appendices still contain some typos.

---

> ### Author Response · Authors · 2024-11-25
> **Thank you**
>
> Thank you for the detailed reply and for raising your score.
>
> 1) **Pythia** - indeed we were quite surprised to see the massive differences with regards to Pythia. As you said, it can be attributed to models being not that big.
>
> 2) **Longer context** - We will follow your suggestion and put these results in the main manuscript. It was very hard to do so in the limited amount of time we had and still be within 10 pages, but we will adjust the paper to contain these results for an eventual camera-ready version of the paper.
>
> 3) **Learning rate** - It might well be the case that our learning rate for larger model is suboptimal, but even if that is the case, the core findings of the paper will not change. At best, the larger models might have a slightly lower (circa 1 perplexity point) and all of them will get shifted in the same direction.
>
> 4) **Overheads** - that is a fair point, but on the other hand, we can see the overheads getting smaller with the size of the model increasing. At the end of the day, we think this overhead is tolerable considering the benefits (effectively, free quantization).
>
> 5) **Adam** - we will connect better the paper with the Appendix.
>
> Remark: Thanks for pointing this out, we have immediately updated the manuscript and changed the captioning.
>
> We thank the reviewer again for increasing their score, for finding the contributions of the paper very promising, and we are happy to further discuss with the reviewer about the paper.

---

### Official Review · Reviewer_SMxg · 2024-11-04

**Soundness:** 2
**Presentation:** 3
**Contribution:** 1
**Rating:** 5
**Confidence:** 4

**Summary:**

This paper studies two phenomena in transformer-based models. The first is the strong dominance of the first token in the attention maps, and the second is the presence of outlier activations in the hidden states. To address these two issues, the paper proposes a new softmax philosophy and a new optimizer. The experiments show the advantages of the proposed methods under the quantization scenario.

**Strengths:**

1. The paper conducts a series of experiments to verify the phenomena, which are convincing.

2. The proposed new softmax and optimizer are simple and effective.

3. The experiments under the quantization scenario show the usefulness of the study, which potentially enlightens future research.

**Weaknesses:**

1. The paper states that the study is on popular large language models (LLMs). However, the experiments in the paper exclude large language models, considering that the largest-sized one in the experiments is GPT2-1.4B, not reaching the bar of a regular sized LLM (>7B). It makes the results of the paper less convincing.

2. I am not convinced that the highlighted two issues in the paper are critical for recent LMs. From the experiments in the paper, one can only see the promising usage under 4-bit/8-bit quant; in other scenarios, still not clear.

3. The paper reports PPL and other metrics to access the LM performances. However, it is not clear whether the resultant models perform well on downstream tasks and instruction following, which are important aspects to evaluate an LM/LLM. Therefore, it is not positive that the study will greatly contribute to the community.

4. Keep up with the last point, the paper does not consider (or explain) whether the study still holds under the instruction-following scenario. This situation is different from doing language modeling; or it is still the same, since the first generated token is not the start of the input sentence.

5. Minorly, it is also not mentioned in the paper the relationship of the two phenomena. It seems that they belong to two independent research aspects.

**Questions:**

1. Why do the authors train the LLaMA2-130M model? It is strange to me.

2. Can the authors explain the how the case would change where the model is used to generate following some instruction. Will the LM attend strongly to the first generated token?


---
After rebuttal
Thanks for the authors' detailed responses. My concerns are below.

1. My first concern is the **originality of the softmax-1**. However, I failed to find the original paper when I first read the paper until I see the comments.

2. The second concern is the **contribution of the paper**. In addition, the first token attention issue was also discussed in [1]. I am afraid the story of the paper on 1 and 2 is overstated in the submission. That is why I give a lower contribution score.

3. The authors state "large language models" in their submission title, which is confusing. While the authors provide the LLaMA-8B results in the rebuttal, only a 8B LLM can not represent the entire group of LLMs. The situation can be very different when the model size comes to 70B or even 130B. A more suitable statement for the current submission would be **"autoregressive language models"**.

Therefore, I decide to keep my score.

[1] Efficient Streaming Language Models with Attention Sinks

---

> ### Author Response · Authors · 2024-11-24
> **Response to Reviewer SMxg**
>
> We thank the reviewer for finding our study **convincing**, **simple and effective** and useful. Below, we address the main concerns of the reviewer.
>
> **Regarding weakness 1 - choice of GPT-2**
> 1) Choice of GPT-2 Models: GPT-2 models are among the best understood LLMs, sharing key characteristics with more modern models (large, causal, decoder-only transformers trained on extensive datasets).
> 2) Computational Constraints: Pretraining larger models (e.g., 7B) on datasets like C4 is **prohibitively expensive given our computational resources**.
> 3) Model Size: We respectfully disagree with the notion that 1.4B models are not LLMs. Models of this size are comparable to other modern LLMs, such as Gemma (Google), Phi (Microsoft), and TinyLlama (Meta).
> 4) Findings Across Model Scales: Our results hold consistently across models ranging from tens of millions of parameters to 1.4B, affirming their generalizability.
> Including results from larger models could certainly add further depth to the paper, but we are confident that our findings remain robust. **Additional inference-only results included below further demonstrate that similar issues (e.g., first token attention and outliers) persist in larger models**. For example, we checked the first-token attendance in Llama-3.1-8B (https://huggingface.co/meta-llama/Llama-3.1-8B) and found out that the first-token attendance in fact increased to **95.45%**. Furthermore, we also checked the cumulative sum of attention to the first token and found it out to be at **73.49%**. In other words, 73.49% of the entire attention in Llama-3.1-8B is in the first token. This can be interpreted that the larger the network, the more specialized the heads are, and most of the heads will simply do nothing. Attending on the first token is the mechanism the Transformer has developed to learn to do nothing. We got similar results for outliers, measured by kurtosis. We integrated these new results in the updated manuscript, Appendix D.
>
> **Regarding weakness 2 - quantization**
>
> The main focus of our paper is to highlight specific training issues in Transformers, which, despite being foundational to recent AI successes, remain poorly understood. Our study uncovers unexpected and nonsensical problems in their training, proposes plausible explanations, and offers straightforward solutions to address them.
> Quantization is not the central result of the paper but rather a practical demonstration of the potential applications of our findings—likely, there are many more to explore.
>
> **Regarding weaknesses 3 and 4, and Question 2 - instruction tuning**
>
> We have merged these two weaknesses and Q2 as they address a similar concern. Supervised fine-tuning shares many similarities with unsupervised pretraining, as both involve training the network using the same optimizer and loss functions. The key difference lies in the objective: Instead of next-token prediction, the model predicts a supervised ground truth, whether provided by humans or other LLMs.
> To address this, we have conducted additional experiments with networks trained in supervised settings. Because of computational resources, we took two pretrained models: Meta-LLama3.1-8B (https://huggingface.co/meta-llama/Llama-3.1-8B) and Meta-Llama3.1-8B-Instruct  (https://huggingface.co/meta-llama/Llama-3.1-8B-Instruct), and we got the results:
> | Method                | %First_attention | Cumulative_Sum_first_token | 1st token kurtosis | Average token kurtosis
> |-----------------------|------------------|----------------------------|----------------------------|----------------------------|
> | Llama-3.1-8B          | 95.45            | 73.49                      | 1227 |  55
> | Llama-3.1-8B-Instruct | 95.53            | 70.13                      | 1228 | 69
>
> As can be seen the first-token attention remains roughly the same from the model trained on next-token prediction to the Instruct model, trained in instruction data. Thus, this problem clearly occurs also for models trained in instruction data. Furthermore, we also compare the 1st token kurtosis and the average kurtosis showing that the Instruct version actually increases the kurtosis, thus applying our modifications in Instruct setting would potentially give an even bigger boost. We integrated these new results in the updated manuscript, Appendix D.
> We hope this clarifies and strengthens the relevance of our study to larger LLMs and LLMs trained on instruction tuning.

---

> > ### Author Response · Authors · 2024-11-24
> > **Continuing the response**
> >
> > **Regarding Weakness 5 - are the phenomena related?**
> >
> > The two phenomena are indeed related. While first-token attention can be effectively mitigated with softmax_1, addressing outliers requires the combined application of both softmax_1 and OrthoAdam. Individually, each method reduces outliers to some extent, but they persist unless both are applied. For instance, in the GPT-2 60M model (as shown in Table 2), the kurtosis starts at 313.8. With softmax_1, it reduces to 105.6; with OrthoAdam, it decreases to 260.8. When both are applied, kurtosis drops to 7.6, effectively eliminating outliers. This pattern is consistent across other models, demonstrating the complementary nature of these solutions in resolving both issues. We hope this clarification highlights the connection between these phenomena!
> >
> > **Answer to Q1 - LLaMA2**
> > We trained the LLaMA2-130M model to demonstrate that our findings are not exclusive to GPT-2 but also manifest in other causal-decoder-only Transformers. LLaMA2-130M was chosen because it is relatively small and efficient to train, making it a practical choice for validating our observations across different architectures.

---

> > > ### Author Response · Authors · 2024-12-01
> > > **Results on instruction tuning (new)**
> > >
> > > To complement the previous experiment, considering the extension of the discussion period, we run a new experiment, doing instruction finetuning (supervised learning) in Alpaca dataset. We compare the results of a model trained with canonical softmax and Adam, compared to our method with softmax_1 and OrthoAdam. We present the results in GPT-2-130m models.
> > >
> > > | Method               |Perplexity full  | Perplexity coarse | △ (Coarse) | Moderate | △ (Moderate) | Fine  | △ (Fine) | %1st_attention | Cumulative Sum 1st token | 1st token kurtosis | Average token kurtosis
> > > |------|------|-----|---|-----|---|----|-----|---|-----|---|----|
> > > | GPT-2-130m | 18.33 | 31.03  | 12.07 | 21.09 | 2.76 | 18.49  | 0.16  | 65.57  | 41.49  | 564.75  | 69.31  |
> > > | GPT-2-130m + S1 + OA  | 18.29 | 18.31  | 0.02 | 18.31 | 0.02| 18.29  | 0.0  | 2.4  | 0.8  | 2.97  | 2.96  |
> > >
> > > We show that while both models reach roughly the same perplexity, only our model keeps the same perplexity under all three quantization schemes. On the other hand, the vanilla model drops for 12.07 points in coarse quantization, 2.76 in moderate quantization, and 0.16 in fine quantization.
> > >
> > > To evaluate that this is a direct effect of the first token attention and outliers, we also present the results in the percentage of maximum attention in the first token, the cumulative attention score of the first token, the 1st token kurtosis, and the average kurtosis. We show that in the vanilla model, the maximum attention is in 65.57% of cases in the first token, and the cumulative attention on the first token is 41.49%. In contrast, our method has maximum attention on the first token is 2.4% and the first token contributes to only 0.8% of the attention. Furthermore, while the vanilla model has kurtosis of 564.75 for the first token, and 69.31 for the average token, our method has 2.97 kurtosis for the first token, and 2.96 kurtosis for average token, very similar to the kurtosis of normal distribution (3).
> > >
> > > In this way, we empirically show that our findings of the main paper stand also for instruction tuning training. Models trained with instruction tuning drop in accuracy under quantization schemes because of their outliers, while our method remains stable and as we showed, has the same kurtosis as normal distribution and not high attention in the first token.
> > >
> > > Considering that this was the main concern of the reviewer, we hope that this experiment addresses it. We are very happy to further discuss in the remaining days of the discussion period if there is any remaining doubt from the reviewer.

---

> > > > ### Author Response · Authors · 2024-12-03
> > > > **Follow up on the rebbutal**
> > > >
> > > > Dear Reviewer,
> > > >
> > > > We hope, and think, that we have addressed all your concerns in our rebuttal, including additional experiments on larger models and instruction tuning, and updated the manuscript accordingly.
> > > >
> > > > We would be very happy to further incorporate any addition feedback in an eventual camera-ready version of the paper. Otherwise, we hope our responses might prompt a reconsideration of the score.
> > > >
> > > > The Authors

---

### Author Response · Authors · 2024-11-24
**General comments to reviewers**

We thank the reviewers for their constructive feedback. We are excited to see our study being called **effective and convincing** (Reviewer SMxg), **well-written, pedagogical and elegant** (Reviewer rxLH),  **interesting, important, well-executed and practical** (Reviewer hVNT). In the specific rebuttals, we addressed most of the points. For guidance, we summarize here the main points of the rebuttal. We also updated the manuscript, highlighting the changes in blue, and added 4 new sections in Appendix (C, D, E, F)

1) **Reviewer SMxg** biggest concern is that they are not convinced that the study is relevant for larger LLMs, and for LLMs trained in Instruction-Tuning datasets (supervised learning). To argue that larger LLMs, regardless if they are trained in unsupervised manner (next token prediction) or in supervised manner (instruction-tuning), show the same behavior, we provide results for Meta-LLama-3.1-8B and Meta-Llama-3.1-8B-Instruct. In both cases we observe that: a) the models heavily attend in the first token; b) the models have outliers; c) there is no noticeable difference in the behavior of the models regardless if they are trained in an unsupervised or a supervised manner.

2) In addition, based on the request of **Reviewer SMxg**, we also run experiments in GPT-2 for instruction tuning, showing that the exact same findings for next-token-prediction unsupervised learning stand also for supervised instruction-finetuning.

3) **Reviewer rxLH** asks for several clarifications and experiments for the optimizer states and OrthoAdam. We have given all of them in Appendix E, F and in the direct rebuttal to the reviewer.

4) **Reviewer rxLH** expresses some concern if our networks have been trained correctly and wonders why our results are lower than those of Pythia. We explain that while some of the mismatch can be attributed to Pythia being more modern, thus better performing networks, the main reason is because of the dataset mismatch. Pythia networks have been trained on Pile-train dataset and tested on Pile-eval dataset. Our GPT-2 networks have been trained on C4-train and evaluated on C4-eval dataset. A Pythia model trained on Pile outperforms a similar size GPT-2 model (trained on C4) when evaluated on Pile, but at the same time our GPT-2  models outperform Pythia models when evaluated on C4. We hope this clarifies the issue. Furthermore, we also show that our models reach better results than OpenAI GPT-2 models from HuggingFace.

5) **Reviewer rxLH** also asks the time and memory increase when using our method. We provide such results and integrate them on the manuscript, showing that our method comes with a tolerable increase in both.

6) **Reviewer rxLH** also expresses some concerns if our findings will stand for networks trained with larger sequence length. We provide results with sequence length 512 and 1024, showing that our findings stand there too.

7) **Reviewer hVNT** expresses some concerns with the understanding of casual masking and OrthoAdam. We address those points, and integrate them in the manuscript.

8) **Reviewer hVNT** asks for clarification on the differences between our work and StreamingLLM, and for a clarification with regards to RMSNorm-s. We provide direct clarifications for both of them under rebuttal's response.

We hope that our responses have clarified these issues and we believe that the new experiments have further solidify the results of the paper. We have integrated the changes in the updated manuscript. We are happy to further discuss with the reviewers if there is any remaining doubt, concern or unaddressed question.

---

### Public Comment · ~Nasim_Nour_Mohammed1 · 2024-11-28
**Softmax-1 is NOT your original method!**

In this paper, the authors claim the following:

> "Thus, we modify the softmax function to the following: ... Eq. (1)"

However, this modification of the softmax function was first proposed by Evan Miller in 2023, as discussed in his blog post [here](https://www.evanmiller.org/attention-is-off-by-one.html), specifically in the "Softmax One and Quiet Attention" section.

It is obvious that the authors had read Miller's blog as they have cited Miller et al.! In line 520, the authors mention Miller et al.’s study on activation outliers, but they do not disclose that Miller also introduced Softmax-1 in the same blog post.

This act appears to be an intentional attempt to mislead reviewers who may not be familiar with Miller's work. For the academic integrity, I respectfully urge the reviewers and chairs to re-evaluate the paper in light of this issue.

---

> ### Author Response · Authors · 2024-11-28
> **Response to accusation of an “intentional attempt to mislead reviewers”**
>
> We thank the reader for raising this point and are happy to provide clarification.
>
> First and categorically, there has not been any “attempt to mislead reviewers”.
> If we had an intention to mislead or obfuscate we simply would not have cited the blogpost by Miller [1], considering it is not a peer-reviewed paper and does not contain any experiments.
>
> Nevertheless, there are several distinctions between the proposals made by Miller in their blogpost and our work:
> 1. The blogpost hypothesises a solution (namely softmax-1) to the problem of outlier activation features, citing [2] as a starting point for understanding outlier activations in LLMs. The blogpost contains _zero_ experiments to empirically validate this hypothesised solution and presents _zero_ theory for why this hypothesised solution should work. Our work experimentally validates all aspects of removing the two phenomena addressed, validating softmax-1 and OrthoAdam across multiple architectures, models sizes and tasks.
>
> 2. We strongly note that our work _disproves_ the hypothesis of the blogpost. "Even though softmax1 is facially quite boring, I’m 99.44% sure that it will resolve the outlier feedback loop that’s making quantization the subject of cascades of research."; “But do let me know how those weight kurtoses and activation infinity norms are looking after a few runs.” — the implication of the work being that using softmax-1 (instead of canonical softmax) will _eliminate_ outlier activations. However, we show through experimental results in Table 2 of the original/updated manuscript (see the first two rows for GPT2-60M, GPT2-130M and Llama-130M) that softmax-1 _does not_ remove outlier activations (kurtosis and max activation remain similar to canonical softmax except for the first token).
>
> 3. We _provide an alternative hypothesis_ for outlier activations, propose a solution (namely OrthoAdam) and experimentally show that OrthoAdam combined with softmax-1 _does eliminate_ outlier activations (either alone is insufficient). See Table 2 and 4.
>
> 4. The blogpost by Miller offers no practical advantage of eliminating these outlier features beyond the aesthetic. In our work we propose our method as a solution to existing quantisation issues, and *validate* its utility.
>
> Our intention is to propose softmax-1 as a solution to the issue of first token dominance in attention (but not outlier activations), rather than to propose the exact formulation of the equation.
>
> Moreover, despite the same formulation, there are substantial conceptual and practical differences between Miller’s blogpost and our paper.
> Suggesting solutions to problems is good but has limited utility compared to work which provides alternative solutions to alternative problems, validation through experiments and practical demonstration.
>
> [1] Miller, Attention Is Off By One, 2023
>
> [2] Bondarenko et al., Quantizable Transformers: Removing Outliers by Helping Attention Heads Do Nothing, 2023

---

> > ### Public Comment · ~Nasim_Nour_Mohammed1 · 2024-11-29
> >
> > Thank you for your response. At least both of us can agree that the form of softmax-1 was first proposed by Miller, so claiming "we modify the softmax function to the following" is inaccurate. I acknowledge your contribution in applying and validating softmax-1, as well as proposing new interpretations, but the credit for inventing softmax-1 should be given to Miller et al.

---

### Meta-Review · Area_Chair_fW5o · 2024-12-14

**Metareview:**

The paper offers two main contributions the dominance of first tokens in the self-attention maps, and the presence of outlier dimensions throughout the hidden representations of these models.

Strengths:
Addresses known issues in the literature, offers practical techniques
Clearly written

Weaknesses:
The main issue with the paper was giving too little credit to previous works which was revised and should be easy to fix in the camera ready.
Limited experiments scope, that if addressed could make the results more significant or convincing (e.g., a focus on perplexity rather than downstream task).

**Additional Comments On Reviewer Discussion:**

Make sure to address the open suggestions from the reviewers and properly update the camera ready with all the rebuttal content.

---

### Decision · Program_Chairs · 2025-01-22

Accept (Poster)